# ATP hydrolysis by yeast Hsp104 determines protein aggregate dissolution and size in vivo

Udhayabhaskar Sathyanarayanan[1,6], Marina Musa[1,2,6], Peter Bou Dib [3], Nuno Raimundo [3], Ira Milosevic[1,4] & Anita Krisko [5✉]

Signs of proteostasis failure often entwine with those of metabolic stress at the cellular level. Here, we study protein sequestration during glucose deprivation-induced ATP decline in *Saccharomyces cerevisiae*. Using live-cell imaging, we find that sequestration of misfolded proteins and nascent polypeptides into two distinct compartments, stress granules, and Q-bodies, is triggered by the exhaustion of ATP. Both compartments readily dissolve in a PKA-dependent manner within minutes of glucose reintroduction and ATP level restoration. We identify the ATP hydrolase activity of Hsp104 disaggregase as the critical ATP-consuming process determining compartments abundance and size, even in optimal conditions. Sequestration of proteins into distinct compartments during acute metabolic stress and their retrieval during the recovery phase provide a competitive fitness advantage, likely promoting cell survival during stress.

[1] European Neuroscience Institute (ENI), Göttingen, Germany. [2] Mediterranean Institute for Life Sciences, Split, Croatia. [3] Universitätsmedizin Göttingen, Institut für Zellbiochemie, Göttingen, Germany. [4] Wellcome Centre for Human Genetics, Nuffield Department of Medicine, NIHR Oxford Biomedical Research Centre, University of Oxford, Oxford OX3 7BN, UK. [5] Department of Experimental Neurodegeneration, University Medical Center Göttingen, Göttingen, Germany. [6]These authors contributed equally: Udhayabhaskar Sathyanarayanan, Marina Musa. ✉email: anita.krisko@med.uni-goettingen.de

Cellular quality control requires the crosstalk between distinct pathways. Maintenance of cellular life relies on the quality control mechanisms supported by a constant supply of energy, most often in the form of ATP. Cells undergoing stress exhibit changes at different levels of their functioning, including transcriptional regulation, new protein synthesis, metabolic reprogramming, and even dramatic changes in subcellular organization[1–3].

Sequestering misfolded proteins into insoluble aggregates is a part of the cellular attempts to remain functional even in the conditions of proteotoxic stress. Their formation is proposed to be beneficial during heat shock for multiple reasons, the dominant one being the removal of toxic protein conformers from the soluble phase[4,5]. In budding yeast Saccharomyces cerevisiae, three major sequestration sites for misfolded proteins exist: IPOD (insoluble protein deposit), INQ (intranuclear quality control compartment), and cytosolic CytoQ (Q-bodies)[6–10]. IPOD is perivacuolar and predominantly sequesters amyloidogenic proteins[6]. INQ and Q-bodies are typically heat stress-induced deposits for misfolded proteins residing in the nucleus and the cytosol, respectively[7–9,11]. Moreover, stress granules (SGs) typically form during various stress conditions, such as arsenic exposure, as well as glucose deprivation[12]. They are dynamic compartments formed through interactions of proteins and RNA by phase separation in the cytosol[12]. Different types of protein aggregates can be distinguished based on the presence or absence of specific markers and chaperone complexes. However, Hsp70/Hsp104, acting as ATP-dependent disaggregation machinery, has a pivotal role in the growth of all aggregate types[13–15]. Protein aggregation has so far mainly been related to degenerative states such as advanced aging, age-related, and protein folding diseases[16,17]; however, its beneficial facets during stressful periods have also been demonstrated and discussed[5,18,19].

When cells face a loss of ATP, specific metabolic programs are triggered to enable cells to adapt to such conditions[20,21]. Typically, during glucose deprivation, global protein synthesis is deactivated, although the translation of specific proteins with critical roles in overcoming stress continues. Glycolysis undergoes repression, while the respiratory chain activity is enhanced, taking charge of ATP production during glucose deprivation[22]. Moreover, recently it has been suggested that metabolic enzymes, like glutamine synthase and CTP synthase, are deposited into filamentous structures in yeast and fruit fly in response to starvation[23,24].

Here, we study the mechanism of how the decline of cellular ATP, as well as its restoration, affect protein sequestration into distinct compartments. We found that glucose starvation-induced ATP decline promotes protein sequestration into two separate compartments; Q-bodies and stress granules (SG). We identify the ATPase activity of Hsp104 disaggregase nucleotide binding domain 1 (NBD1) and nucleotide binding domain 2 (NBD2) domains as the key ATP-consuming processes enabling aggregate dissolution. We put forward aggregate dissolution as the critical process that determines the steady-state aggregate size, rather than their assembly/growth by protein deposition. Depositing proteins into aggregates and their timely dissolution during recovery from the starvation stress provide a significant fitness advantage during glucose deprivation, which offers new perspectives on the role of protein sequestration into isolated deposits in cell adaptation to stress. Understanding the interplay between protein disaggregation and metabolic activity of the cell is of universal importance due to the association of protein aggregation with aging and many medical conditions.

## Results

**Increased protein aggregation accompanies ATP decline.** In optimal conditions, budding yeast grows in 2% glucose, which they mainly consume during their exponential growth stage[25]. Therefore, to gain insight into the protein aggregate abundance during the decline in cellular ATP, cells were exposed to glucose starvation in 0.2% and 0.02% glucose medium during the exponential growth stage (OD 0.4–0.6).

To study the extent of protein aggregation during glucose starvation, we used the endogenously expressed Hsp104, C-terminally tagged with the green fluorescent protein (GFP). We compared these results to the ones obtained via Hsp104-GFP fusion under the control of the constitutive promoter, TEF1, integrated into the yeast genome (see "Methods"). Employing both approaches allowed us to control for the starvation-induced upregulation in the Hsp104 level when regulated by the native promoter.

Based on the images obtained using spinning disk confocal microscopy (throughout the study), we quantified two parameters: (i) protein aggregate abundance was measured as a fraction of cells with one Hsp104-GFP focus, and with 2 or more such foci; and (ii) diameter of individual aggregates. When the shape of some aggregates was oval, we measured the longest dimension.

After exposure to 0.2% glucose for 90 min, the percentage of cells bearing one Hsp104-tagged aggregate increased from ~4% to ~40%, with ~25% cells displaying 2 or more aggregates (Fig. 1a). Aggregate sizes were estimated to group around a median of 800 nm, compared to those under 200 nm in diameter in 2% glucose (below the microscope resolution limit) ($p = 0.0005$, Kolmogorov–Smirnov test) (Fig. 1b). Similarly, in 0.02% glucose ~40% of cells displayed one, and almost 50% of cells displayed 2 or more aggregates (Fig. 1c). Their sizes grouped around a median of 1200 nm, with 12% of aggregates above 1500 nm ($p = 0.0003$, Kolmogorov–Smirnov test) (Fig. 1d).

In 0.2% glucose, ATP levels decreased >2-fold in the first 30 min and ~5-fold after 90 min (Supplementary Fig. 1a). In 0.02% glucose, the effect was even more pronounced: In the first hour, the ATP level decreased 10-fold and remained at this level up to 90 min from the onset of glucose starvation. This trend was accompanied by gene expression changes typical for glucose starvation, including repression of glycolysis, as well as upregulation of the TCA cycle and respiratory chain components, as observed previously[22]. Interestingly, the cells activated the responses typical for heat shock, including upregulation of Hsp104, Hsp82, Hsp70 (Ssa1), Hsp42, and Hsp26 (Supplementary Fig. 1b), in line with the results of Hahn and Thiele showing the activation of the heat shock factor during glucose starvation stress[26].

We further sought to test if the ATP decline during starvation was related to the increased abundance of protein aggregates. We exposed the cells in starvation conditions to external ATP, 60 min after the onset of starvation. Therefore, ATP was added to the culture 30 min before harvesting the cells. Dimethyl sulfoxide (DMSO) was added simultaneously to facilitate the entry of ATP into the cells. DMSO alone did not affect the Hsp104-GFP tagged aggregate abundance (Supplementary Fig. 2a, b). We confirmed that ATP entered the cells in the presence of DMSO, and the cellular ATP levels after supplementation of the starved cells were comparable to the control assessed in the same experiment (Supplementary Fig. 3a–c).

Supplementation with ATP during the last 30 min of cell starvation led to a dramatic decrease in aggregation extent with only ≈6 and 7% of cells bearing one aggregate in 0.2% and 0.02% glucose, respectively, and only 2% with 2 or more aggregates in both conditions (Fig. 1a, c). The aggregate size distribution in the presence of ATP, in 0.2% and 0.02% glucose, shifted back to the sizes characteristic of aggregates in optimal glucose with a diameter below the resolution limit of the microscope ($p =$

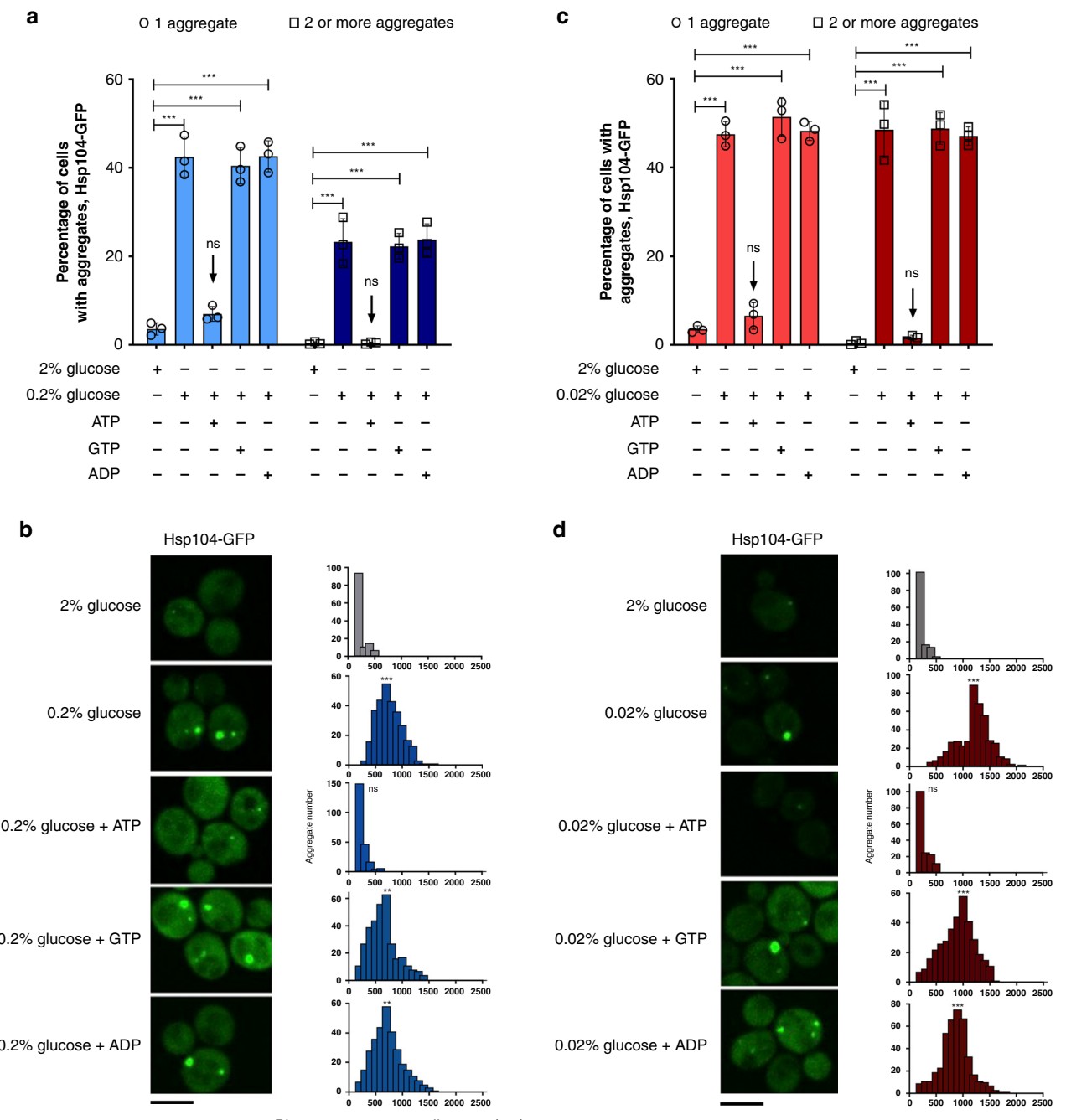

**Fig. 1 ATP prevents protein aggregate accumulation during glucose starvation. a** The percentage of cells bearing 1 (light blue) and 2 or more (dark blue) aggregates strongly increases during a 90-min starvation in 0.2% glucose. Treatment with ATP, but not GTP or ADP (all in the presence of DMSO), causes the fraction of cells bearing aggregates to return to levels comparable to control. **b** Mean aggregate diameter shifts to larger sizes during starvation in 0.2% glucose. The addition of ATP, but not GTP or ADP (all in the presence of DMSO), results in aggregates reverting to the sizes observed in control conditions. **c** The percentage of cells bearing 1 (light red) and 2 or more (dark red) aggregates strongly increases during a 90-min starvation in 0.02% glucose. Treatment with ATP, but not GTP or ADP (all in the presence of DMSO), causes the fraction of cells bearing aggregates to return to levels comparable to control. **d** Mean aggregate diameter shifts to larger sizes during starvation in 0.02% glucose. The addition of ATP, but not GTP or ADP, results in aggregates reverting to the sizes observed in control conditions. In panels **a** and **c**, $n = 1200$ cells were screened for aggregates starting from three independent exponential yeast cultures for each condition. Data are mean ± SD from at least three independent cultures, each performed in triplicate, and the means of technical replicates are represented as individual data points. ***$p < 0.001$; **$p < 0.01$; *$p < 0.05$ (one-way ANOVA plus Tukey post hoc). In panels **b** and **d** the data represent binned values of individual aggregate diameters for $N = 127, 297, 224, 330,$ and 320 aggregates from three independent exponential yeast cultures for each condition: 2% glucose, starvation, starvation + ATP, starvation + GTP, and starvation + ADP, respectively. The images are representative of three biological replicates, each in three technical replicates. Kolmogorov–Smirnov test was used to compare the statistical significance of the observed differences between each studied condition with the control (2% glucose) ***$p < 0.001$; **$p < 0.01$; *$p < 0.05$. The black bar represents 8 μm.

0.9999, $p = 0.9999$, respectively, Kolmogorov–Smirnov test) (Fig. 1b, d). Supplementation with ADP or GTP did not yield the same effect in 0.2% or 0.02% glucose, showing that they are not relevant energy sources for the observed processes. The aggregates were as abundant as without any supplementation during starvation in both starvation regimes (Fig. 1a, c). Aggregate median diameter was ~2.8x larger than the control sample median size in the presence of ADP and GTP, in both 0.2% and 0.02% glucose ($p = 0.002$, $p = 0.007$, respectively, in 0.2% glucose; $p = 0.0005$, $p = 0.0007$ in 0.02% glucose, Kolmogorov–Smirnov test) (Fig. 1b, d).

Next, we analyzed the extent of protein aggregation, with Hsp104-GFP under the control of the TEF1 promoter integrated into the genome. The results showed a similar increase in protein aggregate abundance in 0.2% and 0.02% glucose (Supplementary Fig. 4a and b). The fraction of cells containing one aggregate increased from 10% in optimal growth conditions to ~40% in 0.2% glucose and ~50% in 0.02% glucose. In optimal conditions, 5% of cells contained 2 or more aggregates. This fraction increased to ~40% in both 0.2% and 0.02% glucose (Supplementary Fig. 4a and b). These results were consistent with those described above, with Hsp104-GFP under the control of the native promoter, thereby excluding the possible effect of Hsp104 upregulation on the aggregate abundance in glucose starvation.

The aggregate sizes increased from a median of ~250 nm in optimal conditions to 650 nm and 800 nm in 0.2% and 0.02% glucose, respectively ($p = 0.0025$, $p = 0.0004$, respectively, Kolmogorov–Smirnov test) (Supplementary Fig. 4c). Therefore, even though aggregates formed during glucose starvation appeared somewhat smaller in the case of TEF1-regulated Hsp104-GFP expression, the trend of significant aggregate growth was consistent between the two cases. The presence of ATP during both starvation regimes (0.2% and 0.02% glucose) led to a decrease in aggregate sizes, comparable to the control ($p = 0.88$, $p = 79$, respectively, Kolmogorov–Smirnov test) (Supplementary Fig. 4c).

Under the presumption that the Hsp104-GFP expression level may influence the aggregate size, it is hard to decide which measurement is more accurate. Endogenous expression of Hsp104 underwent an increase during glucose starvation; however, the TEF1 promoter is regulated by glucose, and glucose starvation causes a slight decline in the promoter activity[27]. Using qPCR, we confirmed such trends in the Hsp104-GFP expression in the employed conditions (Supplementary Fig. 5). However, even with this in mind, TEF1 appears to be one of the best promoters for heterologous expression in yeast, with the glucose-dependent expression rather stable in the glucose concentration range employed here[27].

Consistent with previous observations, these results strongly suggested that cells undergoing acute glucose starvation, faced with a decline in ATP levels, experienced proteotoxic stress, and responded by the sequestration of proteins into aggregates[24,26]. The appearance of protein assemblies during glucose starvation has previously been reported for several cytosolic proteins, which did not colocalize with vacuoles, autophagosomes, or any major organelle[24].

Based on these results, we concluded that an ATP-consuming process determined the aggregate abundance and steady-state aggregate size. During glucose starvation-induced ATP decline, this process was unable to perform efficiently. We further set out to determine the identity and molecular mechanism of this ATP-consuming process and its involvement in protein aggregation.

**Aggregate dissolution relies on ATP hydrolysis by Hsp104.** Previous results strongly suggest that an ATP-consuming process

influences final aggregate size and abundance. Hsp104 consumes ATP for each disaggregation cycle (20 molecules per minute)[28]. It consists of two ATP-binding domains, NBD1 and NBD2. Each of these domains harbors Walker A and Walker B motifs, responsible for ATP binding and hydrolysis, respectively. Therefore, to investigate the importance of ATP hydrolysis by Hsp104 in determining the aggregation extent during glucose starvation, we employed two mutants of Hsp104, both in the Walker B motifs: Hsp104[E285Q] in the NBD1 domain, and Hsp104[E687Q] in the NBD2 domain[29]. These mutants are characterized by abolished ATP hydrolysis in the affected NBD while maintaining its binding. While Hsp104[E285Q] mutant is characterized by a 3-fold increase in ATP hydrolysis rate in vitro, the ATP hydrolysis by the Hsp104[E687Q] mutant is rather similar to the one of the wild type[28,30]. Here, we employed HAP variants of Hsp104 due to their ability to translocate the client proteins into the proteolity chamber of the ClpP protease[31]. HAP is an Hsp104 variant bearing three missense mutations (G739I:S740G:K741F) that enable interaction with the protease, ClpP[31,32]. However, HAP variants of Hsp104 were previously shown to behave in the same way as the WT Hsp104[32].

We tested the effect of each mutation on the aggregation extent during glucose starvation. In an attempt to not overestimate the sizes of aggregates, we employed Hsp104[mut]-GFP genomic fusion under the control of the TEF1 promoter (the genomic copy of Hsp104 is deleted).

Hsp104[E285Q] mutant strain showed extensive protein aggregation (Fig. 2a, c), as well as localization of the Hsp104[E285Q]-GFP into the vacuoles (examples are labeled with red arrows in Fig. 2c). We would like to point out that we did not take into account such cases for the aggregate size measurement nor in scoring for aggregate abundance. Already in 2% glucose, Hsp104[E285Q] strain was characterized by the presence of >2 aggregates per cell (>90% of cells) with the median diameter of 625 nm (Fig. 2a, c). This diameter is ~3x larger compared to the median aggregate size in the equivalent WT strain under optimal conditions. When exposed to 0.2% glucose, the fraction of cells with two or more aggregates remained above 90% (Fig. 2a); however, the aggregate size increased further 1.3-fold (median diameter app. 800 nm) ($p = 0.03$, Kolmogorov–Smirnov test) (Fig. 2c). The effect was much more profound in 0.02% glucose, with aggregate abundance showing an interesting trend. The fraction of cells containing 1 aggregate averaged at ~40% (Fig. 2a), while with 2 or more aggregates at ~62% (Fig. 2a). The median diameter of the aggregates increased ~2-fold during 0.02% glucose starvation (~1100 nm) ($p = 0.007$, Kolmogorov–Smirnov test) (Fig. 2c). These results combined point towards the possibility of fusion of smaller aggregates into a larger one, thus giving rise to a lower number of larger compartments. Indeed, the quantification of the average aggregate number per cell confirmed this assumption: while in the WT <1 aggregate appeared per cell at 2% glucose, ~5 aggregates appeared per cell in the Hsp104[E285Q] mutant (Supplementary Fig. 6). After 90 min of starvation in 0.2% and 0.02% glucose, in the WT the number increased up to ~1.8 aggregates, while in the Hsp104[E285Q] mutant the number decreased to ~2 aggregates.

In the Hsp104[E687Q] mutant, already in 2% glucose, ~30% of cells contained 1 aggregate, which is further aggravated during glucose starvation (Fig. 2b). In 0.2% glucose, 56% of cells contained 1 aggregate, while ~25% of cells contained 2 or more aggregates (Fig. 2b). In 0.02% glucose starvation, ~60% of cells carried 1 aggregate, and 33% cells 2 or more aggregates (Fig. 2b). From ~210 nm in optimal conditions (similar as WT), median aggregate diameter increased, 3.1x (median 625 nm) and 3.8x (median 800 nm) in 0.2% and 0.02% glucose, respectively ($p = 0.008$, $p = 0.0001$, respectively, Kolmogorov–Smirnov test) (Fig. 2d).

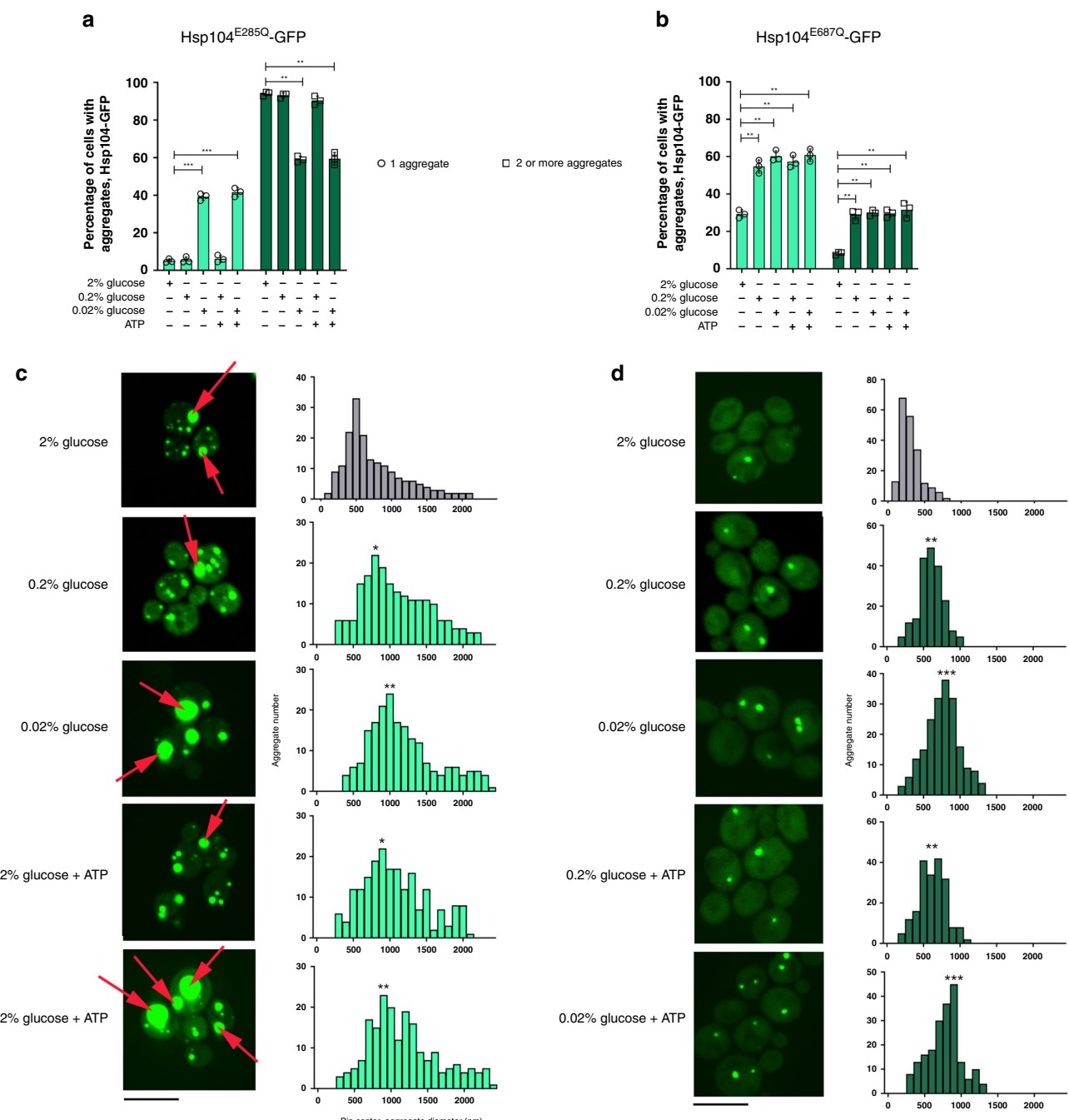

**Fig. 2 ATP hydrolysis by Hsp104 counteract starvation-induced protein aggregation. a** In cells expressing Hsp104[E285Q] mutant, percentage of cells bearing aggregates is strongly increased in optimal conditions of 2% glucose and increases further in 0.2% and 0.02% glucose. Treatment with ATP (in the presence of DMSO) does not lead to aggregate dissolution. **b** In cells expressing Hsp104[E687Q] mutant, percentage of cells bearing aggregates is increased in optimal conditions of 2% glucose and increases further in 0.2% and 0.02% glucose medium. Treatment with ATP (in the presence of DMSO) does not lead to aggregate dissolution. In panels **a** and **b**, $n = 1200$ cells were screened for aggregates starting from three independent exponential yeast cultures for each condition. Data are mean ± SD from at least three independent cultures, each performed in triplicate, and the means of technical replicates are represented as individual data points. ***$p < 0.001$; **$p < 0.01$; *$p < 0.05$ (one-way ANOVA plus Tukey post hoc). **d** Mean aggregate diameter shifts to larger sizes during starvation in 0.2% and 0.02% glucose in Hsp104[E285Q] mutant. The addition of ATP (in the presence of DMSO) did not result in aggregates reverting to the sizes observed in control conditions. Red arrows point to vacuolar localization of the signal and represent structures that were not taken into account during image analysis. Data represent binned values of individual aggregate diameters for $N = 185, 200, 199, 200, 200$ aggregates from three independent exponential yeast cultures for each condition: 2%, 0.2%, and 0.02% glucose, 0.2% glucose + ATP, 0.02% glucose + ATP, respectively. Mean aggregate diameter shifts to larger sizes during starvation in 0.2% and 0.02% glucose in Hsp104[E687Q] mutant. The addition of ATP (in the presence of DMSO) did not result in aggregates reverting to the sizes observed in control conditions. Data represent binned values of individual aggregate diameters for $N = 200$ aggregates from three independent exponential yeast cultures for each condition. In panels **c** and **d**, Kolmogorov–Smirnov test was used to compare the statistical significance of the observed differences between each studied condition and the control (2% glucose) (***$p < 0.001$; **$p < 0.01$; *$p < 0.05$). The images are representative of three biological replicates, each in three technical replicates. The black bar represents 8 μm.

The presented results show that even in optimal conditions, with no additional protein stress, the aggregate size and abundance rely strongly on protein dissolution rather than their deposition into aggregates. The two mutants in the Walker B motif of the Hsp104 NBD1 and NBD2 domains display striking differences in their aggregation propensities during glucose starvation, which likely stem from the allosteric interactions between NBD1 and NBD2[29]. Finally, exposure to external ATP during glucose starvation (0.2% or 0.02%) failed to reduce the abundance of the aggregates or their size in both studied mutants (Fig. 2a–d), suggesting that ATP alone, without the ATPase activity of Hsp104, is not sufficient to dissolve the aggregates.

**Hsp104 ATPase activity determines the aggregate size**. The observed effects of glucose deprivation on the steady-state protein aggregation could have arisen, for example, from ATP-dependent processes involved in aggregate assembly, aggregate dissolution, or both. To distinguish between these scenarios, we monitored the kinetics of Hsp104-containing aggregate assembly during glucose deprivation using spinning disc microscopy and correlated their diameters with declining ATP levels. The strain with Hsp104-GFP genomic fusion under the control of TEF1 promoter was used.

During 60 min of glucose starvation in 0.02% glucose, we observed a 5-fold decline of ATP levels already after 10 min, reaching a minimum of ~1.1 mM (Fig. 3a, c). The median aggregate diameter grew ~4.4x in 60 min (maximal median value of nearly 800 nm), achieving this size already after 10 min from starvation onset (Fig. 3a, c). This trend was accompanied by a gradual increase in the fraction of cells bearing at least one Hsp104-containing deposit (Fig. 3d). Comparable results were obtained for the Hsp104[E687Q] ATPase activity defective mutant (Fig. 3e–g), confirming that aggregate abundance was not affected by the inability of Hsp104 NBD2 domain to hydrolyze ATP. Conversely, placing the WT cells back on 2% glucose after 90 min of glucose deprivation led to the rapid dissolution of protein aggregates (Fig. 3b, c). The average aggregate diameter decreased to initial values of less than 200 nm (below the microscope resolution limit) within 30 min from the reintroduction of glucose. The onset of dissolution corresponded with the restoration of ATP production, culminating in the re-establishment of the initial concentration (~10 mM) also within 30 min from the reintroduction of glucose (Fig. 3b, c). The fraction of cells with at least one Hsp104-containing aggregate also gradually decreased within the same timeframe (Fig. 3d). These experiments demonstrate that aggregate growth mirrors the ATP level decline while their dissolution correlates with the restoration of ATP production. When placed back in 2% glucose after the 90-min starvation period, the Hsp104[E687Q] mutant failed to dissolve the accumulated aggregates, despite the restoration of the ATP level (Fig. 3g, h). This result implies that protein dissolution from aggregates, but not protein deposition into aggregates, is an ATP-dependent process likely determining the steady-state aggregate size. However, a previous study demonstrated that fusion of small Q-bodies into larger ones requires the Hsp104 activity, suggesting that ATP hydrolysis may be important also for the Q-body assembly[5]. Together with our results, this may imply that the activity of either NBD1 or NBD2 is sufficient for the assembly of Q-bodies, even at low ATP concentrations. The quantification of changes in the average aggregate number per cell during the 90 min of starvation revealed differences between the WT strain and the Hsp104[E687Q] mutant (Supplementary Fig. 7). In the WT, the average per cell aggregate number varied during time, stabilizing at ~2 aggregates per cell, likely reflecting aggregate growth and fusion events

(Supplementary Fig. 7). On the other hand, in the Hsp104[E687Q] mutant, average aggregate per cell number displayed a steady increase, reaching an average of 5 aggregates per cell (Supplementary Fig. 7). These results suggested that, while deposition of proteins into aggregates is not a process relying on Hsp104, the aggregate fusion seems to require at least the NBD2 domain activity. Nevertheless, the sizes of individual aggregates in the Hsp104[E687Q] mutant do not differ from those in the WT strain, which may suggest that, even though the individual aggregates do not fuse, they continue to grow individually. This will be a topic of further in-depth research.

To validate further the importance of ATPase activity of Hsp104 in aggregate dissolution, we returned the starved cells to 2% glucose and simultaneously exposed them to 3 mM guanidinium hydrochloride (GdnHCl). GdnHCl is known to inhibit Hsp104 activity at low concentrations[33]. We monitored the aggregate dissolution during the recovery period using spinning disk confocal microscopy. We found that, despite the restoration of the ATP levels, in the presence of GdnHCl, the dissolution did not take place during the 90 min of the recovery period (Supplementary Fig. 8a, b), while in the control assessed in the same experiment dissolution was successful (Supplementary Fig. 8c, d).

Altogether, these results relate the starvation-induced ATP decline with increased protein aggregation and underscore the importance of ATPase activity of Hsp104 in determining the steady-state aggregate size.

**Stress granules and Q-bodies form during glucose starvation**. Proteins tend to aggregate into different types of compartments, which can differ in composition and mechanisms of formation[1,7,15,24,28]. Therefore, we characterized the Hsp104-GFP containing aggregates for the presence of other chaperones and marker proteins that would allow us to identify the compartment types formed under glucose starvation conditions. Here, we used two strains, each with a genomic Hsp104-mCherry fusion together with Ssa1-GFP fusion (strain yYB5841), or Hsp82-GFP fusion (strain yYB11406) (courtesy of Prof. Yves Barral, Supplementary Table 1). It should be noted that for colocalization experiments, all fusions were under the control of native promoters. We opted for the usage of these strains due to their availability, however, we only restrict ourselves to colocalization analysis, and not any aggregate size quantification.

The fraction of cells displaying 1 or more Hsp104-mCherry aggregates corresponded to the abundance of Hsp70-containing (Ssa1-GFP) aggregates: an average of 76% in 0.2 and 85% in 0.02% glucose (Fig. 4a). Moreover, 94% ± 2.3% of individual aggregates with Hsp104-mCherry were also characterized by the presence of Hsp70 (Ssa1-GFP) at 0.2% and 0.02% glucose (Fig. 4a), with the median Pearson's colocalization coefficient of ~0.78 and ~0.82 (Supplementary Fig. 9a). Hsp82-GFP did not display distinct foci under glucose deprivation conditions (Supplementary Fig. 9b).

Further, we analyzed whether the formation of the observed deposits relied on the Hsp42 aggregase, required for Q-body formation. hsp42Δ strain was generated by the deletion of HSP42 gene in the strains bearing the genomic fusion of Hsp104-GFP under the control of native promoter. In the hsp42Δ background, the fraction of cells bearing one or more Hsp104 aggregate decreased dramatically: in 2% glucose, less than 1% of cells displayed any aggregation (Fig. 4b), consistent with literature[34]. Aggregate abundance increased to 17 and 18% in hsp42Δ cells exposed to 0.2% and 0.02% glucose, respectively (Fig. 4b). However, compared to the abundance of all Hsp104-containing aggregates in the WT background during GS, these fractions

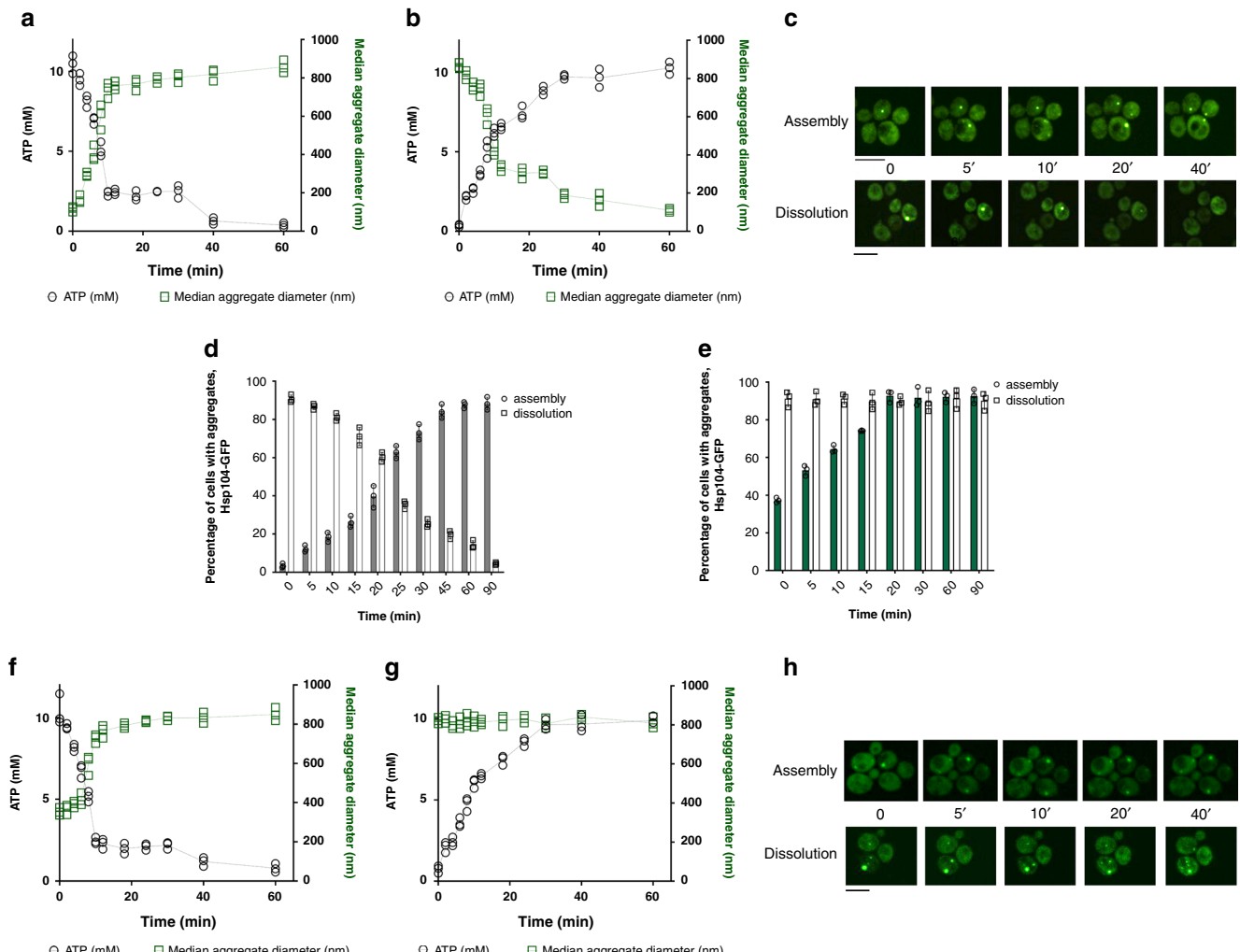

**Fig. 3 Aggregate dissolution by Hsp104 determines protein aggregate steady-state size. a** Cellular ATP declines during starvation (black circles), accompanied by the increase in median aggregate diameters (green squares). **b** During recovery period, the ATP levels are restored (black circles), and median aggregate sizes decrease with similar dynamics (green squares). In **a** and **b**, the individual data points are means of three technical replicates for each of the three biological replicates. The dashed line is connecting means of the three biological replicates at each time point. **c** Representative images of Hsp104-GFP foci at indicated times of assembly and dissolution. This experiment was performed in three technical replicates for each of the three biological replicates. Black bar represents 8 μm. **d** The percentage of cells displaying at least one Hsp104-GFP aggregate increased during starvation in 0.02% glucose and decreased back to control level during recovery in 2% glucose. **e** The percentage of cells bearing at least one Hsp104$^{E687Q}$-GFP aggregate is increasing during starvation in 0.02% glucose, however, did not decrease during recovery in 2% glucose in the Hsp104$^{E687Q}$ strain. In **d** and **e**, n = 1200 cells were screened for aggregates starting from three independent exponential yeast cultures for each condition. Bar height represents mean ± SD from three separate cultures, each performed in triplicate. The mean of three technical replicates for each biological replicate is displayed as single data points. **f** Cellular ATP declines during starvation in 0.02% glucose, followed by the increase in median aggregate diameters in the Hsp104$^{E687Q}$ strain. **g** During recovery period, the ATP levels are restored; however, the median aggregate diameter does not decrease in the Hsp104$^{E687Q}$ strain. In **f** and **g**, the displayed data points are the mean of three technical replicates for each of three biological replicates. The dashed line is connecting means of the three biological replicates at each time point. **h** Representative images of Hsp104$^{E687Q}$-GFP foci at indicated times of assembly and dissolution. This experiment was performed in three technical replicates for each of the three biological replicates. The black bar represents 8 μm.

represented a sharp decline. These results suggest that the fraction of aggregates lost in the absence of Hsp42 is Q-bodies[7].

To support further this observation, we monitored Hsp42-GFP aggregates during glucose deprivation. The strain had an Hsp42-GFP genomic fusion under the control of the native promoter (courtesy of Prof. Ivan Matic). The diameter distributions of Hsp42-tagged aggregates are comparable to those of Hsp104-GFP aggregates (Fig. 4c). Median aggregate diameter at 0.2% and 0.02% glucose was ~700 nm and ~1300 nm, respectively, which represented a significant increase relative to those at 2% glucose (p = 0.0005, p = 0.0008, respectively, Kolmogorov–Smirnov test). We observed an average of 10% of cells bearing at least one

Hsp42-GFP aggregate in 2% glucose (Fig. 4d). This number increased to an average of 45 and 70% in 0.2% and 0.02% glucose, respectively (Fig. 4d). Importantly, this result indicated that the Hsp104-tagged aggregates observed in the WT background are, in fact, the sum of Hsp42-dependent and Hsp42–independent compartments.

To determine the type of the remaining aggregates, we expressed mCherry-tagged Rnq1 (Rnq1-mCherry), a model amyloidogenic protein. Rnq1-mCherry was expressed from a plasmid, transformed into the strain with Hsp104-GFP genomic fusion with the native promoter. Rnq1 was previously thought to associate with IPOD6 exclusively[6] but has recently also been

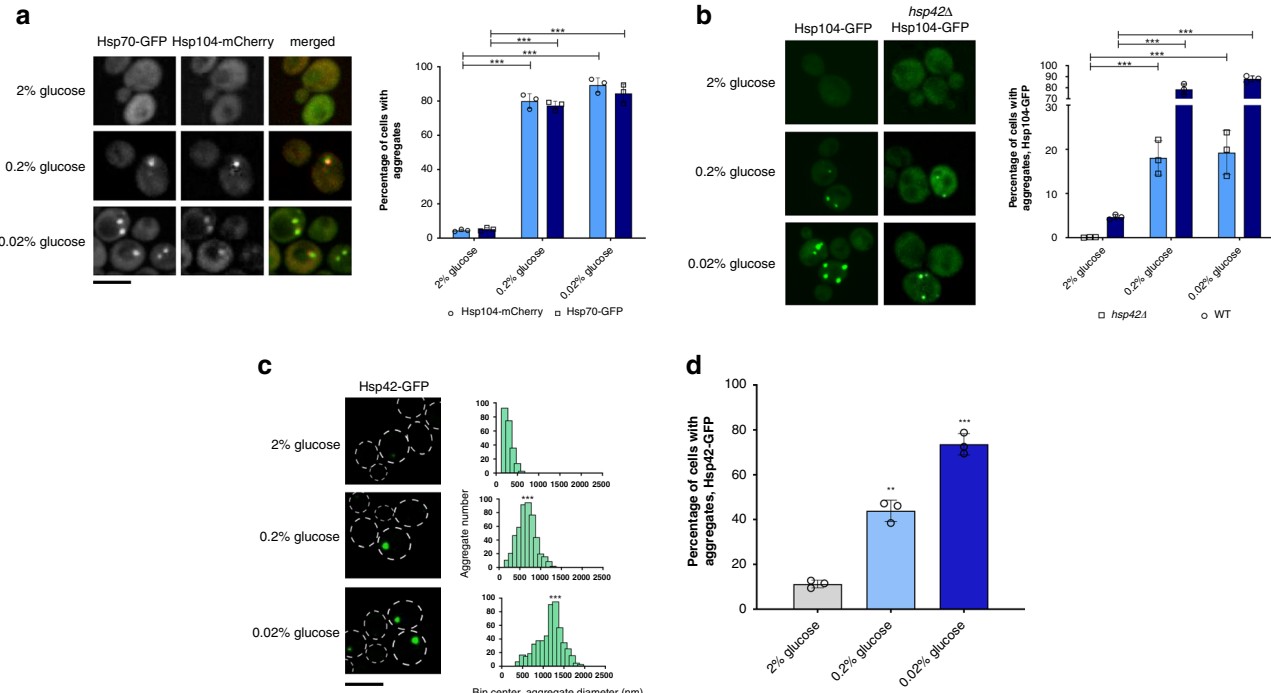

**Fig. 4 Proteins are sequestered into Q-bodies during glucose starvation. a** Hsp70 colocalizes with Hsp104 in aggregates formed during glucose starvation. The percentage of Hsp70-containing (Ssa1-GFP) aggregates is, on average, 76% in 0.2 and 85% in 0.02% glucose and corresponds to the abundance of Hsp104-mCherry aggregates. **b** In the *hsp42Δ* strain, the percentage of cells with Hsp104-GFP aggregates is reduced in both 0.2% and 0.02% glucose. In **a** and **b**, $n = 1100$ cells were screened for aggregates starting from three independent exponential yeast cultures for each condition. Bar height represents mean ± SD from three separate cultures, each performed in triplicate. The mean of three technical replicates for each biological replicate is displayed as single data points. ***$p < 0.001$; **$p < 0.01$; *$p < 0.05$ (one-way ANOVA plus Tukey post hoc). The images are representative of three biological replicates, each in three technical replicates. The black bar represents 8 μm. **c** The number and the median diameter of Hsp42-bearing aggregates increase during starvation. Data represent binned values of individual aggregate diameters for $n = 221$, 494, and 541 aggregates from three independent exponential yeast cultures for each condition: 2%, 0.2% and 0.02% glucose, respectively. Kolmogorov–Smirnov test was used to compare the statistical significance of the observed differences between each studied condition and the control (2% glucose) (***$p < 0.001$; **$p < 0.01$; *$p < 0.05$). The images are representative of three biological replicates, each in three technical replicates. White dotted lines mark cell boundaries. The black bar represents 8 μm. **d** The fraction of cells bearing at least one Hsp42-GFP aggregate increase from an average of 10% in control to 45% in 0.2% glucose and 70% in 0.02% glucose. $N = 550$ cells were screened for Hsp42-containing aggregates starting from three independent exponential yeast cultures for each condition. Bar height represents mean ± SD from three separate cultures, each performed in triplicate. The mean of three technical replicates for each biological replicate is displayed as single data points. ***$p < 0.001$; **$p < 0.01$; *$p < 0.05$ (one-way ANOVA plus Tukey post hoc).

observed as a part of SGs[1]. We found that, on average, 22% of cells had an Rnq1-mCherry aggregate (Fig. 5a), while ~68% and ~90% of cells contained Hsp104-GFP aggregates in 0.2% and 0.02% glucose, respectively. Moreover, median Pearson's colocalization coefficient (Hsp104 vs. Rnq1) was ~0.35 for both starvation regimes (Supplementary Fig. 9c). A similar result was also observed in the *hsp42Δ* background, where almost all (96% ± 3.6%) Rnq1-mCherry aggregates colocalized with Hsp104-GFP (Fig. 5b), with median Pearson's colocalization coefficient of ~0.78 and ~0.8 (Supplementary Fig. 9d). Besides, Rnq1-mCherry aggregates colocalized poorly with the Hsp42-GFP tagged aggregates (Supplementary Fig. 9e), which were present in ~45 and 75% cells in 0.2% and 0.02% glucose, respectively (Fig. 5c).

Based on the so far described results, we speculated that the second type of observed deposits were SGs. To test this, we expressed mCherry-tagged Nrp1 (Nrp1-mCherry), an RNA binding protein established as an SG marker[1,35]. Nrp1-mCherry was expressed from a plasmid, transformed into the strain with Hsp104-GFP genomic fusion with the native promoter. In both 0.2% and 0.02% glucose, ~25% of cells contained Nrp1-mCherry foci (Fig. 5d). Further, 95% ± 2.7% of Nrp1-mCherry foci colocalized with 25% ± 2.5% of Hsp104-tagged aggregates in WT background (Fig. 5d), with median Pearson's colocalization coefficient of ~0.75 (Supplementary Fig. 9f). In *hsp42Δ*

background, Hsp104 and Nrp1 colocalized in 94% ± 4.1% aggregates (Fig. 5e), with median Pearson's colocalization coefficient of ~0.38.

These experiments demonstrated that two distinct types of aggregates were formed during glucose starvation, namely Q-bodies and SGs, similar to the previous observations on protein sequestration into aggregates during heat shock[4].

**Misfolded proteins are secluded during glucose starvation.** Next, we sought to investigate the content of the observed protein aggregates, and first, we tested if they contain misfolded proteins. For the experiments described below we employed the strain with Hsp104-GFP fusion under the control of TEF1 promoter. Cells in starvation conditions were exposed to azetidine-2-carboxylic acid (A2C), a proline analog that induced misfolding upon incorporation into nascent polypeptides, for 10 min. In the presence of A2C, over 80% of cells formed one or more Hsp104-containing aggregate in both 0.2% and 0.02% glucose, higher compared to the control conditions assessed in the same experiment (Supplementary Fig. 10a, b). Approximately half of them rely on Hsp42 for their formation (Q-bodies), i.e., ~45% of cells contained at least one aggregate in *hsp42Δ* background, an increase compared to the control conditions (Supplementary Fig. 10a, b).

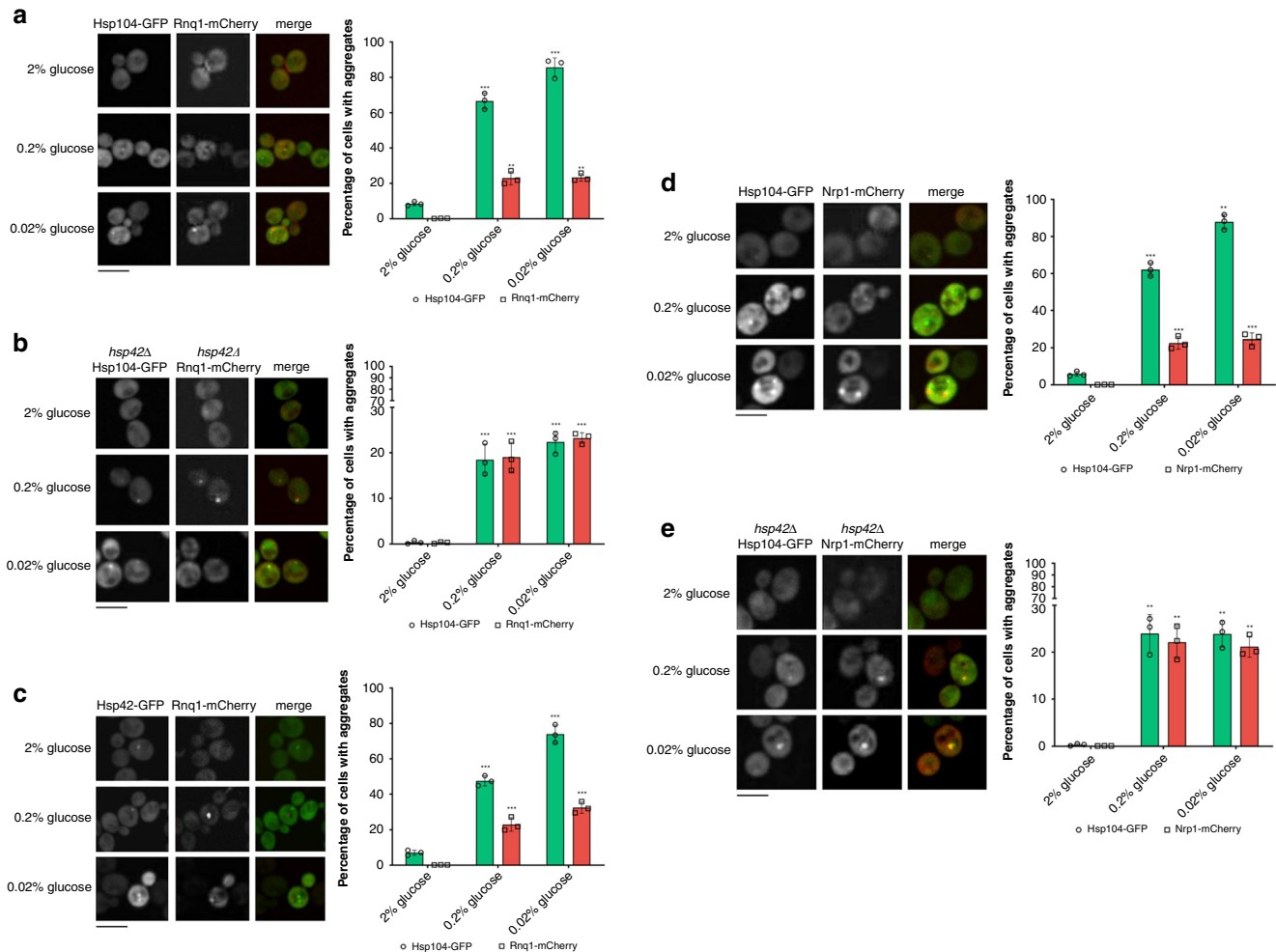

**Fig. 5 Stress granules are formed during glucose starvation. a** On average, 22% of cells had an Rnq1-mCherry aggregate during glucose starvation. **b** In *hsp42Δ* background, the abundance of Hsp104-GFP containing aggregates is reduced to ~20% compared to the WT strain. Rnq1-mCherry aggregates are present in ~25% of cells during glucose starvation. **c** Hsp42-GFP aggregates are present in ~50% and ~75% of cells at 0.2% and 0.02% glucose, respectively. **d** In WT background, ~25% of cells display an Nrp1-mCherry-tagged compartment during glucose starvation. **e** In *hsp42Δ* background, an average of 25% of cells remain with Hsp104-GFP aggregates and ~22% with Nrp1-mCherry puncta during glucose starvation. For all panels, $n = 1000$ cells were screened for aggregates starting from three independent exponential yeast cultures for each condition. Bar height represents mean ± SD from three separate cultures, each performed in triplicate. The mean of three technical replicates for each biological replicate is displayed as single data points. \*\*\*$p < 0.001$; \*\*$p < 0.01$; \*$p < 0.05$ (one-way ANOVA plus Tukey post hoc). The images are representative of three biological replicates, each in three technical replicates. The black bar represents 8 μm.

Further, we aimed to investigate the nature of the remaining aggregates, i.e., if all of them were SGs or an additional type of Hsp104-containing aggregate was formed. In 0.2% and 0.02% glucose, ~25% of cells displayed Nrp1-mCherry foci, identifying them as SGs (Supplementary Fig. 10c). Moreover, in 2% glucose none of the Hsp104-containing aggregates colocalize with Nrp1-mCherry, suggesting that even when exposed to A2C, cells are not prone to SG formation in optimal growth conditions (Supplementary Fig. 10d, e). In 0.2% and 0.02% glucose, colocalization analysis of Hsp104 and Nrp1 in *hsp42Δ* showed median Pearson's colocalization coefficient of 0.37 and 0.4, respectively (Supplementary Fig. 10e). Clearly, in the presence of A2C other types of Hsp104-containing aggregates are formed independently of glucose starvation. However, despite the presence of an additional aggregate type, these results showed that aggregates are in general characterized by larger sizes in the presence of A2C during glucose starvation, suggesting that misfolded proteins are deposited into the compartments formed in these conditions.

The median aggregate size increased during A2C treatment: in 0.2% glucose, the median diameter increased ~3 times (~900 nm), while in 0.02% glucose 5 times (~1500 nm), compared to 2% glucose (300 nm) ($p = 0.009$, $p = 0.0005$, Kolmogorov–Smirnov test) (Supplementary Fig. 10f).

We, next, aimed to address if the formation of the observed aggregates required active translation. Cells were treated with translation inhibitor cycloheximide (CHX) 15 min before the onset of and continued during starvation. Aggregate abundance remained unchanged in the presence of CHX compared to untreated control assessed in the same experiment (Supplementary Fig. 10g, h). In 0.2% and 0.02% glucose, ~24% of cells displayed Nrp1-mCherry foci, identifying them as SGs (Supplementary Fig. 10i). Colocalization analysis of Hsp104 and Nrp1 in the *hsp42Δ* strain showed median Pearson's colocalization coefficient of 0.68 and 0.73 in 0.2% and 0.02% glucose, respectively, suggesting that all remaining aggregates in this background are SGs (Supplementary Fig. 10j, k). The aggregates formed in the presence of CHX, typically grew not >2.5 times in

diameter ($p = 0.021$, $p = 0.033$, Kolmogorov–Smirnov test) (Supplementary Fig. 10l). The decreased diameters of the aggregates in the presence of CHX suggested that the active translation is partially required for the aggregate growth during metabolic stress.

The formation of such mixed compartments by deposition of misfolded proteins into SGs has previously been noted in the conditions of mild heat shock[4]. The authors described that in heat shock, the misfolded proteins in SGs serves as an indicator of protein damage regulating the re-initiation of translation only after the damaged proteins have been eliminated.

To determine if the observed aggregates contain β-amyloid structure, we further performed staining using the Aggresome detection kit (Abcam, see Methods), whereby β-amyloid structures are stained red. We found no red staining in the starved yeast cells colocalizing with Hsp104-GFP, nor independent red-stained foci. As a positive control, we used cells expressing a yeast prion protein, Sup35-GFP, which underwent transient heat shock at 37 °C. Distinct green foci of Sup35-GFP were formed during heat shock and were stained red, suggesting the presence of amyloid structures. Therefore, no evidence exists that would suggest the presence of β-amyloid structure during glucose starvation (Supplementary Fig. 11).

**PKA activation is required for the aggregate dissolution**. Protein kinase A (PKA) reactivation is known to be a significant determinant of cellular recovery from glucose deprivation stress, responsible, among other functions, for reactivation of glycolysis[36]. We, therefore, tested the relevance of its activation in post-starvation aggregate dissolution. For the experiments described below we employed the strain with Hsp104-GFP fusion under the control of TEF1 promoter. After a 90-min starvation in 0.02% glucose, cells were transferred back to 2% glucose supplemented with Rp-cAMPS, a PKA inhibitor. The cells treated with Rp-cAMPS appeared with a faint cytosolic Hsp104-GFP signal, potentially due to strong preferential localization of Hsp104-GFP into the aggregates, which may render the cytosol dark under the applied imaging conditions. Moreover, no aggregate dissolution could be observed, even after 40 min (Fig. 6a–d), while in the control conditions (absence of Rp-cAMPS), aggregate dissolution was completed within 60 min (Fig. 6c). Moreover, no decrease in the fraction of cells bearing Hsp104-containing deposits could be detected (Fig. 6b). Interestingly, the ATP level remained low, comparable to the minimal levels reached during exposure to 0.02% glucose (Fig. 6c, d). Supplementation of the medium with ATP (in the presence of DMSO) led to a decrease in the aggregate abundance during the recovery period, even in the presence of Rp-cAMPS (Supplementary Fig. 12).

PKA-mediated activation of Pfk2 (via phosphorylation) is essential for reactivation of glycolysis upon the reintroduction of glucose in the recovery phase[37]. We, therefore, set out to elucidate if specifically glycolysis reactivation, downstream of PKA reactivation, was required for aggregate dissolution. We first monitored the aggregate abundance in pfk2Δ background, which in optimal conditions, barely displayed any protein aggregation (Supplementary Fig. 13). After the 90-min starvation in 0.2% and 0.02% glucose pfk2Δ strain showed no significant difference in the abundance of the Hsp104-containing aggregates compared to the wild type (Supplementary Fig. 13). However, we observed no aggregate dissolution in the absence of Pfk2 (Fig. 6a–d), consistent with the failure of ATP levels to restore after reintroducing glucose (Fig. 6d). Upon PKA inhibition using Rp-cAMPS in the pfk2Δ background, no aggregate dissolution took place (Fig. 6a–c), implying an epistatic relationship between Pfk2 and PKA.

These results substantiate PKA-mediated reactivation of glycolysis as key to restoring the ATP levels and, consequently, aggregate dissolution during recovery from metabolic stress.

**Protein degradation is not vital for aggregate dissolution**. We have shown that misfolded proteins are a component of both compartments formed during glucose starvation, and one of their possible destinies is proteasomal degradation. We, therefore, aimed to investigate if the inhibition of proteasomal degradation would aggravate the post-starvation aggregate dissolution. For the experiments described below we employed the strain with Hsp104-GFP fusion under the control of TEF1 promoter. We monitored aggregate dissolution in the presence of 25 μM proteasomal inhibitor lactacystin. Both glucose and lactacystin were simultaneously introduced into the yeast culture at the end of the 90-min 0.02% glucose starvation. The results show that aggregate dissolution is unaffected by the presence of lactacystin (Supplementary Fig. 14a). The aggregate abundance decreased to the level of the control conditions within 10 min, consistent with the aggregate dissolution in the absence of lactacystin, assessed in the same experiment (Supplementary Fig. 14b). Median aggregate diameter decrease during the recovery period is also unaffected by the presence of lactacystin and it mirrors the increase in the ATP levels, consistent with the trends observed in the absence of lactacystin, assessed in the same experiment (Supplementary Fig. 14c, d). Representative images are shown in Supplementary Fig. 14e. As a control, to show that the lactacystin indeed inhibited proteasomal degradation, we employed a previously described qPCR-based assay, which postulates that several genes from the heat shock response, namely Sis1, Ydj1, Hsp104 and Ssa1, are upregulated due to proteasomal inhibition by lactacystin[38]. Based on our results (Supplementary Fig. 14f), the transcript levels of all aforementioned genes are increased in the presence of lactacystin, in all studied conditions, compared to the equivalent condition without lactacystin, suggesting successful proteasomal inhibition by lactacystin in the studied conditions.

This result suggests that protein degradation does not play a significant role in the aggregate dissolution, and renders unlikely the conclusion that upon their dissolution, the proteins would be targeted to proteasomal degradation.

**Cytosol pH does not modulate starvation-induced aggregation**. Acidification of the cytosol during glucose deprivation has previously been shown to promote its transition from a fluid- to a solid-like state, thereby also triggering a widespread assembly of cytosolic proteins[39]. We, therefore, set out to investigate the extent to which the cytosolic pH decline that occurs during glucose starvation affects the extent of protein aggregation in experimental conditions studied here.

First, using a pH-sensitive form of GFP (yeast pHluorin, courtesy of Dr. Patricia Kane)[40], we confirmed that the pH of the cytosol indeed acidifies during glucose deprivation (Supplementary Fig. 15a). The fluorescence of GFP decreased as the pH of the cytosol acidifies at 0.2% and 0.02% glucose. We did not measure absolute values of pH, as we did not standardize the probe, and have focused solely on the direction of the change of GFP fluorescence intensity. Using Western blot, we confirmed that the observed decline in fluorescence during glucose starvation is not a consequence of the decrease in pHluorin expression (Supplementary Fig. 15b).

Next, we aimed to test if the decline in pH alone could yield the same effects on protein aggregation as glucose deprivation. To test this, we exposed the cells to protonophore DNP in the presence of 2% glucose, which allowed the pH equilibration between the extracellular environment (the medium) and the

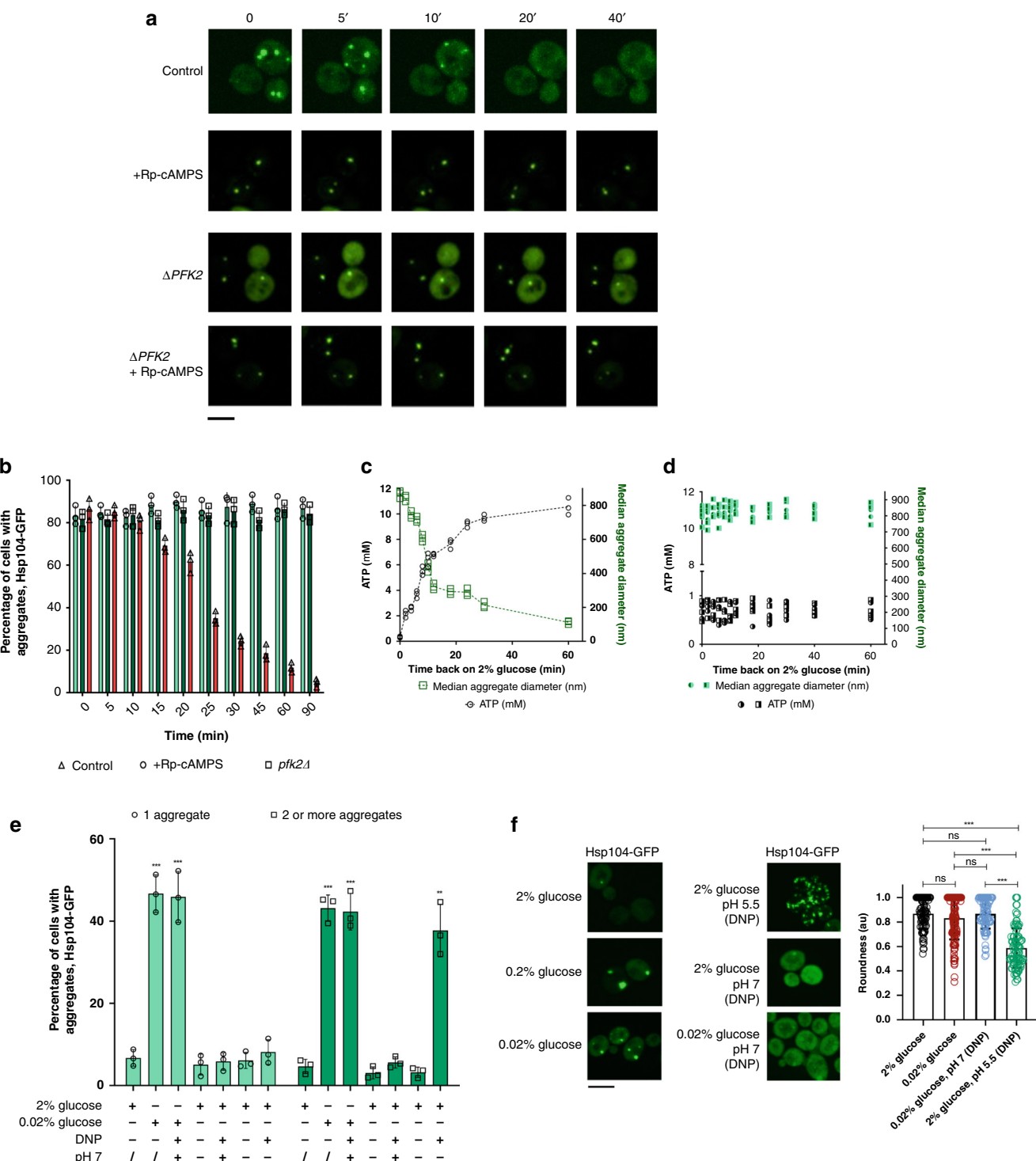

cytosol. We applied two different pH conditions: (i) standard YPD medium with pH 5.5, and (ii) the YPD medium in which pH was set to 7. The latter condition displayed no changes in the aggregate abundance in optimal (2%) glucose (Fig. 6e). It should be noted that, as a protonophore, DNP may present some side-effects resulting from its effect on the proton-motive force across the inner mitochondrial membrane, followed by a reduced ATP synthesis. Below, we considered the possibility of our samples being affected by such side-effects.

Furthermore, using the pH-sensitive GFP, we confirmed that the cytosolic pH of DNP treated cells corresponded to the one of

the medium (Supplementary Fig. 16). We then quantified the abundance of protein aggregates in the conditions listed above and confronted the results to those of glucose starvation (Fig. 6e). In optimal (2%) glucose, in the presence of DNP at pH 5.5, an increase in Hsp104-GFP aggregate abundance could be seen: the fraction of cells with 1 aggregate was ~5% and with 2 or more aggregates ~45%. However, at pH 5.5, we observed the formation of multiple speckles per cell, characterized by decreased roundness, not resembling the spherical shape of the Q-bodies and SGs formed during glucose starvation (Fig. 6f). In 2% glucose at pH 7 (in the presence of DNP), the aggregates are present in ~10% of

**Fig. 6 Protein kinase A post-starvation reactivation enables aggregate dissolution. a** Representative images of Hsp104-GFP aggregates in the presence of Rp-cAMPS (protein kinase A inhibitor), *pfk2Δ* strain and both, at indicated times of the recovery period are displayed, and compared to the control. This experiment was performed in three technical replicates for each of the three biological replicates. The black bar represents 8 μm. **b** Percentage of cells bearing at least one aggregate during the recovery in 2% glucose in the presence of Rp-cAMPS, *pfk2Δ*, and both. **c** In control conditions (2% glucose), during the recovery from starvation in 0.02% glucose, ATP levels increase, and median aggregate diameter is decreased. **d** In the presence of Rp-cAMPS and in *pfk2Δ*, there is no restoration of ATP level during the recovery period, despite the presence of 2% glucose. Median aggregate diameter fails to decrease during the recovery period. **e** Percentage of cells with aggregates increases in 0.02% glucose, both with cytosolic acidification, as well as at pH 7. In **b–e**, *n* = 1100 of cells were screened for aggregates starting from three independent exponential yeast cultures for each condition. Bar height represents mean ± SD from three separate cultures, each performed in triplicate. The mean of three technical replicates for each biological replicate is displayed as single data points. ***$p < 0.001$; **$p < 0.01$; *$p < 0.05$ (one-way ANOVA plus Tukey post hoc). Representative images of cells in different regimes of pH and glucose. The images are representative of three biological replicates, each in three technical replicates. Roundness of the aggregates was quantified using the Shape descriptors plugin for ImageJ. Bar height represents mean ± SD from three separate cultures, each performed in triplicate. Individual data points are given for the roundness of each analyzed aggregate. *N* = 90, 164, 95, and 96 aggregates for the conditions of 2% and 0.02% glucose, 0.02% glucose pH 7, and 2% glucose pH 5.5, respectively. ***$p < 0.001$; **$p < 0.01$; *$p < 0.05$ (one-way ANOVA plus Tukey post hoc). The black bar represents 8 μm.

---

cells, and the aggregates are round (Fig. 6e, f). Moreover, in 0.02% glucose at pH 7 (in the presence of DNP), ~90% of cells contained at least one round aggregate (Fig. 6e, f). Therefore, even though we cannot entirely exclude that some of the observed effects may be a consequence of the DNP side-effects, based on the described results they seem unlikely.

Moreover, to test additionally the contribution of cytosol acidification in glucose starvation-induced protein aggregation, we performed the following experiment: yeast cells were placed into the medium with 0.02% glucose and pH 7 and were exposed to DNP to equilibrate the pH between medium and cytosol. In this way, the cells were exposed to glucose starvation; however, without the decline in pH (Supplementary Fig. 16). In these conditions, the aggregate abundance corresponded to the one observed in 0.02% glucose alone, suggesting that the decrease in pH to 5.5 alone was not sufficient for the glucose starvation-triggered protein aggregation in the studied conditions (Fig. 6e).

**Vacuolar pH does not contribute to aggregate clearance**. We further aimed to determine if the vacuolar pH had any impact on protein aggregate dissolution during the recovery from metabolic stress. As already reported[41], the pH within the vacuole acidifies during glucose deprivation. Here, we measured the vacuolar pH using quinacrine staining[41] during and after glucose starvation, followed by live-cell imaging. First, we confirmed acidification of the vacuole during glucose deprivation that reverses to an initial less acidic pH upon return to 2% glucose. The percentage of cells with acidic pH during exposure to 0.2% and 0.02% glucose increases to 86 and 95%, respectively (Supplementary Fig. 17).

Further, we inhibited vacuolar acidification during glucose starvation using concanamycine A (concA)[41] (Supplementary Fig. 17). We monitored the abundance of protein aggregates in terms of cell fractions with one and with two or more aggregates. Aggregate abundance was unaffected by concA (Supplementary Fig. 18). Aggregate dissolution dynamics was not changed either, compared to conditions of normal vacuolar acidification: the fraction of cells bearing the aggregates decreased as the cells were placed back to 2% glucose in the presence of concA (Supplementary Fig. 18). Based on these results, we concluded that vacuolar acidification does not play a significant role in aggregate clearance under the observed conditions.

**Formation of distinct compartments provides a fitness gain**. The question arises if protein sequestration into compartments under the conditions of metabolic stress has any functional consequences, i.e., does it provide a fitness advantage to the cells. Therefore, we performed competitive fitness experiments according to the scheme presented in Fig. 7a. First, we aimed to

test if the presence of the kanamycin-resistance cassette influences the competitive fitness of the cells. No significant changes in fitness can be observed in the wild type strain due to the presence of the kanamycin-resistance cassette in any of the studied conditions (Fig. 7b).

The *hsp42Δ* cells display no loss of fitness in 2% glucose, while Hsp104[E687Q] mutant cells show a slight loss of fitness, compared to the control (Fig. 7c). The *hsp42Δ* cells that underwent glucose starvation experienced a significantly larger loss of fitness, compared to the same strain in optimal conditions (Fig. 7c). Moreover, the Hsp104[E687Q] mutant suffered the most extensive fitness loss; the strain is able to form, but not to dissolve either Q-bodies or SGs during recovery from metabolic stress (Fig. 7c). These results demonstrate that protein placement into distinct compartments, as well as their efficient dissolution back into the soluble phase (as opposed to degradation and neosynthesis of the same cell components), are required to overcome the metabolic stress efficiently. This result reinforces the argument that not only misfolded proteins are sequestered under these conditions, but perhaps also cellular components likely to be essential for the early stages of the recovery from metabolic stress, like glycolytic enzymes.

## Discussion

There is a growing body of evidence supporting the concept of protein sequestration into various types of cellular aggregates being an adaptive mechanism helping cells to overcome stressful conditions[18,42]. Here, we show that proteins are sequestered into stress granules (SGs) and Q-bodies during metabolic stress caused by declining ATP levels and that their deposition into these compartments and their timely dissolution are both required for the maintenance of cellular fitness. During the recovery from stress, proteins are disaggregated promptly and likely re-sorted between pathways, leading to their refolding, degradation, or continued translation. Our results show that inhibition of protein degradation does not hamper the aggregate dissolution, but based solely on this result we cannot rule out the possibility that some of the dissolved proteins (e.g., misfolded proteins) are degraded shortly after their dissolution.

We identify Hsp104 chaperone ATP hydrolysis activity acting as a determining process for the steady-state size of SGs and Q-bodies, as well as their per-cell number. In average, the cells display two aggregates. A previous study demonstrated that high levels of ATP might preserve soluble protein state, by preventing their aggregation in vitro in a chaperone-independent fashion[43]. In vivo, our results show that ATP, at levels comparable to physiological, promotes the maintenance of soluble protein state by enabling protein retrieval from the aggregates. This reasoning can likely be extended to other ATP-dependent chaperones like

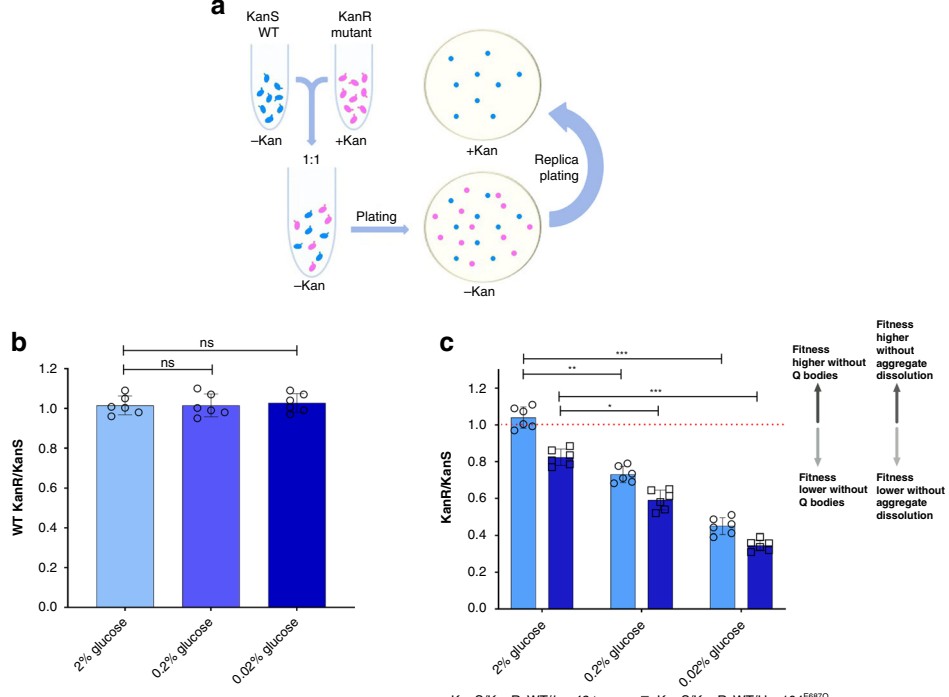

**Fig. 7 Protein compartmentalization during metabolic stress confers fitness advantage. a** Scheme of the competition assay. In the mid-exponential phase the cells were pelleted, and then resuspended in YPD medium with 0.2% and 0.02% glucose for 90 min. Equal representation of the cells was ensured. This was followed by plating the serial dilutions on YPD-agar plates without kanamycin, and dilutions with between 50 and 200 colonies were replica plated on YPD-agar plates with kanamycin (50 μg/mL). Colonies were counted on both types of plates, and the ratio of kanamycin-resistant (mutant) to kanamycin-sensitive (WT) colonies was calculated. Following the same centrifugation procedure, the cells were transferred back into YPD medium with 2% glucose and grown for 24 h counting from the moment of initial dilution. In parallel, a series of the same samples was continuously kept in 2% glucose for control. The same treatment was applied to the control cells to take into account the potential effects of the centrifugation on cell survival. KanS stands for a kanamycin-sensitive phenotype; KanR stands for a kanamycin-resistant phenotype. **b** Competitive fitness of the WT strain is unaffected by the kanamycin-resistant phenotype in optimal conditions, and during the starvation. **c** Competitive fitness of yeast cells is strongly decreased in the $hsp42\Delta$ and Hsp104$^{E687Q}$ mutant in both 0.2% and 0.02% glucose. Bar height represents mean ± SD from six separate cultures, each performed in triplicate. The mean of three technical replicates for six biological replicates is displayed as single data points. ***$p < 0.001$; **$p < 0.01$; *$p < 0.05$ (one-way ANOVA plus Tukey post hoc).

Hsp70 and Hsp90, which can also be expected to contribute to the maintenance of the protein soluble state. Even in optimal conditions (2% glucose, and optimal growth temperature), the inability of individual NBD domains of Hsp104 to hydrolyze ATP correlates with accumulation and growth of protein sequestration sites (Fig. 2). These results are in accord with the previous observation that the lack of ATP influences the protein aggregate behavior in budding yeast5. The results of Kroschwald et al. further support our data, showing that the dissolution of SG relies on the activity of Hsp104 chaperone[1]. Here, we distinguish the consequences of inhibiting Hsp104 domains NBD1 and NBD2 on protein dissolution and steady-state aggregate size. In this context, the deactivation of the ATPase activity of the NBD1 domain has by far more substantial consequences than NBD2. In Hsp104, NBD1 and NBD2 domains each form a hexamer ring that stack on top of each other. ATP binding to NBD1 strongly enhances the ATP turnover by NBD2, suggesting the existence of an allosteric signaling network[29]. In vitro experiments showed that ATP permanently occupies the ATP-binding site of NBD1 in the Hsp104$^{E285Q}$ mutant, resulting in permanent stimulation of ATP hydrolysis by NBD2 and faster ATP hydrolysis[30]. This phenomenon may render this mutant more vulnerable to the decreased ATP levels. In Hsp104$^{E687Q}$ (NBD2 mutant), the affinity for ATP binding to NBD1 is higher[29], which may result in efficient ATP hydrolysis at NBD1 even at low ATP concentrations during starvation and enable at least some protein

dissolution activity – a possibility that does not exist in Hsp104$^{E285Q}$ (NBD1 mutant). In vitro measurements on Hsp104$^{E687Q}$ mutant point not only towards the increased binding affinity for ATP in NBD1, but also that ATP binding to NBD1 governs ATP binding and hydrolysis at NBD2. These results underscore that NBD1 is an allosteric center that controls ATP turnover in both NBD1 and NBD2, which could explain the severe consequences of NBD1 ATPase deactivation[29]. Hsp104 is characterized by a high affinity to bind ADP, which makes it a potent inhibitor of ATP hydrolysis by the chaperone[14,29,44]. Interestingly, inhibition by ADP of the Hsp104$^{E285Q}$ mutant is comparable to WT, while of Hsp104$^{E687Q}$ mutant reduced compared to WT. The in vitro studies are congruent with our observations that the supplementation of cells with external ADP yielded no positive effect on protein disaggregation since the aggregate abundance and size remained at the level triggered by the glucose starvation.

Compartment assembly during glucose starvation has previously been identified as a process aided by the decline in cytosolic[42,45], but not vacuolar pH. In optimal glucose and acidic pH, we observed the formation of multiple compartments per cell containing Hsp104 chaperone (albeit not with the same abundance or shape), in line with the previous observations that acidic pH triggers the phase separation into a solid-like state and leads to macromolecular assembly[24,39]. However, these results should be interpreted with caution since the exposure of cells to DNP

was found to mildly protect the cells from metabolic and oxidative stress by increasing cAMP levels and lowering mitochondrial free radical production[46].

In optimal growth conditions in 2% glucose, the yeast cells can counteract cytosol acidification, as previously reported[47], predominantly by the activity of the proton pump Pma1 in the cell membrane. In addition to Pma1, Pma2 also exists, however it is present in lower amounts and its catalytic function is not entirely clear. Pma1 requires ATP to maintain the cytosolic pH neutral. Therefore, due to the ATP decline during glucose deprivation, it is not able to maintain the neutral pH, rendering the cytosol acidic[47]. Another possible explanation exists, which would account for the cytosol acidification during glucose starvation. V1 subunit of the vacuolar V-ATPase, facing the cytosol, also requires ATP hydrolysis for proton transport into the vacuole. Similarly to Pma1, the reduced activity of the V1 subunit may lead to the acidification of the cytosol[48]. The cytosol acidification observed here might be a result of any of these mechanisms.

In starved cells, upon glucose addition, both aggregate types dissolve readily as soon as ATP levels begin to restore. Our results suggest that PKA-mediated reactivation of glycolysis is required for aggregate dissolution to occur. The relationship between PKA post-starvation reactivation and cell recovery has already been described[49,50]; however, our results extend the knowledge by showing the importance of PKA reactivation for protein aggregate dissolution by activating glycolysis, a significant source of ATP for aggregate dissolution. Interestingly, a previous study suggested glycolysis as the primary source of ATP for aggregate dissolution after heat shock, but whether PKA activation is also relevant remains to be seen[51]. The fact that both types of compartments dissolve upon the ATP level restoration, once again, confirms previous observations that aggregation is not necessarily a terminal event[3,24,45]. During the onset of and the recovery from glucose starvation, the variation in the ATP level is a process that closely interacts with the status of protein folding. Protein sequestration into distinct compartments represents not only a defense mechanism but also an adaptation to ongoing stress that can be easily reversible upon restoration of favorable conditions. A sharp decline in the competitive fitness of cells unable to form compartments during stress supports this conclusion.

During cellular ATP decline, misfolded proteins are sequestered into distinct compartments, presumably due to the low efficiency of usual ATP-consuming proteostatic processes at low ATP (e.g., refolding or degradation). Thereby, cytosolic proteostasis could be alleviated, and the handling of misfolded proteins postponed until the stressful conditions are overcome. Interestingly, the deposition of misfolded cytosolic proteins into SG is not typical, and so far has been observed only during heat shock[24,52]. A complete composition of the observed compartments, as well as the key to misfolded protein sorting between Q-bodies and SG, remains to be investigated.

In conclusion, our results demonstrate that protein sequestration into distinct compartments is essential in overcoming metabolic stress, and represents a part of the short-term cellular adaptation to the starvation conditions with benefits to cellular fitness. Finally, they accentuate the importance of studying metabolic and proteostasis pathways in concert, as opposed to separately, as their crosstalk is of significant importance in understanding aging and age-related diseases.

## Methods

**Strains and growth conditions**. The list of strains used in this study and the media used for their growth is provided in Supplementary Table 1. All strains were grown either in YPD medium (1% yeast extract, 2% peptone, w/v), or Uracil drop-out medium (yeast nitrogen base, yeast synthetic uracil drop-out supplement) in the presence of 2% (w/v) glucose at 30 °C with shaking. All experiments were

performed on cells from mid-exponential phase: cells were grown until OD 0.6–0.8, harvested by 5-min centrifugation at $3000 \times g$, washed, and treated accordingly. The control cells were treated the same way as the cells destined for starvation experiment. For glucose starvation, cells were transferred into 0.2% or 0.02% glucose and incubated with shaking at 30 °C for 90 min. To monitor the effect of ATP, ADP, and GTP on the extent of protein aggregation during glucose starvation, in the last 30 min of the 90-min starvation, ATP, ADP, and GTP were added into the cultures together with 5% DMSO. The final concentration of ATP, ADP, and GTP was 200 mM. ADP and GTP were included in order to test if they can be converted into or serves as a relevant energy source for the process governing the protein aggregation during starvation. At the end of the starvation, the cells were pelleted, washed and prepared for imaging.

**Cloning and generation of genomic fusions of fluorescently tagged Hsp104 disaggregase**. The HAP versions of the wild type Hsp104, as well as the mutants Hsp104$^{E285Q}$ and Hsp104$^{E687Q}$ were amplified from pET24a-Hsp104_HAP (lab collection), pET24a-HAP E285Q and pET24a-HAP-E687Q (courtesy from Dr. Krzysztof Lieberek), using primers as listed in the Supplementary Table 2, and inserted into pTYES2-EGFP using EcoRI and XhoI/SalI. pTYES2 vector was constructed from the pYES2® vector (Thermo Fischer), containing the URA3 gene for selection of transformants by uracil prototrophy in S. cerevisiae. GAL1 and T7 promoter DNA fragments from the pYES2® vector were replaced with TDH3 promoter amplified from pUTDH3myc-EYFP (courtesy of Dr. Antonios Makris, The Institute of Applied Biosciences, Greece) using primers with additional SpeI and EcoRI sites (listed in the Supplementary Table 2). pTYES2 vector was initially designed for the constitutive overexpression of recombinant proteins in S. cerevisiae under the TDH3 promoter, a glucose-dependent constitutive promoter. Then, the enhanced green fluorescent protein (EGFP) gene was amplified from pEGFP-N1 (Clontech) (primers listed in Supplementary Table 2) and inserted into pTYES2 using EcoRI and XhoI.

WT and mutant Hsp104-EGFP were digested from the corresponding pTYES2-EGFP vectors. They were inserted into the pDK-HT vector, containing an integrative module for efficient genome integration into S. cerevisiae histidine loci (courtesy of Dr. Daniel Kaganovich). The integrative module of pDK-HT contains a TEF1 promoter for the constitutive overexpression of recombinant proteins in S. cerevisiae and a TEF1 transcriptional terminator for efficient termination of mRNA. All constructs were verified by sequencing and control restriction digestions.

Hsp42, Nrp1, and Rnq1 were amplified and inserted into pUTDH3myc_mCherry plasmid to obtain a C-terminal fusion of each protein with mCherry, using EcoRI and XhoI restriction sites (for primers see Supplementary Table 2). They were then inserted into the pDK-HT vector, containing an integrative module for efficient genome integration into S. cerevisiae histidine loci (courtesy of Dr. Daniel Kaganovich). Hsp42 and Hsp104 genes were deleted using PvuII and SacI restriction sites, employing homologous recombination-based procedure[53] (for primers see Supplementary Table 2).

**Western blot detection of GFP levels**. Exponentially growing cells were pelleted, washed and incubated in 1 mL lysis buffer (TBS, 60 U zymolyase, protease inhibitor cocktail) for 1 h at 37 °C and spheroplasts were collected after 5 min centrifugation at $1800 \times g$. Pellet of spheroplasts was resuspended in spheroplast lysis buffer (1 mL buffer per 0.5 g of cell pellet; 0.6 M Sorbitol, 10 mM Tris-HCl pH 7.4, 1 mM PMSF). The pellet was vigorously vortexed for 1 min, and left on ice for 30 min, with occasional vortexing. After centrifugation, supernatant was collected, and protein concentration was measured using Bradford reagent (Sigma).

Fifty micrograms of protein for each sample was mixed with Laemmli sample buffer (10% SDS, 20% glycerol, 10 mM 2-mercaptoethanol, 0.05% bromophenol blue), heated to 95 °C for 5 min, and loaded onto 8% sodium dodecyl sulfate polyacrylamide gel electrophoresis gel. The gel was then transferred onto a nitrocellulose membrane at 200 mA for 1 h. The membrane was then blocked using 5% milk in PBS containing 0.1% Tween-20 for 1 h. Anti-GFP antibody (Abcam, ab1218, mouse) was diluted 5000x in blocking buffer, and incubated overnight at 4 °C. Anti-tubulin antibody (Abcam, ab6161, rat) was diluted 3000x in blocking buffer, and also incubated overnight at 4 °C. Both primary antibodies were added simultaneously. Detection was done by horseradish peroxidase-conjugated goat anti-mouse IgG secondary antibody (Abcam, ab97023), and horseradish peroxidase-conjugated rabbit anti-rat igG (Abcam, ab6734), both diluted 20,000x in blocking buffer.

Protein amount in the pHluorin and tubulin bands was quantified by using ImageJ software, and intensity of pHluorin was normalized to tubulin (Tub1) for each sample.

**Cytosolic pH measurement and adjustment**. Cytosolic pH was measured using a pH-sensitive GFP, yeast pHluorin, expressed under the control of a yeast promoter (courtesy of Dr. Patricia Kane). WT yeast was transformed with the plasmid bearing pHluorin and grown on the selection medium lacking uracil (-URA).

The intracellular pH of S. cerevisiae cells was adjusted by incubation in YPD medium of pH 5.5 or 7 in the presence of 2 mM 2,4-dinitrophenol (DNP) as described previously[23,54].

**Quinacrine staining**. Quinacrine (Sigma) staining was performed as previously described[41] (Hughes and Gottschling). Briefly, cells were washed once in YEPD + 100 mM HEPES, pH 7.6 and resuspended in 100 μl of the same buffered media containing 200 μM quinacrine. Cells were incubated for 10 min at 30 °C and then 5 min on ice, followed by pelleting and washing twice with ice cold 100 mM HEPES, pH 7.6 + 2% glucose. Cells were resuspended in 100 mM HEPES, pH 7.6 + 2% glucose for imaging. Quinacrine staining was performed for cells in several conditions: (i) exponentially growing cells from 2% glucose, (ii) cells after 90 min of starvation in 0.2% glucose, (iii) cells after 90 min of starvation in 0.02% glucose, (iv) cells after a 20-min recovery in 2% glucose following starvation in 0.2% glucose, and (v) cells after a 20-min recovery in 2% glucose following starvation in 0.02% glucose. Prior to imaging, cells were kept on ice and all images were obtained within 30 min of staining. Supplementary Fig. 17b presents an example of quica-crine stained (acidified) vacuole, and only such vacuoles were scored as positive for quinacrine staining, i.e., acidified. Concanamycin A (Sigma) was added to cultures at a final concentration of 500 nM to achieve the inhibition of vacuolar acidification.

**Aggresome detection**. In order to characterize the detected Hsp104-positive aggregates for the presence of amyloid structures, we employed the Aggresome detection kit (Abcam) according to the manufacturer's instructions. Briefly, exponentially growing cells were subjected to starvation in 0.02% glucose for 90 min. The negative control cells were kept constatntly in 2% glucose. The positive control yeast strain, bearing the Sup35-GFP genomic fusion, were subjected to a 20-min heat shock at 37 °C in exponential growth stage. This was followed by washing the cells in 10 mM PBS and staining in the Aggresome detection kit in the presence of 8 mg/L DMSO to facilitate the entry of the dye into the yeast cells. Colocalization between Hsp104-GFP, as well as Sup35-GFP and red-stained foci (indicative of amyloid structure), was scored in at least 500 cells from three biological replicates, each in three technical replicates.

**Microscopy: slide preparation**. Microscope slides were prepared as follows: 200 μL of YPD media containing 2% agarose was placed on a preheated microscope slide, and cooled, before applying yeast cells to obtain a monolayer. The glucose concentration in the agarose pad was adapted according to the needs of an experiment. For the quantification of aggregate abundance during starvation, the cells were placed on agarose pads with the same glucose concentration as during starvation. To monitor the kinetics of aggregate assembly, the cells were placed on agarose pads with 0.02% glucose. To monitor the kinetics of aggregate dissolution, the cells were placed on agarose pads with 2% glucose. The cells were previously centrifuged at $3000 \times g$ for 3 min and resuspended in 50 μL YPD. Once dry, the coverslip was placed and sealed.

**Live-cell imaging and image analysis**. The slide was mounted on the Volocity software (version 6.3; Perkin Elmer) driven, temperature-controlled Nikon Ti-E Eclipse inverted/UltraVIEW VoX (Perkin Elmer) spinning disc confocal setup. We also employed the auto-focus system (Perfect Focus, Nikon), and Nano Focusing Piezo Stage (NanoScanZ, Prior Scientific). Images were recorded through 60xCFI PlanApo VC oil objective (NA 1.4) using coherent solid-state 488 nm/50 mW diode laser with DPSS module, and $1000 \times 1000$ pixels 14 bit Hamamatsu (C9100-50) electron-multiplied, charge-coupled device (EMCCD). The exposure time was 150 ms, and 5–10% laser intensity was used. The images were analyzed using Image J software. The number of cells with Hsp104-GFP foci was counted manually. Shape of the aggregates (roundness) was quantified using the Shape descriptors plugin for Image J.

For measuring the abundance of protein aggregates, a total of >1000 cells was scored manually for the presence of 1 or two or more aggregates. The aggregate diameter was measured manually using Image J software. In the case of oval-shaped aggregates, the longest dimension was measured.

**ATP level measurement**. ATP levels were measured using ATP Bioluminescence Assay Kit CLS II (Roche Life Science) according to the enclosed manufacturers' instructions.

Briefly, this method utilizes the luciferase from *Photinus pyralis* (American firefly), which catalyzes the reaction of ATP with D-luciferin, resulting in the emission of green light with an emission maximum at 562 nm.

In all, $2 \times 10^7$ cells were lysed in 100 μL of Dilution Buffer in the presence of zymolyase (0.1 U/μL) and incubated for 1 h at 30 °C. Protein concentration was measured with Bradford reagent (Sigma). The ATP standard curve was prepared by serial dilutions of one ATP standard with a dilution buffer. The amounts of ATP in the samples were calculated according to the standard curve and normalized to total protein concentration.

**A2C and CHX treatments**. In all, 1 mg/mL azetidine-2-carboxylic acid (A2C) was added into exponentially growing cells in 2%, 0.2% and 0.02% glucose and incubated with shaking at 30 °C for 10 min. Cells were then harvested and prepared for imaging. Cycloheximide (100 μg/mL final) was added to exponentially growing cells while still in 2% glucose and incubated for 15 min. Cells were then transferred

to 0.2% or 0.02% glucose and continued to be exposed to CHX for additional 15 min. After that, the cells were prepared for imaging.

**Rp-cAMPS treatment**. Following the exposure to 0.2% or 0.02% glucose, the cells were pelleted, resuspended in 2% glucose, and exposed to 0.5 mM RpcAMPS (Sigma) for 30 min. After that, cells were collected and prepared for imaging.

**RNA extraction**. Cells were harvested from exponential growth stage, washed in RNase free water, and incubated in 100 μL of Y1 buffer (1 M sorbitol, 0.1 M EDTA, pH 7.4) with 50-100U of zymolyase for 1 h at 30 °C. Total RNA was isolated from yeast cells following the procedure of the NucleoSpin RNA kit (Macherey&Nagel) for up to $10^8$ yeast cells. RNA quantification and quality control were done using Nanodrop (PeqLab).

**Quantitative real-time PCR**. Complementary DNA (cDNA) was synthesized from 1000 ng of total RNA using the iScript™ cDNA Synthesis Kit (Biorad). cDNA was diluted 1:100, and each 8 μl reaction contained 4 μl diluted cDNA, 0.2 μl dilutions of each primer (from 25 μM stock), and 3.6 μl iTaq Universal SYBR Green Supermix (BioRad).

All primer pairs were designed to have a melting temperature of 60 °C. The qPCR reaction was run on a QuantFlexStudio 6 (Life Technologies) using 40 cycles, after which the melting curves for each well were determined. qPCR differential expression was estimated from three- and fourfold replicates[55], by first removing Ct values over 1 standard deviation (SD) from the mean Ct for each gene/strain combination. Final fold change values were estimated relative to the UBC6 gene in the control strain replicates.

**Competition assays**. WT and mutant strains were grown in YPD medium in 2% glucose until saturation. The next day, an equal number of cells (confirmed by plating to YPD-agar plates) from WT culture (kanamycin-sensitive) and each mutant (kanamycin-resistant) were mixed in fresh YPD medium, 2% glucose, so that each was diluted ~200x. For control, the kanamycine-resistant wild type strain was competed against the kanamycine-sensitive version to test if the kanamycine resistance affects the fitness of the strains in any of the tested conditions.

In the mid-exponential phase, OD 0.4–0.6, the cells were pelleted by centrifugation at $3000 \times g$ for 5 min, washed twice using YPD medium without glucose, and then resuspended in YPD medium with 0.2% and 0.02% glucose for 90 min. Prior to placing the cells to the starvation conditions, we ensured that the competing strains are equally represented. This was done by plating the serial dilutions on YPD-agar plates without kanamycin, and dilutions with between 50 and 200 colonies were replica plated on YPD-agar plates with kanamycin (50 μg/mL). Colonies were counted on both types of plates, and the ratio of kanamycin-resistant (mutant) to kanamycin-sensitive (WT) colonies was calculated.

Following the same centrifugation procedure, the cells were transferred back into YPD medium with 2% glucose and grown for 24 h counting from the moment of initial dilution. In parallel, a series of the same samples was continuously kept in 2% glucose for control. The same treatment was applied to the control cells to take into account the potential effects of the centrifugation on cell survival.

Serial dilutions were plated on YPD-agar plates without kanamycin, and dilutions with between 50 and 200 colonies were replica plated on YPD-agar plates with kanamycin (50 μg/mL). Colonies were counted on both types of plates, and the ratio of kanamycin-resistant (mutant) to kanamycin-sensitive (WT) colonies was calculated.

**Statistical analysis**. Statistical analysis of data was performed using GraphPad Prism 8 software. All groups were tested for normality of distribution using Shapiro–Wilk test. Since data followed normal distribution, multiple groups were compared using parametric one-way analysis of variance (ANOVA), followed by Tukey's post hoc test, unless otherwise stated. The cumulative frequency distributions of aggregate sizes were compared using Kolmogorov–Smirnov test. For all tests significance level was set at $p < 0.05$.

Person's colocalization coefficient was calculated for all colocalization analyses using Image J. For each experiment, 30 coefficients were calculated, based on images with at least 50 cells. When required, images were cropped to eliminate the parts of the image without cells.

**Reporting summary**. Further information on research design is available in the Nature Research Reporting Summary linked to this article.

## Data availability

Other data are available from the corresponding author upon reasonable request. Source data are provided with this paper.

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

## Acknowledgements

A.K. is supported by the Heisenberg grant from the Deutsche Forschungsgemeinschaft. M.M. was supported by the Mediterranean Institute for Life Sciences, as well as by FEBS and EMBO short-term fellowships; grant 337327 from the European Research Council to N.R.; I.M. is supported by an Emmy Noether Award from the Deutsche For-schungsgemeinschaft. The authors are grateful to Mr. Dirk Schwitters for technical assistance. The authors would like to thank Prof. Yves Barral for sharing the yeast strains; to Dr. Patricia Kane, Dr. Krszysztof Liberek, and Dr. Antonios Makris for sharing plasmids. A.K. is grateful to Prof. Tiago Outeiro, Dr. Diana Lazaro, Prof. Miroslav Radman, and Prof. Yves Barral for valuable discussions. The authors are grateful to anonymous Reviewers for evaluating this work and for their help in making it better.

## Author contributions

The study was conceived and designed by A.K.; data were generated and analyzed by M.M., U.S., P.B.D., N.R., and A.K.; I.M. provided expertize on spinning disk imaging and image analysis; M.M. and A.K. wrote the paper.

## Funding

## Competing interests

The authors declare no competing interests.
