## [Peer Review File · Nature Communications]

Reviewers' comments:

Reviewer #1 (Remarks to the Author):

This paper by Musa et al. investigates the formation of protein aggregates in starved budding yeast cells. The authors report the formation of protein quality control bodies (Q bodies), IPODs and stress granules in starved yeast cells. They further show that aggregate dissolution is dependent on ATP and requires Hsp104. Although some of the reported findings are interesting, I do not think that the paper is suitable for publication in Nature Communications in its current form.

This is for several reasons. First, this study is very descriptive and it provides only little mechanistic insight. Second, many conclusions are not justified by the data. Third, there are several experiments where important controls are missing. Fourth, the paper is written in a way that it is VERY difficult to follow and there are many grammatical errors and typos. And finally, the paper needs a clearer presentation of the data.

Major points:

- 1) Line 25-26: "The observation that amyloidogenic proteins are compartmentalized during glucose deprivation provides new insights into the molecular basis of protein folding diseases, as well as aging." This is a study in yeast. It is completely unclear whether the findings presented here give insight into human diseases. The authors should remove this sentence.
- 2) I do not find information in the manuscript on whether the proteins were tagged endogenously or overexpressed from a plasmid. This is important information that the reader needs to have in order to interpret the results.
- 3) Line 182 and throughout the manuscript: the authors assume that a deletion of Nrp1 prevents SG formation. This is not true. To my knowledge there is no single mutation that leads to the complete absence of SGs in yeast cells. Also, the authors fail to provide data in the manuscript where they compare SG formation in wild type and nrp1- cells. This is a major problem, because many conclusions are based on the assumption that SGs are absent in their experiments.
- 4) Figure 4C: A wildtype control is missing. This experiment only makes sense when nrp1- cells are compared to wildtype cells.
- 5) Line 230-231: I disagree that this implies that translation factors and mRNAs are components of these compartments. There are many alternative explanations.
- 6) Line 235-252: This has to come much earlier in the text. Exogenous addition of ATP is already used in the experiments in Figure 1. This part has to be moved up in the results section.
- 7) Figure S3: The authors have to compare DMSO- and DMSO+ conditions, otherwise the figure is meaningless.
- 8) Figure S2: The cells are very difficult to see. Increase the contrast or put boundaries around the cells.
- 9) Figure S4: a control has to be included in which there was no ATP added.
- 10) Figure S5: a wildtype control has to be added.
- 11) Figure S6: quinacrine experiment: this has to be quantified. I cannot see the difference by looking at a few representative cells.
- 12) Figure 6E and F: this has to be compared directly to the wild type. The authors should also perform an experiment where they inhibit Hsp104 in the recovery phase with GdnHCl. This experimental setup is much better, because the cells have no preformed aggregates as in the case of the Hsp104 mutant.
- 13) Line 334-353: This paragraph seems to be completely disconnected from the rest of the paper. Also, if the authors goal is to show that the aggregates are not degraded, there are much better ways of doing this, for example by comparing wild type cells with autophagy/proteasome mutants.
- 14) Line 378-379. "Here, we show that proteins and translation machineries (RNA) are sequestered into stress granules (SGs) and Q-bodies during metabolic stress caused by declining ATP levels..." There is no data in this manuscript that shows that translation factors and RNAs are sequestered

in these aggregates.

15) Line 387-388: "We identify the Hsp104 chaperone ATP hydrolysis activity as the active process determining the final size of SGs and Q-bodies, as well as their per-cell number." The authors do not know whether this is the only active process. They have to rephrase this.

16) Line 423-427: "The presence of nascent polypeptides in the aggregates may hold the key as to why their dissolution is so important for timely recovery from metabolic stress. It is possible that their deposition into aggregates represents a way of 'freezing' the protein synthesis, acting as reservoirs of translation machineries, as well as metabolic enzymes." I do not understand at all what the authors are trying to say here. Why is the deposition of nascent polypeptides in deposits freezing protein synthesis? This does not make any sense.

17) Line 437-439: "Time-lapse imaging shows that single aggregates do not have a rigid, fixed shape but are undergoing deformation over time, likely aiding their growth and, conversely, dissolution." This is not shown in the manuscript. Either provide evidence or remove.

Minor points:

1) Line 15: "ATP demise". Demise sounds strange to my ears. I suggest using the word "decrease".

2) Line 16 and throughout the manuscript: "seclusion". This is a strange word to use in this context. I suggest using the word "sequestration".

3) Line 160: Were the cells used rnq- or RNQ+? This is information that the reader needs in order to interpret the data.

4) Line 212: Why is this striking?

5) Line 263: Please replace "discern" with "distinguish".

6) Line 384: Please replace "Following" with "In the recovery phase", otherwise this is misleading.

7) Line 395: "cellular" has to be replaced with "cytosolic".

8) It is unclear what the authors mean with the term translation machineries. If they mean translation factors, then they should say so. Machineries are something different.

Reviewer #2 (Remarks to the Author):

In a first set of experiments, the authors use glucose starvation, which results in multiple physiological changes, including ATP depletion, which was confirmed and focused on here. One long known response to glucose starvation is activation of the heat shock factor 1 (Hsf1) (compare for example Hahn and Thiele, JBC 1996 and citations therein), responsible for induction of heat shock proteins, amongst others. Not unexpected then, the authors also observe up-regulation of heat shock protein transcription.

Using Hsp104-GFP as a marker for protein aggregates, the authors monitor aggregates formed during the glucose starvation period. They found that number and size of the Hsp104 bound aggregates strongly increased during prolonged glucose starvation. This is in agreement with the previous finding that starvation causes accumulation of hundreds of cytosolic proteins into distinct foci (Narayanaswamy, Marcotte, PNAS 2009). In a follow-up study, the group of Marcotte suggested that several of these foci represent aggregates that also contain molecular chaperones (O'Connell et al., Mol. BioSyst. 2014). Musa et al do not discuss or mention the O'Connell et al work.

Next, the authors show that the majority of Hsp104-labelled foci contain also Hsp70. Further, they find that the formation of a majority of the foci, but not all, required Hsp42, which argues that they represent Q-bodies. For the remaining foci, the authors found that they represent stress granules. The reasoning described here of how the authors get to the idea that the remaining non-Q-body aggregates could be stress granules (via the amyloidogenic protein Rnq1 that can amorphyously aggregate and localize to stress granules) was somewhat surprising to me because it is known that stress granules can be composed of mixed entities of RNA and misfolded proteins

during for example heat stress (Cherkasov et al., *Curr Biol.* 2013). Therefore the authors could have tested for this directly. Since the Cherkasov work is to my knowledge the first one to describe that protein aggregates and stress granules can form mixed assemblies that are disaggregated by Hsp104 after the inducing stress eases, they should mention/discuss this work. In a next series of experiments, the authors enhanced protein misfolding through A2C and ask if aggregates are distributed to additional sites besides Q-bodies and SG, which is the case. The identity of these additional sites was not determined. Further, they found evidence that the proteins that are deposited during glucose starvation represent proteins that were misfolded before and during the metabolic stress as well as newly synthesized proteins. In additional experiments, the authors find that ATP added to starved cells could dramatically reverse the aggregation almost to a level found without glucose starvation and that net growth of aggregate size correlated with ATP level decline whereas restoration of ATP correlated with net decline in aggregate size. Further, it was observed that a mutant form of Hsp104 that cannot hydrolyze ATP does not affect the size and number of aggregates formed during glucose starvation, but effected the dissolution after ATP restoration. The ATP that was experimentally supplied was then suggested to derive in the living cell from reactivated glycolysis via PKA and Pfk2. Vacuolar acidification did not play a detectable role in aggregate formation and dissolution during glucose starvation. Finally, competitive fitness was reduced in deletion strains where stress granule formation, Q-body formation or protein disaggregation was impaired.

In general, I feel that some parts of the manuscript are not novel enough to provide the reader with a really new conceptual framework. While protein aggregation upon specifically glucose starvation was monitored here, it was already known that general nutrient starvation caused aggregation and deposition into specific cellular sites (which were not identified previously though).

The authors now characterize formation, deposition and dissolution of these aggregates and find that these processes are very similar as for aggregates induced by other kind of stresses, e.g. heat stress. In particular, as for heat stress, misfolded proteins after glucose starvation can be targeted to Q-bodies and Stress granules. As shown for heat-denatured proteins in SGs (Cherkasov et al., 2013, Kroschwald et al., 2015), Hsp104 was also found in this study to be required for disassembly of the aggregates after restoration of glucose levels. Finally it was shown before that cells that cannot sequester aggregates into Q-bodies display reduced competitive fitness (Escusa-Toret, *Nat Cell Biol.* 2013).

This does not mean that the paper does not provide new findings. It is for sure high quality work that should be published, because it works out very nicely several details in this mechanism. Nevertheless, I leave the decision up to the editor whether the study offers substantial novelty and originality to warrant publication in *Nature Communications*.

Should the authors be given the opportunity to submit a revised version, I recommend several changes that should be made:

1) Initially when I read the manuscript for the first time, I thought that a major problem for the quantifications of aggregate number and size in this work would be the tool that was used to monitor aggregation, to measure aggregate sizes etc.. It is Hsp104-GFP, that is differentially expressed in the different conditions that were compared to each others. This becomes immediately visible in the figures where Hsp104-GFP is used. The expression levels and the fluorescence signals are significantly lower in 2 % of glucose as compared to 0,2 % or 0,02 %.

This is not surprising because it is known that Hsp104 is massively induced during starvation and stationary phase. Therefore, aggregates may appear much bigger and brighter after starvation just because there is more Hsp104-GFP present in the cell to monitor aggregates! The same problem holds true for Hsp42-GFP, because Hsp42 is also induced during these conditions.

Then I was puzzled why an Hsp104-mcherry construct was used as well in Fig 1D and E. I am still not sure, but I wonder whether this is a construct under control of a promoter that is NOT induced

during starvation? If this is the case, it has not only be indicated somewhere, but it should really be emphasized to make the reader aware that equal expression levels of the reporter Hsp104 were used to make sure the otherwise different expression levels in the figures with Hsp104-GFP do not change the quantification results.

In contrast, if this Hsp104-mcherry construct was under control of its own endogenous promoter (or a genomic fusion to Hsp104), then one should use a strain with an extra copy of Hsp104-GFP under control of a promoter that is NOT induced during starvation (in a background that is has untagged wild type Hsp104) and test in 1 or 2 proof of principle experiments that with equal expression levels of the reporter, the same results are obtained. If in these controls it was a concern that higher levels of Hsp104-GFP could already change steady state levels of aggregates, one could maybe use the ATP hydrolysis mutant that only binds, but does not dissolve aggregates. This is also not the ideal situation because the mutant might compete for binding to aggregates in such a setting with the untagged wild type Hsp104 in the strain, but the combination of both experiments would at least give a hint whether it is really the aggregate load that is different under these conditions, but not just the expression level of the reporter.

2) The authors use Rnq1 and Sup35 as aggregating substrates, but it stays unclear what aggregate type these proteins form under the conditions of the experiment. Are they aggregating amorphyously or as amyloid aggregates? This should be tested by SDDS-PAGE or ThioflavinT staining. These experiments should also include as positive controls strains where both proteins are known to aggregate as amyloid in the IPOD (in a PIN+ strain for Rnq1 and a PSI+ strain for Sup35).

The question of whether Rnq1 and Sup35 are in an amyloid conformation or amorphyously aggregated is particularly important because the authors discuss that the "glucose depletion induced aggregation mechanism" may have important implications for neurodegenerative diseases.

3) The authors observe in competitive fitness experiments that protein placement into deposition sites as well as dissolution from them after the stress eased, is giving a fitness advantage to the cells. Therefore, it would be highly interesting to find out what the fate of the proteins after their dissolution from the Q-bodies and SGs is. Is the fitness advantage due to preventing damage of misfolded proteins in the cytoplasm by sequestering them away, or because it saves resources when the proteins can be recovered from the aggregates and refolded. Using proteasomal inhibitors and model substrates that are inactive when misfolded and active when natively folded (e.g. firefly luciferase) could give clues about this.

4) In the experiments shown in Fig 6 G and H, the authors observe that inhibition of PKA as well as deletion of downstream Pfk2, aggregate dissolution is impaired. They conclude that due to a defect in restoring glycolysis, no ATP is produced. It would be a good control to add ATP to these cells after starvation to see if the effect can be overcome.

5) In the text describing the experiments, some of the already known features, including:

- that starvation produces aggregates (Narayanaswamy et al., 2009, O'Connell et al., 2014),
- that misfolded proteins can associate with stress granules and require Hsp104 for disassembly from SGs (Cherkasov et al., 2013, Kroschwald et al., 2015)
- and that failure to sequester misfolded proteins into protein quality control compartments reduces competitive fitness (Escusa-Toret et al., 2013).

were sometimes not stated clear enough in my opinion. Although, often they were mentioned in the discussion, I would suggest that the authors report them ideally when they describe the rationale behind their experiment in the results section. Something like "it was known before that,, therefore we wanted to further explore,..." would make it more clear right in the beginning what was known.

Minor points

6) The writing of the paper could be strengthened. For example, in the beginning, the rationale behind the single experiments, but also the experimental details should be explained in a least a bit more detail to demonstrate to the reader that appropriate methods were used etc. The latter is fine to do only in the first one or two experiments, because the methods to monitor aggregation don't change. When I read the paper for the first time without having read the methods in detail, I was not sure whether the Hsp104-GFP fusion was a genomic fusion, what the Hsp104-mcherry fusion was about, that the authors used a spinning disc microscope (which is important to know, because measuring diameters of aggregates from a widefield microscopic image without deconvolution would be difficult due to the spreading of light into different layers.). All these details should be given briefly so that the reader knows that the appropriate methods were used and controls were performed.

7) In most figures in the quantification of the percentage of cells with one or more aggregates, this should be given in % on the Y-axes, because it is also described as % in the main text.

8) In the discussion, it is stated in lines 380-381: "and that their deposition into these compartments and their timely dissolution are both required for cell survival." This is not true! When these processes are inhibited by the deletions in HSP42 or NRP1, or in the ATP hydrolysis mutant of Hsp104, the cells will not die. They are just slightly less fit, as seen in the competitive fitness assays!

9) In the discussion, the authors discuss that due to observations of the aggregates in time-lapse microscopy (which is not shown), they assume that these aggregates may be liquid-like compartments. To my understanding, the work of Kroschwald et al., suggested that SG in yeast (in contrast to mammalian SG) appear more likely in a solid state. Therefore, there is some apparent contradictory point here, which is hard to judge without seeing the data. Therefore, I would like to ask to include these data.

10) The authors observe here that aggregate formation does not require ATP hydrolysis by Hsp104, but dissolution does. Although this does not necessarily need to be a contradiction, Escusa-Toret and co-workers observed that coalescence of smaller aggregates into larger units during Q-body formation requires Hsp104, but degradation of their model substrate (Ubc9-ts) from Q-bodies does not need Hsp104.

It may be helpful to discuss this apparent contradiction or explain why these data may not be exclusive.

Reviewer #3 (Remarks to the Author):

The manuscript by Musa et al includes a detailed characterization of the protein aggregates that accumulate during acute glucose starvation of yeast cells. The authors find that during acute glucose starvation, misfolded proteins are compartmentalized into aggregates with hallmarks of Q-bodies (peripheral aggregates) and stress granules. Dissolution of the aggregates upon glucose refeeding depends on the disaggregating ATPase Hsp104 and the restoration of ATP levels via glycolysis. Finally, mutations that hamper aggregation management or ATP level restoration display impaired fitness when growing cells were subjected to acute glucose starvation followed by outgrowth in high glucose medium.

The data presented in the paper are in line with the present understanding of aggregation formation and dissolution. Similarly, the data showing that Hsp104 is the key disaggregating factor for aggregates accumulated during acute glucose starvation fits with the present understanding

(Powis et al 2013 JCS 126(2)). It is also well established that Hsp104 is exquisitely sensitive to different nucleotide levels, including activation by ATP and inhibition by ADP (Klosowska, Chamera & Lieberk 2016 eLife. 2016; 5:e15159). The fitness experiments are reflecting the results of previously published stress-experiments (Escusa-Toret, Vonk & Frydman 2013. Nat Cell Biol 15(10)). Thus the study represents a detailed characterization of aggregate biology during acute glucose starvation of yeast.

Experimental suggestions

1. The authors document that cellular ATP levels inversely correlates with size of the aggregates and impressively, addition of ATP to the cultures restores intracellular ATP levels and aggregate dissolution. The authors suggest (abstract) that Hsp104 is the key ATP-consuming component that is affected to determine aggregate abundance and size. It would be valuable to document that intracellular nucleotide levels (ATP or ADP) impact on Hsp104 disaggregation activity in vivo as has been suggested by earlier in vitro experiments. Yet the authors present no convincing evidence to substantiate this notion. The experiments with the Hsp104 hydrolysis mutant N728A is not helpful to resolve this matter since this is a dead protein in disaggregation and only shows that Hsp104-dependent disaggregation is involved. These experiments are also confounded by the setup in which the very same mutant Hsp104-GFP is used both to manipulate the system and to monitor aggregate morphology. In fact it could be other ATP-dependent chaperones that are affected energy-stress for example Hsp40, Hsp70 and Hsp110, or even indirect effects (maybe Pma1, see below). Perhaps the authors can strengthen the notion that Hsp104 activity is impaired and rate limiting under energy depletion by quantifying the fraction of cells with aggregates in Hsp104 mutants that have residual function in disaggregation but are impaired for ATP binding. Ideally ATP affinity mutations should be used. Walker B mutants are available and characterized (Klosowska, Chamera & Lieberk 2016 eLife. 2016; 5:e15159).

2. Loss of fermentation upon acute glucose starvation results in that the plasma membrane ATPase Pma1 cannot maintain the plasma membrane proton gradient. Acidification of the cytoplasm subsequently drives protein aggregation (Munder et al 2016 eLife 5:e09347 and references within). This raises the concern that the accumulation of the aggregates upon acute glucose starvation in the present study is the result of cytoplasmic acidification. Thus the effects would be an indirect effect of loss of Pma1 function. The level of acidification depends on the level of ATP depletion and the pH of the conditioned medium used in the experiment. Intracellular pH can be checked with pH-sensitive variants of GFP and the pH of the medium can be controlled with buffers.

3. The authors interpret the experiment in Fig 7A-C to suggest that Hsp42, Hsp104 and Nrp1 all are required for cells survive glucose starvation. However, according to the experimental protocol (p. 18) control cells ('2% glucose', Fig 7C) are not treated with the same extensive centrifugation and washes as the glucose starved cells, raising the concern that some other confounding factor (changed pH, temperature, oxygen or exposure to unconditioned medium et c) is the real stressor. To rule out such indirect effects the experiment should be redone and the control cells should be subjected to the same treatment as the glucose starved cells. Perhaps is this best achieved by simply diluting the growing cells from 2% glucose to the new media and thereby avoid centrifugation all together. Moreover performing these experiments with kanMX knocks without complementation controls is not an ideal setup.

Formatting and data presentation

1. Throughout the manuscript more detailed descriptions are needed regarding how the experiments were performed, particularly regarding the time cells were subjected to glucose starvation in each experiment. Perhaps editing the figure legends and focusing more on description of the experiment and less on interpretation of the data can solve this?

2. The microscopy pictures are generally too dark and do not survive printing.
4. Fig 6 needs extensive graphical editing - labels and microscopy data is simply too small to be assessed.
5. Update figure labels and text to follow standard yeast nomenclature: For example hsp104 Δ (it.) for a genotype and Hsp104 for a protein.
6. The Results section contains both past and present tense when describing experiments and it makes it hard to read.
7. Line 306: The sentence starts 'Remarkably' but it is not remarkable at all that a disaggregation defective Hsp104 does not support dissolution of aggregates.
8. The graphical artwork in Fig 7D is not very helpful. It does not summarize the data in a clear way and it does not function as a model to generate new hypotheses.

Response to Reviewers

Reviewer #1 (Remarks to the Author):

This paper by Musa et al. investigates the formation of protein aggregates in starved budding yeast cells. The authors report the formation of protein quality control bodies (Q bodies), IPODs and stress granules in starved yeast cells. They further show that aggregate dissolution is dependent on ATP and requires Hsp104. Although some of the reported findings are interesting, I do not think that the paper is suitable for publication in Nature Communications in its current form.

We greatly appreciate the Reviewer's careful evaluation of our work and great suggestions that have improved our manuscript. In retrospect, we agree that the original version had several issues – we continued working on them and advancing this project for the past two years. The manuscript is now referred to as Sathyanarayanan, Musa et al.

This is for several reasons. First, this study is very descriptive and it provides only little mechanistic insight.

Having performed many additional experiments that corroborate our conclusions, we are confident to say that our study presents several mechanistic novelties. Thanks to Reviewers' suggestions, our work now explains why glucose starvation triggers protein aggregation: the decline in ATP levels (due to starvation) prevents Hsp104 disaggregase from performing its function and leads to an increase in the abundance and size of Q-bodies and stress granules. We elucidate the relevance of the PKA signaling pathway, and glycolysis as the source of ATP for the appropriate function of Hsp104. Further, we distinguish between the contributions of NBD1 and NBD2 domains.

Second, many conclusions are not justified by the data. Third, there are several experiments where important controls are missing. Forth, the paper is written in a way that it is VERY difficult to follow and there are many grammatical errors and typos.

We realized that the presentation of the results, and sometimes also their interpretation, was indeed weak in the original manuscript. This may have prevented noting that our manuscript presents significant advances. Owing to the effort of Reviewer #1, as well as the others, we resolved the issues mentioned above. We believe the presentation quality has increased, and our conclusions now find support in experiments.

And finally, the paper needs a clearer presentation of the data.

We agree, and hope that our revised manuscript meets this goal.

Major points:

1) Line 25-26: "The observation that amyloidogenic proteins are compartmentalized during glucose deprivation provides new insights into the molecular basis of protein folding diseases, as well as aging." This is a study in yeast. It is completely unclear whether the findings presented here give insight into human diseases. The authors should remove this sentence.

We thank the Reviewer for drawing our attention to this issue. We removed this sentence, as well as the paragraph containing a discussion on the relevance of our results in age-related human conditions.

2) I do not find information in the manuscript on whether the proteins were tagged endogenously or overexpressed from a plasmid. This is important information that the reader needs to have in order to interpret the results.

We thank the Reviewer for raising this point. The additional information on the mode of protein expression is now added throughout revised manuscript. Of note, in the original manuscript, the Hsp104 was tagged endogenously via a genomic fusion with GFP only. We have now added results obtained by using Hsp104-GFP fusion, integrated into the genome, under the control of constitutive TEF1 promoter. We measured protein abundance and size and found that aggregate abundance is in accord with the one obtained with the native promoter, in optimal (2%) glucose, as well as in 0.2% and 0.02% glucose. On the other hand, the aggregates appear ~25% smaller at lower glucose concentrations when Hsp104-GFP with TEF1 promoter is used, compared to Hsp104-GFP with the native promoter. However, the trend of aggregate size increase at low glucose concentrations is preserved, and we believe that result to be crucial (Supplementary Figure 2 in the manuscript).

3) Line 182 and throughout the manuscript: the authors assume that a deletion of Nrp1 prevents SG formation. This is not true. To my knowledge there is no single mutation that leads to the complete absence of SGs in yeast cells. Also, the authors fail to provide data in the manuscript where they compare SG formation in wild type and *nrp1*- cells. This is a major problem, because many conclusions are based on the assumption that SGs are absent in their experiments.

Figure R1. In the absence of Nrp1 no Pab1 foci are detected, but Q-bodies still form with their typical abundance at 0.2% and 0.02% glucose. Hsp104-GFP forms foci in app. 45% and 70% of cells in 0.2% and 0.02% glucose, respectively. More than 1000 cells were screened for aggregates starting from three independent exponential yeast cultures for each condition. Data are mean \pm SD from at least 3 independent cultures, each performed in triplicate. *** $p < 0.001$; ** $p < 0.01$; * $p < 0.05$ (ANOVA plus post hoc).

In the yeast wild-type strain used in this study (S288C), the deletion of Nrp1 leads to the complete absence of SGs. That said, it was a mistake to generalize the conclusion to other wild type backgrounds. In the original version of the manuscript, in the *nrp1*-null strain, no foci of Pab1-mCherry were possible to observe even under stress (Figure R1), and we believed that would have been evidence supporting our conclusion. However, the Reviewer is correct: such a conclusion would require a more elaborate analysis and discussion. We have now removed the results regarding the *nrp1*-null cells and supported the previously existing data with additional controls to reinforce our conclusion. We modified Results and Discussion accordingly.

4) *Figure 4C: A wildtype control is missing. This experiment only makes sense when nrp1- cells are compared to wildtype cells.*

We agree. Data from Figure 4C are not a part of this manuscript anymore: we will address this question in a new study.

5) *Line 230-231: I disagree that this implies that translation factors and mRNAs are components of these compartments. There are many alternative explanations.*

We agree and have removed this speculation from the manuscript.

6) *Line 235-252: This has to come much earlier in the text. Exogenous addition of ATP is already used in the experiments in Figure 1. This part has to be moved up in the results section.*

We thank the reviewer for this valuable suggestion. We have entirely reorganized the revised manuscript: the mentioned part now appears much earlier in the manuscript.

7) *Figure S3: The authors have to compare DMSO- and DMSO+ conditions, otherwise the figure is meaningless.*

We agree with the Reviewer – thank you. We have now modified this graph to display both DMSO- and DMSO+ conditions (Figure R2). In the manuscript, this data is now presented in Supplementary Figure 3.

Figure R2. The addition of DMSO does not influence protein Hsp104-GFP tagged aggregate formation. **(A)** Representative images of Hsp104-GFP containing aggregates in the presence of 5% DMSO. The black bar represents 8 μ m. **(B)** Percentage of cells with one, two, or more aggregates is not changed between untreated control and after addition of DMSO, both in control as well as in 0.2% and 0.02% glucose. More than 500 cells were screened for aggregates starting from three independent exponential yeast cultures for each condition. Data are mean \pm SD from at least 3 independent cultures, each performed in triplicate. *** p<0.001; ** p<0.01; * p<0.05 (ANOVA plus post hoc).

8) *Figure S2: The cells are very difficult to see. Increase the contrast or put boundaries around the cells.*

We have now modified this figure and other figures in the revised manuscript as well. We have either improved the quality of presented images or added boundaries around the cells.

9) Figure S4: a control has to be included in which there was no ATP added.

We agree and have now included an additional panel with the results of conditions where DMSO was added, and ATP was not (Figure R3). In the revised manuscript, these results are presented in Supplementary Figure 4.

Figure R3. Exposure of cells to external ATP during A) 0.2% glucose, and B) 0.02% glucose starvation in the presence of DMSO, the cellular ATP levels are comparable with WT levels. Data are mean \pm SD from at least 3 independent cultures, each performed in triplicate. *** $p < 0.001$; ** $p < 0.01$; * $p < 0.05$ (ANOVA plus post hoc).

10) Figure S5: a wildtype control has to be added.

We agree and have now added the results of the wild-type control in the same figure. In the revised manuscript, this data is included in the Supplementary Figure 10.

Figure R4. Aggregate abundance in the A) WT, and B) *pfk2Δ* background is not significantly different. More than 500 cells were screened for aggregates starting from three independent exponential yeast cultures for each condition. Data are mean \pm SD from at least 3 independent cultures, each performed in triplicate. *** $p < 0.001$; ** $p < 0.01$; * $p < 0.05$ (ANOVA plus post hoc).

11) Figure S6: quinacrine experiment: this has to be quantified. I cannot see the difference by looking at a few representative cells.

We agree with the Reviewer, which is why we quantified it in the first place. We designed the presentation of the results according to Hughes and Gottschling (*Nature* **492**, 261–265 (2012).): the representative images indeed only illustrated different tested conditions, while the numbers underneath the images represented the percentage of cells with a fluorescent vacuole. Even though we explained this in the corresponding figure caption, the comment of the Reviewer helped us to understand that this figure needed re-design. We have, therefore, included a graph showing the quantification of this data as Supplementary Figure 14.

12) Figure 6E and F: this has to be compared directly to the wild type. The authors should also perform an experiment where they inhibit Hsp104 in the recovery phase with GdnHCl. This experimental setup is much better, because the cells have no preformed aggregates as in the case of the Hsp104 mutant.

Thank you for this excellent suggestion; we have performed the proposed experiment and included the results in the revised manuscript. The following protocol was used: yeast cells were subjected to 90-minute starvation in 0.02% glucose. At the end of starvation, the cells were pelleted and resuspended in YPD medium with 2% glucose and 3 mM (final concentration) GdnHCl, followed by monitoring of dissolution of compartments formed during starvation. The results showed that in the presence of GdnHCl, no dissolution of the preformed compartments took place. We have included these results in the revised manuscript (presented in Supplementary Figure 5). We believe that this result reinforces our conclusion that the ATP hydrolysis activity of Hsp104 is the ATP-consuming process responsible for aggregate growth at low ATP and aggregate dissolution at normal ATP levels.

We have also performed other experiments with two additional mutants of Hsp104. We have used two Walker B mutants, generously provided by Dr. Krzysztof Lieberk: Hsp104^{E285Q} in the NBD1 domain, and Hsp104^{E687Q} in the NBD2 domain. These mutants are characterized by abolished ATP hydrolysis in the affected NBD, while maintaining ATP binding. The presented results show that even in optimal conditions, with no additional protein stress, the aggregate size and abundance relies strongly on protein dissolution rather than their deposition into aggregates.

The two mutants in the Walker B motif of the Hsp104 NBD1 and NBD2 domains display sharp differences in their aggregation propensities during GS, with the consequences of NBD1 inactivation being by far more substantial. These differences likely stem from the allosteric interactions between NBD1 and NBD2, whereby NBD1 was recognized as an allosteric center of Hsp104 (Franzmann et al. *J. Biol. Chem.* **286**, 17992–18001 (2011)). Finally, exposure to external ATP during glucose starvation (0.2% or 0.02%) failed to reduce the abundance of the aggregates or their size in both studied mutants (Figure 2A-D), suggesting that ATP alone, without the ATPase activity of Hsp104, is not sufficient to dissolve the aggregates.

Altogether, these results support our conclusion that the ATP hydrolysis by Hsp104 is an ATP consuming event, prominent during the recovery phase.

13) Line 334-353: This paragraph seems to be completely disconnected from the rest of the paper. Also, if the authors goal is to show that the aggregates are not degraded, there are much better ways of doing this, for example by comparing wild type cells with autophagy/proteasome mutants.

We appreciate this suggestion. The experiments described in this paragraph aimed to test the role of vacuolar acidity in the clearance of the compartments forming during glucose starvation (GS) since vacuolar acidity is one of the phenotypes usually discussed in the context of aggregate removal from the cell. We edited this paragraph, and agree that autophagy mutants would have been perhaps a more elegant way to test this. However, since there is no indication of autophagy acting in the clearance of aggregates formed in GS, we did not think it was necessary. We have, however, performed an experiment whereby the aggregate dissolution during the recovery phase is monitored in the presence of proteasomal inhibitor lactacystine. We find no influence of proteasomal inhibition on aggregate dissolution during the recovery phase.

14) Line 378-379. “Here, we show that proteins and translation machineries (RNA) are secluded into stress granules (SGs) and Q-bodies during metabolic stress caused by declining ATP levels...” There is no data in this manuscript that shows that translation factors and RNAs are sequestered in these aggregates.

We agree and have removed this speculation from the manuscript.

15) Line 387-388: “We identify the Hsp104 chaperone ATP hydrolysis activity as the active process determining the final size of SGs and Q-bodies, as well as their per-cell number.” The authors do not know whether this is the only active process. They have to rephrase this.

We agree – this indeed may not be the only active process determining the final size of SGs and Q-bodies. We did not wish to state that either: this is indeed an example of poor wording on our side. This sentence is now rewritten to avoid confusion.

16) Line 423-427: “The presence of nascent polypeptides in the aggregates may hold the key as to why their dissolution is so important for timely recovery from metabolic stress. It is possible that their deposition into aggregates represents a way of ‘freezing’ the protein synthesis, acting as reservoirs of translation machineries, as well as metabolic enzymes.” I do not understand at all what the authors are trying to say here. Why is the deposition of nascent polypeptides in deposits freezing protein synthesis? This does not make any sense.

We removed this speculation from the manuscript.

17) Line 437-439: “Time-lapse imaging shows that single aggregates do not have a rigid, fixed shape but are undergoing deformation over time, likely aiding their growth and, conversely, dissolution.” This is not shown in the manuscript. Either provide evidence or remove.

We are thankful that this comment was raised: we have now removed this sentence from the manuscript. The phenomenon mentioned in the quoted sentence above will be described in a separate publication.

Minor points:

1) Line 15: “ATP demise”. Demise sounds strange to my ears. I suggest using the word “decrease”.

Thank you for this suggestion.

2) Line 16 and throughout the manuscript: “seclusion”. This is a strange word to use in this context. I suggest using the word “sequestration”.

Thank you!

3) Line 160: Were the cells used r_{nq}- or RNQ+? This is information that the reader needs in order to interpret the data.

We used r_{nq}- cells.

4) Line 212: Why is this striking?

We got carried away by the enthusiasm for our results. We removed this word.

5) Line 263: Please replace “discern” with “distinguish”.

Done.

6) Line 384: Please replace “Following” with “In the recovery phase”, otherwise this is misleading.

Done – thank you!

7) Line 395: “cellular” has to be replaced with “cytosolic”.

Done.

8) It is unclear what the authors mean with the term translation machineries. If they mean translation factors, then they should say so. Machineries are something different.

We have removed this confusing phrase.

Reviewer #2 (Remarks to the Author):

In a first set of experiments, the authors use glucose starvation, which results in multiple physiological changes, including ATP depletion, which was confirmed and focused on here. One long known response to glucose starvation is activation of the heat shock factor 1 (Hsf1) (compare for example Hahn and Thiele, JBC 1996 and citations therein), responsible for induction of heat shock proteins, amongst others. Not unexpected then, the authors also observe up-regulation of heat shock protein transcription. Using Hsp104-GFP as a marker for protein aggregates, the authors monitor aggregates formed during the glucose starvation period. They found that number and size of the Hsp104 bound aggregates strongly increased during prolonged glucose starvation. This is in agreement with the previous finding that starvation causes accumulation of hundreds of cytosolic proteins into distinct foci (Narayanaswamy, Marcotte, PNAS 2009). In a follow-up study, the group of Marcotte suggested that several of these foci represent aggregates that also contain molecular chaperones (O’Connell et al., Mol. BioSyst. 2014). Musa et al do not discuss or mention the O’Connell et al work.

We are grateful to the Reviewer for raising this issue. While we are familiar with the work of the Marcotte group, we found that Narayanaswamy et al. (cited in our manuscript several times) is of high importance to our results. Even though we admire the work presented in O’Connell et al. (*Mol. Biosyst.* **10**, 851 (2014)), we did not find sufficient parallels with our work to be drawn in our manuscript. We have, however, mentioned and discussed O’Connell et al. in the revised manuscript.

Next, the authors show that the majority of Hsp104-labelled foci contain also Hsp70. Further,

they find that the formation of a majority of the foci, but not all, required Hsp42, which argues that they represent Q-bodies. For the remaining foci, the authors found that they represent stress granules. The reasoning described here of how the authors get to the idea that the remaining non-Q-body aggregates could be stress granules (via the amyloidogenic protein Rnq1 that can amorphously aggregate and localize to stress granules) was somewhat surprising to me because it is known that stress granules can be composed of mixed entities of RNA and misfolded proteins during for example heat stress (Cherkasov et al., Curr Biol. 2013). Therefore the authors could have tested for this directly. Since the Cherkasov work is to my knowledge the first one to describe that protein aggregates and stress granules can form mixed assemblies that are disaggregated by Hsp104 after the inducing stress eases, they should mention/discuss this work.

Regarding the issue with Rnq1: the path we took to identify the non-Q-body compartments was not meant to ignore the work of Cherkasov et al. Of note, we have done the Rnq1 experiment much before we found out that the misfolded proteins are also present in these compartments. Once we had the result that some of the compartments were Rnq1-positive and knowing that they readily dissolve, we tested if they are SGs. We thought that it is worth reporting that these compartments are positive for Rnq1.

We are grateful to the Reviewer for reminding us to cite Cherkasov et al., which we did not omit on purpose in the previous version of the manuscript.

In a next series of experiments, the authors enhanced protein misfolding through A2C and ask if aggregates are distributed to additional sites besides Q-bodies and SG, which is the case. The identity of these additional sites was not determined. Further, they found evidence that the proteins that are deposited during glucose starvation represent proteins that were misfolded before and during the metabolic stress as well as newly synthesized proteins.

In additional experiments, the authors find that ATP added to starved cells could dramatically reverse the aggregation almost to a level found without glucose starvation and that net growth of aggregate size correlated with ATP level decline whereas restoration of ATP correlated with net decline in aggregate size.

Further, it was observed that a mutant form of Hsp104 that cannot hydrolyze ATP does not affect the size and number of aggregates formed during glucose starvation, but effected the dissolution after ATP restoration. The ATP that was experimentally supplied was then suggested to derive in the living cell from reactivated glycolysis via PKA and Pfk2. Vacuolar acidification did not play a detectable role in aggregate formation and dissolution during glucose starvation. Finally, competitive fitness was reduced in deletion strains where stress granule formation, Q-body formation or protein disaggregation was impaired.

We appreciate this summary of our work.

In general, I feel that some parts of the manuscript are not novel enough to provide the reader with a really new conceptual framework. While protein aggregation upon specifically glucose starvation was monitored here, it was already known that general nutrient starvation caused aggregation and deposition into specific cellular sites (which were not identified previously though).

It is correct that our work is the first one to systematically identify the type and some of the content of two sequestration sites formed during starvation. Besides, our work explains why glucose starvation triggers protein aggregation: the decline in ATP levels, characteristic of starvation,

prevents Hsp104 disaggregase from performing its function, leading to an increase in the abundance and size of Q-bodies and SGs. We elucidate the importance of PKA and glycolysis as the source of ATP for the appropriate function of Hsp104. These results underscore the importance of proteostasis-metabolism crosstalk, which is a fresh perspective in this line of research, and we believe that they bring mechanistic insights into previously reported observational studies.

The authors now characterize formation, deposition and dissolution of these aggregates and find that these processes are very similar as for aggregates induced by other kind of stresses, e.g. heat stress. In particular, as for heat stress, misfolded proteins after glucose starvation can be targeted to Q-bodies and Stress granules. As shown for heat-denatured proteins in SGs (Cherkasov et al., 2013, Kroschwald et al., 2015), Hsp104 was also found in this study to be required for disassembly of the aggregates after restoration of glucose levels. Finally it was shown before that cells that cannot sequester aggregates into Q-bodies display reduced competitive fitness (Escusa-Toret, Nat Cell Biol, 2013).

This does not mean that the paper does not provide new findings. It is for sure high quality work that should be published, because it works out very nicely several details in this mechanism. Nevertheless, I leave the decision up to the editor whether the study offers substantial novelty and originality to warrant publication in Nature Communications.

We appreciate this positive outlook.

Should the authors be given the opportunity to submit a revised version, I recommend several changes that should be made:

1) Initially when I read the manuscript for the first time, I thought that a major problem for the quantifications of aggregate number and size in this work would be the tool that was used to monitor aggregation, to measure aggregate sizes etc.. It is Hsp104-GFP, that is differentially expressed in the different conditions that were compared to each others. This becomes immediately visible in the figures where Hsp104-GFP is used. The expression levels and the fluorescence signals are significantly lower in 2 % of glucose as compared to 0,2 % or 0,02 %. This is not surprising because it is known that Hsp104 is massively induced during starvation and stationary phase. Therefore, aggregates may appear much bigger and brighter after starvation just because there is more Hsp104-GFP present in the cell to monitor aggregates! The same problem holds true for Hsp42-GFP, because Hsp42 is also induced during these conditions.

Then I was puzzled why an Hsp104-mcherry construct was used as well in Fig 1D and E. I am still not sure, but I wonder whether this is a construct under control of a promoter that is NOT induced during starvation? If this is the case, it has not only be indicated somewhere, but it should really be emphasized to make the reader aware that equal expression levels of the reporter Hsp104 were used to make sure the otherwise different expression levels in the figures with Hsp104-GFP do not change the quantification results.

In contrast, if this Hsp104-mcherry construct was under control of its own endogenous promoter (or a genomic fusion to Hsp104), then one should use a strain with an extra copy of Hsp104-GFP under control of a promoter that is NOT induced during starvation (in a background that is has untagged wild type Hsp104) and test in 1 or 2 proof of principle experiments that with equal

expression levels of the reporter, the same results are obtained. If in these controls it was a concern that higher levels of Hsp104-GFP could already change steady state levels of aggregates, one could maybe use the ATP hydrolysis mutant that only binds, but does not dissolve aggregates. This is also not the ideal situation because the mutant might compete for binding to aggregates in such a setting with the untagged wild type Hsp104 in the strain, but the combination of both experiments would at least give a hint whether it is really the aggregate load that is different under these conditions, but not just the expression level of the reporter.

We appreciate the Reviewer raising this issue, which we will now try to clarify. In the initial version of the manuscript, we used a genomic fusion of Hsp104-GFP under the control of the endogenous promoter. Indeed, this was responding to the declining levels of glucose accompanied by the increasing Hsp104-GFP expression.

The Hsp104-mCherry construct was used to demonstrate that the glucose starvation-induced aggregation shows consistent results regardless of the fluorescent reporter protein used. We have now removed those results for clarity.

On the other hand, in the new version of the manuscript, we present the results obtained using Hsp104-GFP fusion, integrated into the genome, and under control of the TEF1 promoter (details are described in the Methods and Results sections). TEF1 is a constitutive promoter for expression in budding yeast, which is regulated by glucose levels. Nevertheless, while the levels of glucose are declining in the concentrations used in this study, the expression level is rather stable [Peng, B., Williams, T. C., Henry, M., Nielsen, L. K. & Vickers, C. E. *Microb. Cell Fact.* **14**, 91 (2015)]. We have repeated the entire experiment measuring aggregate abundance and size using the TEF1-regulated Hsp140-GFP and found that aggregate abundance is in accord with the one obtained using the native promoter (Figure R5). In optimal (2%) glucose, the aggregate abundance corresponds to the one with the native promoter, and the same stands for 0.2% and 0.02% glucose. The situation is different for the estimated aggregate size: in general, the aggregates appear ~25% smaller at lower glucose concentrations when Hsp104-GFP with TEF1 promoter is used, compared to Hsp104-GFP with the native promoter. However, the trend of aggregate size increase at low glucose concentrations is preserved, and we believe that to be crucial (Supplementary Figure 2 in the manuscript).

Figure R5. Percentage of cells bearing aggregates increases almost five-fold during glucose starvation observed using Hsp104-GFP under the control of TEF1 promoter. This is observed in (A) 0.2% (blue) and (B) 0.02% (red) glucose medium. Treatment with ATP, but not GTP or ADP, causes the percentage of cells bearing aggregates to return to levels comparable to control. Protein aggregation propensity is expressed as the percentage of cells with at least one aggregate. More than 500 cells were screened for aggregates starting from three independent exponential yeast cultures for each condition. Data are mean \pm SD from at least 3 independent cultures, each performed in triplicate. *** $p < 0.001$; ** $p < 0.01$; * $p < 0.05$ (ANOVA plus post hoc). (C) Mean aggregate diameter shifts to larger sizes during starvation in 0.2% and 0.02% glucose. Addition of ATP results in aggregates reverting to the sizes observed in control conditions. Data represent binned values of individual aggregate diameters for more than 200 cells from three independent exponential yeast cultures for each condition. Black bar represents 8 μm .

2) The authors use Rnq1 and Sup35 as aggregating substrates, but it stays unclear what aggregate type these proteins form under the conditions of the experiment. Are they aggregating amorphyously or as amyloid aggregates? This should be tested by SDDS-PAGE or ThioflavinT staining. These experiments should also include as positive controls strains where both proteins are known to aggregate as amyloid in the IPOD (in a PIN+ strain for Rnq1 and a PSI+ strain for Sup35).

The question of whether Rnq1 and Sup35 are in an amyloid conformation or amorphyously aggregated is particularly important because the authors discuss that the “glucose depletion induced aggregation mechanism” may have important implications for neurodegenerative diseases.

We agree with the issues raised in this comment. Along these lines, we have performed several steps. We have performed staining using the Aggresome detection kit (Abcam), proven to be more sensitive than ThioflavinT staining. We have found no staining in 0.02% glucose, implying that neither of the two aggregate types contains any amyloid structures (Supplementary Figure 8). These results were added to the new version of the manuscript. Due to the absence of amyloid structures, we found irrelevant to further observe and characterize Sup35 behavior during GS. We have removed those results from the manuscript as they offered no novel insight. Also, we have removed any speculations regarding the relevance of our results in neurodegeneration, aging, or any other human condition.

3) The authors observe in competitive fitness experiments that protein placement into deposition sites as well as dissolution from them after the stress eased, is giving a fitness advantage to the cells. Therefore, it would be highly interesting to find out what the fate of the proteins after their dissolution from the Q-bodies and SGs is. Is the fitness advantage due to preventing damage of misfolded proteins in the cytoplasm by sequestering them away, or because it saves resources when the proteins can be recovered from the aggregates and refolded. Using proteasomal inhibitors and model substrates that are inactive when misfolded and active when natively folded (e.g. firefly luciferase) could give clues about this.

We are grateful to the Reviewer for posing this intriguing question, which we have indeed tried to address adequately. However, in the process of performing the experiments to address this question in its entirety, we have found that the situation is more complicated than we have expected. Reporting about the possible destinies of proteins after the aggregate dissolution would by far exceed the scope of this manuscript. We hope we will be able to communicate these results independently in due time.

However, in the new version of the manuscript, we present the results describing the effect of proteasome inhibition (using lactacystine) on aggregate dissolution, which was introduced to the cells at the end of starvation (beginning of the recovery phase), simultaneously as the glucose. We have found that the presence of this proteasomal inhibitor does not influence the post-starvation aggregate dissolution, as the dissolution occurs in the way that is consistent with the control. We added these results to the manuscript (Supplementary Figure 11).

4) In the experiments shown in Fig 6 G and H, the authors observe that inhibition of PKA as well as deletion of downstream Pfk2, aggregate dissolution is impaired. They conclude that due to a defect in restoring glycolysis, no ATP is produced. It would be a good control to add ATP to these cells after starvation to see if the effect can be overcome.

This is an excellent idea, and we have now performed this experiment – we thank the Reviewer for the suggestion. Indeed, even in the presence of PKA inhibitor Rp-cAMPS, simultaneous exposure to ATP enables the aggregate dissolution during the recovery period (Figure R6). After 90-minute starvation in 0.02% glucose, WT control cells were transferred into 2% glucose; an aliquot of WT cells was transferred into 2% glucose with Rp-cAMPS or with Rp-cAMPS, DMSO, and ATP. ATP levels were measured at the beginning of the recovery period, while the percentage of cells with aggregates after 30 minutes of recovery. We have now included these results in the manuscript (Supplementary Figure 9).

Figure R6. The exposure to ATP enables aggregate dissolution even during PKA inhibition with Rp-cAMPS., evidenced by the aggregate abundance, i.e., percentage of cells with aggregates. Bar height represents mean \pm SD from 3 separate cultures, each performed in triplicate. The mean of three technical replicates for each biological replicate is displayed as single data points. *** $p < 0.001$; ** $p < 0.01$; * $p < 0.05$ (ANOVA plus post hoc).

5) In the text describing the experiments, some of the already known features, including:

- that starvation produces aggregates (Narayanaswamy et al., 2009, O'Connell et al., 2014),
- that misfolded proteins can associate with stress granules and require Hsp104 for disassembly from SGs (Cherkasov et al., 2013, Kroschwald et al., 2015)
- and that failure to sequester misfolded proteins into protein quality control compartments reduces competitive fitness (Escusa-Toret et al., 2013).

were sometimes not stated clear enough in my opinion. Although, often they were mentioned in the discussion, I would suggest that the authors report them ideally when they describe the rationale behind their experiment in the results section. Something like “it was known before that,, therefore we wanted to further explore, ...” would make it more clear right in the beginning what was known.

We appreciate this comment. We have now clarified the relationship between the mentioned citations and our results. We have also included the O'Connell paper among the citations.

Minor points

6) The writing of the paper could be strengthened. For example, in the beginning, the rationale

behind the single experiments, but also the experimental details should be explained in a least a bit more detail to demonstrate to the reader that appropriate methods were used etc. The latter is fine to do only in the first one or two experiments, because the methods to monitor aggregation don't change. When I read the paper for the first time without having read the methods in detail, I was not sure whether the Hsp104-GFP fusion was a genomic fusion, what the Hsp104-mcherry fusion was about, that the authors used a spinning disc microscope (which is important to know, because measuring diameters of aggregates from a widefield microscopic image without deconvolution would be difficult due to the spreading of light into different layers.). All these details should be given briefly so that the reader knows that the appropriate methods were used and controls were performed.

We are grateful to the Reviewer for this comment. We have now substantially re-written the manuscript in the light of the new results, and paid attention to the issues raised by the Reviewer in this comment.

7) In most figures in the quantification of the percentage of cells with one or more aggregates, this should be given in % on the Y-axes, because it is also described as % in the main text.

We have corrected this issue.

8) In the discussion, it is stated in lines 380-381: "and that their deposition into these compartments and their timely dissolution are both required for cell survival." This is not true! When these processes are inhibited by the deletions in HSP42 or NRP1, or in the ATP hydrolysis mutant of Hsp104, the cells will not die. They are just slightly less fit, as seen in the competitive fitness assays!

The Reviewer is entirely correct; we have corrected this and similar statements.

9) In the discussion, the authors discuss that due to observations of the aggregates in time-lapse microscopy (which is not shown), they assume that these aggregates may be liquid-like compartments. To my understanding, the work of Kroschwald et al., suggested that SG in yeast (in contrast to mammalian SG) appear more likely in a solid state. Therefore, there is some apparent contradictory point here, which is hard to judge without seeing the data. Therefore, I would like to ask to include these data.

The Reviewer is correct. We were careless to make such a statement without presenting evidence, and we have now removed it from the manuscript. We aim to present these results in a separate study.

10) The authors observe here that aggregate formation does not require ATP hydrolysis by Hsp104, but dissolution does. Although this does not necessarily need to be a contradiction, Escusa-Toret and co-workers observed that coalescence of smaller aggregates into larger units during Q-body formation requires Hsp104, but degradation of their model substrate (Ubc9-ts) from Q-bodies does not need Hsp104.

It may be helpful to discuss this apparent contradiction or explain why these data may not be exclusive.

We thank the Reviewer for drawing our attention to this results of Escusa-Toret et al. As the Reviewer also mentioned, this is not necessarily a contradiction to our results. In Escusa-Toret et

al., the authors are working with a deletion of Hsp104, while we work with mutants with inactivation of ATPase activity in NBD1 or in NBD2. This is one difference that could give rise to the discrepancies in observations between us and Escusa-Toret et al. This might suggest that it could be sufficient to have either NBD1 or NBD2 active for the coalescence of smaller Q-bodies into larger to take place.

Reviewer #3 (Remarks to the Author):

The manuscript by Musa et al includes a detailed characterization of the protein aggregates that accumulate during acute glucose starvation of yeast cells. The authors find that during acute glucose starvation, misfolded proteins are compartmentalized into aggregates with hallmarks of Q-bodies (peripheral aggregates) and stress granules. Dissolution of the aggregates upon glucose refeeding depends on the disaggregating ATPase Hsp104 and the restoration of ATP levels via glycolysis. Finally, mutations that hamper aggregation management or ATP level restoration display impaired fitness when growing cells were subjected to acute glucose starvation followed by outgrowth in high glucose medium.

The data presented in the paper are in line with the present understanding of aggregation formation and dissolution. Similarly, the data showing that Hsp104 is the key disaggregating factor for aggregates accumulated during acute glucose starvation fits with the present understanding (Powis et al 2013 JCS 126(2)). It is also well established that Hsp104 is exquisitely sensitive to different nucleotide levels, including activation by ATP and inhibition by ADP (Klosowska, Chamera & Lieberk 2016 eLife. 2016; 5:e15159). The fitness experiments are reflecting the results of previously published stress-experiments (Escusa-Toret, Vonk & Frydman 2013. Nat Cell Biol 15(10)). Thus the study represents a detailed characterization of aggregate biology during acute glucose starvation of yeast.

We appreciate this summary of our work. We want to point out that our work is not only in line with previous studies but also presents several novel perspectives. We identified which compartment types are formed during glucose starvation. Moreover, our work explains why glucose starvation triggers protein aggregation: the decline in ATP levels prevents Hsp104 disaggregase from performing its function, leading to an increase in the abundance and size of these compartments. We elucidate the importance of PKA and glycolysis as the source of ATP for the appropriate function of Hsp104. Finally, our results suggest that the steady-state aggregate size is determined by the disaggregase activity of Hsp104 and not by the deposition of substrate proteins into the aggregates.

Experimental suggestions

1. The authors document that cellular ATP levels inversely correlates with size of the aggregates and impressively, addition of ATP to the cultures restores intracellular ATP levels and aggregate dissolution. The authors suggest (abstract) that Hsp104 is the key ATP-consuming component that is affected to determine aggregate abundance and size. It would be valuable to document that intracellular nucleotide levels (ATP or ADP) impact on Hsp104 disaggregation activity in vivo as has been suggested by earlier in vitro experiments. Yet the authors present no convincing evidence to substantiate this notion. The experiments with the Hsp104 hydrolysis mutant N728A is not helpful to resolve this matter since this is a dead protein in disaggregation and only shows that Hsp104-dependent disaggregation is involved. These experiments are also confounded by

the setup in which the very same mutant Hsp104-GFP is used both to manipulate the system and to monitor aggregate morphology. In fact it could be other ATP-dependent chaperones that are affected energy-stress for example Hsp40, Hsp70 and Hsp110, or even indirect effects (maybe Pma1, see below). Perhaps the authors can strengthen the notion that Hsp104 activity is impaired and rate limiting under energy depletion by quantifying the fraction of cells with aggregates in Hsp104 mutants that have residual function in disaggregation but are impaired for ATP binding. Ideally ATP affinity mutations should be used. Walker B mutants are available and characterized (Klosowska, Chamera & Lieberk 2016 eLife. 2016; 5:e15159).

We entirely agree with the Reviewer on this issue. We have therefore used two Walker B mutants, generously provided by Dr. Krzysztof Lieberk: Hsp104^{E285Q} in the NBD1 domain, and Hsp104^{E687Q} in the NBD2 domain (HAP variants). These mutants are characterized by abolished ATP hydrolysis in the affected NBD, while maintaining its binding. The presented results show that even in optimal conditions, with no additional protein stress, the aggregate size and abundance rely strongly on protein dissolution rather than their deposition into aggregates. The two mutants in the Walker B motif of the Hsp104 NBD1 and NBD2 domains display considerable differences in their aggregation propensities during glucose starvation, which might stem from the allosteric interactions between NBD1 and NBD2. Finally, exposure to external ATP during glucose starvation (0.2% or 0.02%) failed to reduce the abundance of the aggregates or their size in both studied mutants (Figure 2A-D), suggesting that ATP alone, without the ATPase activity of Hsp104, is not sufficient to dissolve the aggregates.

2. Loss of fermentation upon acute glucose starvation results in that the plasma membrane ATPase Pma1 cannot maintain the plasma membrane proton gradient. Acidification of the cytoplasm subsequently drives protein aggregation (Munder et al 2016 eLife 5:e09347 and references within). This raises the concern that the accumulation of the aggregates upon acute glucose starvation in the present study is the result of cytoplasmic acidification. Thus the effects would be an indirect effect of loss of Pma1 function. The level of acidification depends on the level of ATP depletion and the pH of the conditioned medium used in the experiment. Intracellular pH can be checked with pH-sensitive variants of GFP and the pH of the medium can be controlled with buffers.

We are grateful to the Reviewer for this comment. We have entirely taken it into account and performed the experiments related to this issue. The results are included in the new version of the manuscript.

We conducted the following experiments. First, using a pH-sensitive form of GFP (yeast pHluorin), we confirmed that the pH of the cytosol indeed acidifies during glucose deprivation (Supplementary Figure 10), observed as the pHluorin fluorescence decrease at 0.2% and 0.02% glucose.

Next, we aimed to test if the decline in pH alone can yield the same effects on protein aggregation as glucose deprivation. To test this, we exposed the cells to protonophore DNP in the presence of 2% glucose, and in two different pH conditions: (i) standard YPD medium with pH 5.5, and (ii) the YPD medium in which pH is brought to 7. The exposure of cells to the DNP enables pH equilibration between the extracellular environment (the medium) and the cytosol. Using the pH-sensitive GFP, we confirmed that the pH of the cytosol of DNP treated cells corresponds to the one of the medium (Supplementary Figure 12 and 13). We then quantified the abundance of protein aggregates in the conditions described above and confronted the results to those of glucose

starvation (Figure 6D). In optimal (2%) glucose, in the presence of DNP at pH 5.5, an increase in Hsp104-GFP containing aggregate abundance can be seen: the fraction of cells with one aggregate is ~5% and with two or more aggregates ~45%. However, at pH 5.5, we observed the formation of multiple (>10) speckles per cell, characterized by irregular shape, not resembling the spherical shape of the Q-bodies and SGs found during glucose starvation.

Moreover, to additionally test the contribution of cytosol acidification in GS-induced protein aggregation, we performed the following experiment: yeast cells were placed into the medium with 0.02% glucose and pH 7 and were exposed to DNP in order to allow the equilibration of the pH between medium and cytosol. In this way, the cells were exposed to starvation; however, without the decline in pH. In these conditions, the aggregate abundance corresponds to the one observed in 0.02% glucose alone, suggesting that the decrease in pH to 5.5 is not essential for the GS-triggered protein aggregation.

While there is no doubt that the acidic pH drives the segregation of proteins into higher-order assemblies, as previously suggested, that alone seems not to be the primary determinant of protein aggregation into Q-bodies and stress granules during acute glucose starvation stress (Figure 6D).

3. The authors interpret the experiment in Fig 7A-C to suggest that Hsp42, Hsp104 and Nrp1 all are required for cells survive glucose starvation. However, according to the experimental protocol (p. 18) control cells ('2% glucose', Fig 7C) are not treated with the same extensive centrifugation and washes as the glucose starved cells, raising the concern that some other confounding factor (changed pH, temperature, oxygen or exposure to unconditioned medium et c) is the real stressor. To rule out such indirect effects the experiment should be redone and the control cells should be subjected to the same treatment as the glucose starved cells. Perhaps this best achieved by simply diluting the growing cells from 2% glucose to the new media and thereby avoid centrifugation all together. Moreover performing these experiments with kanMX knocks without complementation controls is not an ideal setup.

We agree with the Reviewer that the control cells were not undergoing the same treatment according to the protocol applied in the original version of the manuscript. We have, therefore, repeated the experiments, and the control cells were treated in the same way as the starved cells. These results are now included in the revised version of the manuscript as Figures 7B and 7C.

We introduced other modifications to this experiment:

- In the original version of this manuscript, we presented the results of the Hsp104^{N728A} mutant, which was, according to the Reviewers, not the best mutant to be tested since the protein is 'dead.' We have, therefore, used the mutant with the mutation in NBD2 (Hsp104^{E687Q}), since this mutant displayed a less severe phenotype and allowed us to measure changes in competitive fitness.
- We removed the results on the Nrp1-null mutant since working with that mutant raised many questions by Reviewer #1, which we will address and present in a separate report.

Formatting and data presentation

1. Throughout the manuscript more detailed descriptions are needed regarding how the experiments were performed, particularly regarding the time cells were subjected to glucose starvation in each experiment. Perhaps editing the figure legends and focusing more on description of the experiment and less on interpretation of the data can solve this?

We are grateful to the Reviewer for this advice. We edited the manuscript in the light of the new results, while also paying attention to the issues raised in this comment.

2. The microscopy pictures are generally too dark and do not survive printing.

Thank you – we have taken care of this issue.

4. Fig 6 needs extensive graphical editing - labels and microscopy data is simply too small to be assessed.

We have taken care of this issue.

5. Update figure labels and text to follow standard yeast nomenclature: For example hsp104Δ (it.) for a genotype and Hsp104 for a protein.

We have now improved the quality of figures in general, while also taking care of the mentioned issues.

6. The Results section contains both past and present tense when describing experiments and it makes it hard to read.

We have corrected the mentioned problem, thank you for drawing our attention to this.

7. Line 306: The sentence starts 'Remarkably' but it is not remarkable at all that a disaggregation defective Hsp104 does not support dissolution of aggregates.

We have now corrected both of these problems.

8. The graphical artwork in Fig 7D is not very helpful. It does not summarize the data in a clear way and it does not function as a model to generate new hypotheses.

We have now adapted the description of this panel and now the description and the graphic are more coherent with the actual results.

REVIEWER COMMENTS

Reviewer #2 (Remarks to the Author):

The revised version of the manuscript by Sathyanarayanan et al has impressively improved! Many new and nice experiments and controls were added. The overall quality of the manuscript is very high and many of my previous concerns have been addressed sufficiently. However, before I can recommend publication, two major points that concern central conclusions made in the paper, namely that Hsp104-GFP positive foci that form upon GS contain misfolded proteins and that part of these foci represent Q-bodies, have to be addressed. At the moment, the experiments presented regarding the above-mentioned two central claims of the paper do not fully justify the conclusions the authors make.

Beyond that, I have a few minor concerns that should also be addressed!

Major concerns:

1) On pages 9-10, lines 303 – 322, the authors conclude that some of the Hsp104-GFP foci formed represent Q-bodies. This conclusion is partly based on the observation that the sum of percentage of cells with Hsp104-GFP foci in an hsp42 knock-out strain (Fig 4 b) and the percentage of cells with Hsp42-GFP foci (Fig 4 c-d) equals the percentage of cells with Hsp104-GFP in Fig 1a, b. Due to this calculation, the authors conclude that the foci with Hsp42-GFP must also contain Hsp104-GFP, because those where the Q-bodies that disappeared upon hsp42 deletion (Fig 4 b). This is a correlation, but no proof at all. Why don't they simply do a co-localization of Hsp42-GFP and Hsp104-mcherry, as they did for Ssa-GFP and Hsp104-mcherry? This could proof their claim easily that the foci observed contain both Hsp42 and Hsp104 and may therefore represent Q-bodies! It should be a very easy experiment to perform and the authors have all the tools.

2) On page 11, lines 351 – 362, the authors describe experiments aiming to answer a very crucial question to one major conclusions of the paper, which is that the Hsp104-GFP foci they observe during GS really contain misfolded proteins! For this, they use A2C to produce misfolded proteins through incorporation of A2C during protein synthesis. The Hsp104-GFP foci that are shown to form upon A2C addition in Supplementary Fig 7 b show a completely different pattern, namely multiple Hsp104-GFP foci per cell. When I compare this to Supplementary Fig 2 c (which is the proper Fig to compare because only here, the same Tef promoter is used as compared to Supplementary Fig 7 b), almost all cells have 1 or at maximum 2 visible aggregates. This also holds true for Fig 1 b by the way! To me, that offers the possibility that the proteins that misfold due to A2C incorporation form additional foci to the ones seen in Supplementary Fig 2 c or Fig 1 b! Thus, this experiment does not show that misfolded proteins are present in the Hsp104-GFP positive foci that form upon GS at all!

Minor concerns:

1) Page 13, lines 420 – 434, and the corresponding Supplementary Fig 11 b, c: different proteasome inhibitors inhibit distinct proteolytic activities of the proteasome, but not all different proteasomal proteolytic activities. Therefore, the experiment lacks a positive control that proteasome inhibition worked sufficiently in these experiments! There are several misfolded protein model substrates known that are degraded by the proteasome that could be included as positive control.

2) The amyloid staining test shown in Supplementary Fig 8 needs a positive control that the staining worked. When I remember correctly, the authors also used Sup35-GFP prion fusions in the original manuscript. That might be a good positive control!

3) In Fig 5c (the corresponding text is on page 10, lines 333 – 335), I see hardly any clear Rnq1-mcherry foci (in contrast to Fig 5 a and b). Because of this, I am not convinced that there is a lack of co-localization with Hsp42-GFP (because there are no Rnq1-mcherry foci visible). The authors

should pick better pictures to demonstrate a lack of co-localization.

4) In the supplementary Fig 5, panel B, one should indicate what "circles" and "squares" represent

5) The figure legend for Fig 2 lacks the description for panel D.

6) On page 6, line 199, I missed an explanation what "HAP" means.

7) In the discussion, the authors claim (page 17, lines 543 – 548) that "the ATP promotes maintenance of soluble protein state, not by directly preventing aggregation, but by enabling protein retrieval from the aggregates". I agree in principle that protein retrieval from aggregates has a major, maybe underestimated, role in maintenance of the soluble protein state. However, I would not say that preventing aggregation is not promoted by ATP. I believe that ATP-dependent chaperones like Hsp70 and Hsp90 will also contribute to maintenance of the soluble protein state, but maybe not to such a great extent as previously assumed.

8) In page 3, line 106, it should say supplementary figure 2, not supplementary figure 1.

Reviewer #3 (Remarks to the Author):

This manuscript has significantly improved during the extensive revision and important experiments have been added. In its current form I consider it a detailed characterization of yeast aggregate biology during acute glucose starvation and, importantly, the study highlights that falling ATP levels prevent the Hsp104 disaggregase from performing its function resulting in increased accumulation of aggregates. The main concerns that was raised in the review have been dealt with.

Aspects of the study have been touched upon by other studies but I am not aware of any study that have provided this clear link between energy status and how strongly it impacts on the accumulation of aggregates due to failed disaggregation. This is conceptually important and in the event of publication the study may turn out to be an important reference for the field. The link between metabolism and proteostasis is timely.

The mechanistic core claim of the study, that cellular Hsp104 is inhibited by falling ATP levels is difficult to experimentally address directly, but in light of previous the biochemical characterization of Hsp104 and the experiments and measurements actually provided in this study, I agree with the authors that this is the best interpretation of the data.

The manuscript still needs a brush over to reach publication standard:

General comments on presentation

- All figures would benefit, if the font size of the labels was increased. Further, some datasets are hard to distinguish, especially those with two y-axes (e.g. Fig. 3A), and might need improvement.
- The description of the statistical analyses need improvement (and should be added as separate paragraph to the "Methods" section). It must be stated, which ANOVA and which post hoc test was applied and if all statistical assumptions required for the respective tests were met (and how they were tested). Further, the minimal number of cells quantified in each experiment should be stated.
- The clarity of the figure legends would improve, if the result description was moved to or solely described in the results-section, while details about measurements/strains were listed in the legend.
- Analysis of aggregate diameter (e.g. Fig.1 B, C; Fig. 2C,D): Even though the distribution is interesting, statistical analyses and providing of mean/median-based graphs would strengthen the described phenotypes.
- Please avoid green-red combinations in micrographs (e.g. Fig 4 A, Fig.5). Either apply grayscale-LUTs for single channels or switch to green-magenta combinations.
- Please avoid describing data from the same subpanel in different results-subsections (e.g. Fig. 1A). This would improve the general understanding for the reader and would help to shorten the

results section.

- Please add control conditions in all experiments. Non-starved and/or untreated cells are required to assess any changes by starvation and/or treatments (e.g. Suppl. Fig. 5)
- Hsp104 is expressed either under its native promoter or under the control of the TEF1 promoter. Even though no striking differences were observed regarding aggregate size, the authors should stick to the same promoter throughout the manuscript and not exchange it arbitrarily. If a change in the promoter used is required, please clearly state it, and provide the rationale.

Specific comments sorted by line number

Title

Line1: The title would improve, if Hsp104 and yeast / *Saccharomyces cerevisiae* would be included.

Introduction

Line 58: The authors introduce the age-associated protein deposit requiring Hsp42. Did they test for this compartment as well? Otherwise, remove this information or approach it at least in the Discussion.

Line78: It would be helpful, if the authors added a short introduction on stress granules as well.

Results

Line106: As no results of Suppl. Fig. 1 are discussed in this paragraph, please remove the cross-reference in brackets.

Line110/111: This sentence is somewhat redundant with the sentence directly above.

Line115ff: As stated above, please describe data from one subpanel in the same paragraph/subsection of the results.

Line115: It would help if subpanel 1A was divided, instead of referring to "left" and "right panel", use "A" and "B" instead.

Line115-120: The determination of aggregate diameter would benefit from a statistical analysis, as the authors even refer to a "significant fraction" (see general comments).

Line140: Please introduce GS (glucose starvation) when it is used for the first time (e.g. Line74, 77, 98...) and use it consistently. But I would rather avoid this abbreviation, as the authors use both "glucose starvation" and "glucose deprivation" in this manuscript.

Line140+146-150: These statements would be strengthened, if the authors added a quantification of Hsp104 protein levels upon different starvation conditions and strains using immunoblotting.

Line142: Refer to Suppl. Fig. 2 C instead of Suppl. Fig. 2

Line144: Please include a statistical analysis of these data to use "trend of significant aggregate growth".

Line163ff: As data described here are already shown in Fig.1, either implement this subsection in the one above or split the graphs to create two independent figures. If the second option is chosen, please add control conditions as well.

Line163ff: Please provide a description of the ATP/ADP/GTP treatment in the "Methods" section and state the concentration and length of incubation.

Line167: The timeframe of this experiment is not clear in this description. Was DMSO and/or ATP added after 60 min of incubation in the respective starvation medium? If so, please adapt the labelling in Suppl. Fig. 4.

Line173ff: As mentioned earlier, either include this paragraph in the subsection above or split Figure 1 and add respective controls.

Line179: Please briefly provide rationale, why ADP and GTP were added.

Line198: Please introduce the mutants better. Even though these mutants are unable to hydrolyse ATP in the respective NBD domain, the second domain is still functional and, according to in vitro measurements, the ATP hydrolysis rate increases by 300% in the E285Q mutant and its inhibition by ADP is comparable to wildtype Hsp104 (PMID 27223323; PMID 17543332). Further, the E687Q mutant should still possess similar ATPase activity as the wildtype, while its inhibition by ADP is reduced (PMID 27223323). It could thus be a possibility to inhibit wt Hsp104 and E687Q-Hsp104 by adding ADP or assessing cellular ADP levels.

Line199: Please briefly introduce the term HAP mutants (as used in Jackrel et al 2014)

Line220: A quantification of aggregate number per cell would support the notion of aggregates fusing together in these strains.

Line233: The observed phenotypic differences between the mutants could also arise from their different ability to hydrolyse ATP and the different degree of inhibition by ADP. Please add a section to the Discussion, where cellular ADP levels and thus inhibition of Hsp104-variant are brought into context. Further, please provide an explanation why addition of external ADP did not inhibit Hsp104 activity.

Line244: Please exchange the word "tagged" with "decorated" or a similar term.

Line248: Please state that 0.02% glucose was used in this experiment.

Line248: This stands in contrast to ATP measurements shown in Suppl. Fig. 4A (where the minimal value assessed was around 4 mM after 30 min). Please briefly provide an explanation for this divergence. Further, the minimum is reached after 60 min at about 1 mM.

Line249: Rather than using "left and right" panel, split this subpanel into A and B (same for Fig. 3 E).

Line250: Please provide the median of aggregate diameter.

Line255: As exemplified above: Even though one Walker B domain is inactive, the other can hydrolyse ATP. Thus, avoid using the term "inability" for the whole protein, rather refer only to NBD2.

Line273/274: Please rephrase this statement.

Line275: To substantiate this hypothesis, it would be helpful to quantify the number of aggregates per cell over time and compare wt-Hsp104 with mutant-Hsp104 upon glucose starvation and control conditions.

Line279: Please briefly state that GdnHCl inhibits Hsp104 activity and provide a reference.

Line279 and Suppl. Fig. 5: A wildtype control is necessary in this experiment.

Line294: Please state, which promoter was used for the endogenous Hsp104-mCherry chimera.

Line298: To provide quantitative data on colocalization, assessment of e.g. Mander's overlap coefficient and Pearson's correlation coefficient would be suitable (easily done with Image J plugins).

Line310: Please include the respective control and avoid referring to datasets in other figures.

Line319: The theory should be substantiated by colocalization experiments (including quantification as described above).

Line327: Please be consistent with abbreviations: Stress granules – SG was introduced in the introduction

Line330ff: Please provide a colocalization quantification (as described above).

Line337: The abbreviation SG was introduced earlier.

Line338: Please provide a reference for Nrp1 as marker for SG.

Line341: Please provide a colocalization quantification, as described above.

Line358: It would be interesting to see to what extent A2C increases aggregate length compared to control cells (see comment on Suppl. Fig. 7)

Line363ff: Please do not include the description of a new figure (Suppl. Fig. 8), before CHX treatment of Suppl. Fig. 7 has been described.

Line373: Please avoid referring to data of other figures and include the respective control.

Line405: It would be interesting to also include a condition, where Rp-cAMPS is added to cells prior to shifting them to starvation conditions to assess aggregate abundance and size.

Lin408: Referring to Fig. 6 A, B does not fit here.

Line411: Please rephrase this statement for more clarity.

Line446: The assessment of cytosolic pH using pHluorin would require at least measurement of signal intensity at two different wavelengths (or other possibilities as described in the "Methods" section in more detail), to exclude any effects from different expression levels.

Line450: Even though Pma1 is a dominant proton pump, there are other possible explanations why the cytosol acidifies upon shifting them to glucose starvation medium. As no experiments with Pma1 are presented here, I would rather move this paragraph to the Discussion, where Pma1 but also other possibilities can be discussed in more detail.

Line455: Even though addition of DNP led to a change in pHluorin fluorescence intensity it may have many side effects since it also interferes massively with mitochondria (respiratory chain...) and immediately increases cAMP levels.

Line468: The observed phenotype at pH 5.5 in the presence of DNP rather hints at unspecific effects, interpret this phenotype with caution.

Line470ff: The same concerns regarding the use of DNP remain, but please provide a respective quantification or remove this paragraph.

Line493: It is unusual that only 20% of exponentially growing cells display acidified vacuoles. Please state the criteria used to distinguish between acidified vs non-acidified vacuoles in the "Methods" section.

Line494: Please avoid the term "physiological pH", as no direct and quantitative assessment of vacuolar pH is presented.

Line497: Please provide details on concA concentration and treatment duration in the "Methods" section.

Line505: Even though autophagy requires a functional vacuole, the authors did not measure pH quantitatively and did not directly test aggregate clearance in autophagy-impaired mutants, thus I would either remove this hypothesis or provide experimental evidence.

Line514: As described below in further detail, it is necessary to check for any artefact by the kanamycin resistance. Thus, the kanamycin-cassette also needs to be introduced into wildtype cells and the experiments should be repeated.

Line536/537: Please rephrase this sentence, as the correlation here is not clear.

Line541: Please provide the quantification of average number of aggregates per cell (as also stated earlier).

Line545: The authors did not show that ATP prevented aggregation. Rather glucose (ATP) depletion led to accumulation of aggregates and its re-feeding resulted in dissociation. The authors may interpret their data but need to rephrase it accordingly.

Line547: As described above, one NBD each remained active, thus the tested Hsp104 variants are not unable to hydrolyse ATP, as only one Walker B domain is mutated.

Line554: Please discuss the fact that E285Q mutant shows a 300% increase in ATPase activity in vitro.

Line561: Please also discuss respective in vitro measurements of the E687Q mutant.

Line568: Even though DNP indeed leads to immediate intracellular pH adaptation to the extracellular milieu, other effects include increase in cAMP levels Please mention other consequences here.

Line570: This statement is not substantiated by experimental evidence provided in this paper.

Line585: Please modify the summary figure by adding more detail. As it is now, it does not help to recapitulate the main findings of the manuscript.

Figures

Fig.1A: Please provide asterisks in the graph and state, which bars were compared to provide the respective significance level.

Figure 1: If DMSO was added here as well to enable ATP/ADP/GTP entering the cells, please state this in "Methods" or in the figure legend.

Suppl. Fig. 1A: Please provide significance asterisks for all brackets given (e.g. between 0.2% glucose 30 min and 60 min)

Suppl. Fig. 1B: please exchange "WT S288C" to "2% glucose" or "control". Further, please refer to the commonly-used style of setting gene names in italics (e.g. HSP42).

Suppl. Fig. 3B: In this graph, the aggregate change upon DMSO treatment is of interest, please include a statistical analysis between untreated/treated cells as well.

Suppl. Fig. 4: Please add a control condition without DMSO, as especially the ATP content of cells incubated in 0.02% glucose for 30 min differs from values in Suppl. Fig. 1A (4 mM vs. 2.5 mM).

Fig. 2 C, D: I would recommend to statistically analyse the aggregate diameter.

Figure2: If DMSO was added here as well to ensure ATP uptake, please state it in the figure legend. Further, the legend does not fit to the data presented.

Fig. 3 A: Please depict the aggregate diameter in nm, as the main text refers to their absolute size.

Fig. 3C, D: This figure is hard to read, please use either different colours for the legend description "assembly" and "disassembly" or depict assembly and disassembly curves behind each other (starting with 90 min of starvation, followed by 90 min of re-feeding).

Suppl. Fig 5B: please consider comments on Fig 3A.

Figure 4,5: Please avoid red-green combinations in micrographs (as stated in general comments).

Fig. 4B: Please include the wildtype control.

Fig. 4C: As stated earlier, please provide a statistical analysis here.

Suppl. Fig. 7: Please add the control condition without any treatment.

Suppl. Fig. 7 B, C: Please provide a statistical analysis.

Fig. 6: Please add the respective control strains. Further, try to improve the comprehensibility of the graphs (implement comments above, especially clearly state, which colour/shape represents what)

Fig. 6A: Please provide an explanation, why no cytosolic Hsp104 signal can be observed upon addition of Rp-cAMPS.

Suppl. Fig. 9: Please clearly describe the conditions used (as already stated above).

Suppl. Fig. 11: Please provide the respective untreated control condition.

Figure 7 B, C: Please provide an explanation, why the competitive fitness of HSP42KO/Hsp104mutant cells is around 0.9 resp. 0.8 in (B) and above 1 in (C).

Methods

Line609: Even though the strains can be found in Suppl. Table 1, the respective growth medium is not listed there.

Line610: Please provide the composition of the used growth medium.

Line613: If control cells (in 2% glucose) were also treated the same way as starved cells (including centrifugation and washing steps), please state it here.

Line617: Did the authors use both WT and HAP mutants in their study? If so, please state it in the respective figure legend or rephrase this sentence here.

Line645: To avoid artefacts by different expression levels in different media (2% - 0.2% - 0.02% glucose), either use the technique of assessing the fluorescence intensity at two wavelengths (e.g. as described in your reference 45), design a tandem fluorophore construct composed of pH-sensitive pHluorin and pH-insensitive mCherry (similar as used in Rosella) or check respective expression levels with immunoblotting.

Line656: Please provide more details on your protocol, as this kit from Abcam is designated for mammalian cells and the protocol might require adaption for yeast, which might help other researchers making use of your method.

Line662: Please state the glucose concentration you used for preparation of the agar slides for microscopy.

Line737: The competition assay would benefit, if both strains were inoculated to the same optical density/cell number. Alternatively, please provide growth curve measurements to ensure that both strains reach the same optical density at "saturation".

Line738: It would be vital to check, if both strains are equally represented in the culture before transferring them to different starvation conditions (e.g. by plating them before treatment as well).

Line746ff: To avoid artefacts by genomic integration of the kanamycin cassette, it is advisable to add the selection marker to wildtype cells and repeat this experiment.

REVIEWER COMMENTS

Reviewer #2 (Remarks to the Author):

The revised version of the manuscript by Sathyanarayanan et al has impressively improved! Many new and nice experiments and controls were added. The overall quality of the manuscript is very high and many of my previous concerns have been addressed sufficiently. However, before I can recommend publication, two major points that concern central conclusions made in the paper, namely that Hsp104-GFP positive foci that form upon GS contain misfolded proteins and that part of these foci represent Q-bodies, have to be addressed. At the moment, the experiments presented regarding the above-mentioned two central claims of the paper do not fully justify the conclusions the authors make. Beyond that, I have a few minor concerns that should also be addressed!

We are grateful to the Reviewer #2 for the positive feedback, which we found motivating. Below, we address the specific questions.

Major concerns:

1) On pages 9-10, lines 303 – 322, the authors conclude that some of the Hsp104-GFP foci formed represent Q-bodies. This conclusion is partly based on the observation that the sum of percentage of cells with Hsp104-GFP foci in an *hsp42* knock-out strain (Fig 4 b) and the percentage of cells with Hsp42-GFP foci (Fig 4 c-d) equals the percentage of cells with Hsp104-GFP in Fig 1a, b. Due to this calculation, the authors conclude that the foci with Hsp42-GFP must also contain Hsp104-GFP, because those where the Q-bodies that disappeared upon *hsp42* deletion (Fig 4 b). This is a correlation, but no proof at all. Why don't they simply do a co-localization of Hsp42-GFP and Hsp104-mcherry, as they did for Ssa1-GFP and Hsp104-mcherry? This could proof their claim easily that the foci observed contain both Hsp42 and Hsp104 and may therefore represent Q-bodies! It should be a very easy experiment to perform and the authors have all the tools.

We thank the Reviewer for raising this question. Indeed, the experiment suggested by the Reviewer was one of the first experiments we performed in this study. Indeed, we found that Hsp42 colocalizes with only a fraction of Hsp104-tagged aggregates, however, we also found Hsp42 to colocalize with some of the Nrp1 foci, and some Rnq1 foci. Therefore, performing the suggested experiment could not distinguish with confidence between Q-bodies and stress granules. Our main evidence (in addition to other analyses) remains the loss of a fraction of Hsp104-GFP aggregates in *hsp42Δ* and the colocalization of the remaining Hsp104-tagged aggregates with Rnq1 and Nrp1. The localization of Hsp42 into the stress granules has already been reported (Specht et al., 2011; Kroschwald et al., 2015), and we are also looking into the details of the role of Hsp42 in stress granules, which will be reported in a separate study.

2) On page 11, lines 351 – 362, the authors describe experiments aiming to answer a very crucial question to one major conclusions of the paper, which is that the Hsp104-GFP foci they observe during GS really contain misfolded proteins! For this, they use A2C to produce misfolded proteins through incorporation of A2C during protein synthesis. The Hsp104-GFP foci that are shown to form upon A2C addition in Supplementary Fig 7 b show a completely different pattern, namely multiple Hsp104-GFP foci per cell. When I compare this to Supplementary Fig 2 c (which is the proper Fig to compare because only here, the same Tef promoter is used as compared to Supplementary Fig 7 b), almost all cells have 1 or at maximum 2 visible aggregates. This also holds true for Fig 1 b by the way! To me, that offers the possibility that the

proteins that misfold due to A2C incorporation form additional foci to the ones seen in Supplementary Fig 2 c or Fig 1 b! Thus, this experiment does not show that misfolded proteins are present in the Hsp104-GFP positive foci that form upon GS at all!

This is indeed correct and it is a very interesting question. We have therefore included a thorough analysis of the consequences of A2C treatment to elucidate it. As already presented in our manuscript, in the presence of A2C, the fraction of cells with Hsp104-tagged aggregates increases to over 80% in both starvation regimes. Their median diameter is 450 nm in 2% glucose, and 900 nm and 1500 nm in 0.2% and 0.02% glucose, respectively. This represents an increase compared to the control, where median diameters on 600 nm and 750 nm were observed. In the *hsp42Δ* background, ~50% of aggregates remained, compared to ~20% in the control. Out of the remaining aggregates, only ~20% colocalized with Rnq1, meaning that there is indeed a third type of inclusions formed in the presence of A2C. In the total 'population' of inclusions formed during A2C treatment, Q-bodies represent ~45%, stress granules ~22% and the third type ~25%. Looking at the distribution of aggregate diameters, in the presence of A2C it was strongly shifted towards larger sizes, with only a minor fraction of the inclusion population being in the size range observed in the absence of A2C. Even though this is not direct evidence, these results strongly suggests that all the inclusion types grow in the presence of A2C. These results are now presented in Supplementary Fig. 10, panels a-f. We hope we have satisfactorily addressed this question.

Minor concerns:

1) Page 13, lines 420 – 434, and the corresponding Supplementary Fig 11 b, c: different proteasome inhibitors inhibit distinct proteolytic activities of the proteasome, but not all different proteasomal proteolytic activities. Therefore, the experiment lacks a positive control that proteasome inhibition worked sufficiently in these experiments! There are several misfolded protein model substrates known that are degraded by the proteasome that could be included as positive control.

We thank the Reviewer for drawing our attention to this issue. We have now included an experiment that serves as the control to show that lactacystin has indeed inhibited the proteasome. We have performed a qPCR based measurement (reported in Lee, D. H. & Goldberg, A. L. Proteasome Inhibitors Cause Induction of Heat Shock Proteins and Trehalose, Which Together Confer Thermotolerance in *Saccharomyces cerevisiae*. *Mol. Cell. Biol.* **18**, 30–38 (1998)) of expression of several genes from the heat shock response, namely *Sis1*, *Ydj1*, *Hsp104* and *Ssa1*. According to the paper mentioned above, these genes are upregulated due to the action of lactacystin, and since we had cDNA stored from the exact samples that were imaged, we opted to use this assay. Our results show that the inhibition of the proteasome was effective in the presence of lactacystin, reflected in the upregulation of the mentioned genes, compared to the same conditions without lactacystin. These results confirm the proteasomal inhibition by lactacystin, and we have now included those results in the manuscript.

2) The amyloid staining test shown in Supplementary Fig 8 needs a positive control that the staining worked. When I remember correctly, the authors also used Sup35-GFP prion fusions in the original manuscript. That might be a good positive control!

We are grateful to the Reviewer for pointing out the missing control experiment. We have now included the positive control. As a positive control, we have used the strain with a genomic fusion of Sup35-GFP, and performed heat shock at 37°C for 20 minutes during their exponential growth stage. We have now included these results in the manuscript.

Response to Reviewers' Comments

3) In Fig 5c (the corresponding text is on page 10, lines 333 – 335), I see hardly any clear Rnq1-mcherry foci (in contrast to Fig 5 a and b). Because of this, I am not convinced that there is a lack of co-localization with Hsp42-GFP (because there are no Rnq1-mcherry foci visible). The authors should pick better pictures to demonstrate a lack of co-localization.

Indeed, the selection of the representative image was not optimal. We have now replaced the previous image with a new one, which better represents the statistical analysis of these conditions.

4) In the supplementary Fig 5, panel B, one should indicate what “circles” and “squares” represent

We have now indicated the meaning of circles and squares in the Supplementary Fig. 5, as well as in the other similar figures.

5) The figure legend for Fig 2 lacks the description for panel D.

We have now taken care of this issue.

6) On page 6, line 199, I missed an explanation what “HAP” means.

We have now included an explanation and relevant references regarding the HAP mutant of Hsp104.

7) In the discussion, the authors claim (page 17, lines 543 – 548) that “the ATP promotes maintenance of soluble protein state, not by directly preventing aggregation, but by enabling protein retrieval from the aggregates”. I agree in principle that protein retrieval from aggregates has a major, maybe underestimated, role in maintenance of the soluble protein state. However, I would not say that preventing aggregation is not promoted by ATP. I believe that ATP-dependent chaperones like Hsp70 and Hsp90 will also contribute to maintenance of the soluble protein state, but maybe not to such a great extend as previously assumed.

This is an excellent point. We agree with the Reviewer and we have rephrased this part of Discussion.

8) In page 3, line 106, it should say supplementary figure 2, not supplementary figure 1.

We have now corrected the wrong figure citation.

Reviewer #3 (Remarks to the Author):

This manuscript has significantly improved during the extensive revision and important experiments have been added. In its current form I consider it a detailed characterization of yeast aggregate biology during acute glucose starvation and, importantly, the study highlights that falling ATP levels prevent the Hsp104 disaggregase from performing its function resulting in increased accumulation of aggregates. The main concerns that was raised in the review have been dealt with.

Aspects of the study have been touched upon by other studies but I am not aware of any study that have provided this clear link between energy status and how strongly it impacts on the accumulation of aggregates due to failed disaggregation. This is conceptually important and in the event of publication the study may turn out to be an important reference for the field. The link between metabolism and proteostasis is timely.

The mechanistic core claim of the study, that cellular Hsp104 is inhibited by falling ATP levels is difficult to experimentally address directly, but in light of previous the biochemical characterization of Hsp104 and

Response to Reviewers' Comments

the experiments and measurements actually provided in this study, I agree with the authors that this is the best interpretation of the data. The manuscript still needs a brush over to reach publication standard:

We thank the Reviewer #3 for the affirmative description of our work. Moreover, we are grateful to the Reviewer #3 for taking the time to provide us with such a detailed revision of our manuscript, which truly help us a lot to improve its quality. Below, we addressed in details the individual concerns listed.

General comments on presentation

- All figures would benefit, if the font size of the labels was increased. Further, some datasets are hard to distinguish, especially those with two y-axes (e.g. Fig. 3A), and might need improvement.

Done.

- The description of the statistical analyses need improvement (and should be added as separate paragraph to the "Methods" section). It must be stated, which ANOVA and which post hoc test was applied and if all statistical assumptions required for the respective tests were met (and how they were tested). Further, the minimal number of cells quantified in each experiment should be stated.

We have now taken care of all these points.

- The clarity of the figure legends would improve, if the result description was moved to or solely described in the results-section, while details about measurements/strains were listed in the legend.

We took care of this issue.

- Analysis of aggregate diameter (e.g. Fig.1 B, C; Fig. 2C,D): Even though the distribution is interesting, statistical analyses and providing of mean/median-based graphs would strengthen the described phenotypes.

We agree with this suggestion, and we have now added a statistical analysis of the aggregate diameter distributions. We discuss the median values in the text.

- Please avoid green-red combinations in micrographs (e.g. Fig 4 A, Fig.5). Either apply grayscale-LUTs for single channels or switch to green-magenta combinations.

We agree, and all cases where red-green combinations were presented, have now been converted into monochrome.

- Please avoid describing data from the same subpanel in different results-subsections (e.g. Fig. 1A). This would improve the general understanding for the reader and would help to shorten the results section.

We took care of this issue.

Response to Reviewers' Comments

- Please add control conditions in all experiments. Non-starved and/or untreated cells are required to assess any changes by starvation and/or treatments (e.g. Suppl. Fig. 5)

Control conditions were now added to all experiments.

- Hsp104 is expressed either under its native promoter or under the control of the TEF1 promoter. Even though no striking differences were observed regarding aggregate size, the authors should stick to the same promoter throughout the manuscript and not exchange it arbitrarily. If a change in the promoter used is required, please clearly state it, and provide the rationale.

We agree, and this is now the case, unless stated otherwise. In the cases where exceptionally TEF1 promoter was not used, we provided an explanation as to why this is the case.

Specific comments sorted by line number

Title

Line1: The title would improve, if Hsp104 and yeast / *Saccharomyces cerevisiae* would be included.

We have now modified the title of the manuscript to reflect the suggestions of the Reviewer, as well as the maximal character count recommended by the Journal.

Introduction

Line 58: The authors introduce the age-associated protein deposit requiring Hsp42. Did they test for this compartment as well? Otherwise, remove this information or approach it at least in the Discussion.

We have now removed this sentence from the Introduction since, indeed, we do not discuss this aggregate type further on.

Line78: It would be helpful, if the authors added a short introduction on stress granules as well.

We agree with this suggestion and we have now included a brief introduction on stress granules.

Results

Line106: As no results of Suppl. Fig. 1 are discussed in this paragraph, please remove the cross-reference in brackets.

We have now removed the cross-reference to Supplementary Fig. 1.

Line110/111: This sentence is somewhat redundant with the sentence directly above.

We agree, and we have now taken care of this issue.

Line115ff: As stated above, please describe data from one subpanel in the same paragraph/subsection of the results.

Response to Reviewers' Comments

We have corrected this issue throughout the entire manuscript.

Line115: It would help if subpanel 1A was divided, instead of referring to “left” and “right panel”, use “A” and “B” instead.

We have divided the panel 1A into panel 1A and 1B.

Line115-120: The determination of aggregate diameter would benefit from a statistical analysis, as the authors even refer to a “significant fraction” (see general comments).

We thank the Reviewer for drawing our attention to this issue. We performed the Kolmogorov-Smirnov statistical test to compare the distributions of aggregate sizes in different conditions to the control conditions. We included the results of the statistical test in the text and figures, and its description in the Methods.

Line140: Please introduce GS (glucose starvation) when it is used for the first time (e.g. Line74, 77, 98...) and use it consistently. But I would rather avoid this abbreviation, as the authors use both “glucose starvation” and “glucose deprivation” in this manuscript.

Thank you for drawing our attention to this. We corrected this issue by avoiding the abbreviation GS, and by giving preference to ‘glucose starvation’.

Line140+146-150: These statements would be strengthened, if the authors added a quantification of Hsp104 protein levels upon different starvation conditions and strains using immunoblotting.

We agree with this comment and we performed an additional experiment to provide an answer. Instead of immunoblotting, we have performed qPCR measurement of Hsp104 gene expression in different conditions since we had cDNA stored from the same samples on which we performed imaging. The results show that, when under the control of the native promoter, Hsp104 undergoes, in average, 6-fold increase in expression level. However, while under the control of TEF1 promoter, at 2% glucose the Hsp104 expression is almost 2-fold larger compared to Hsp104 under the control of native promoter. During starvation in 0.2% and 0.02% glucose, the expression of Hsp104 is decreased approximately to the level of the Hsp104 under the control of the native promoter in 2% glucose. We have included these results in the manuscript.

Line142: Refer to Suppl. Fig. 2 C instead of Suppl. Fig. 2

Done.

Line144: Please include a statistical analysis of these data to use “trend of significant aggregate growth”.

As explained above, we have taken care of this issue.

Response to Reviewers' Comments

Line163ff: As data described here are already shown in Fig.1, either implement this subsection in the one above or split the graphs to create two independent figures. If the second option is chosen, please add control conditions as well.

We have now taken care of this issue by adding this subsection to the one above.

Line163ff: Please provide a description of the ATP/ADP/GTP treatment in the "Methods" section and state the concentration and length of incubation.

We have now included this and other relevant information into the Methods section.

Line167: The timeframe of this experiment is not clear in this description. Was DMSO and/or ATP added after 60 min of incubation in the respective starvation medium? If so, please adapt the labelling in Suppl. Fig. 4.

We thank the Reviewer for drawing our attention to this problem. In the previous version of the manuscript, we have made several mistakes in the presentation of these results. The ATP was indeed added after 60 minutes of starvation, and was present in the culture for the final 30 minutes of starvation. Therefore, the first time-point at which the measurement of ATP level is reported is in fact 62 minutes, etc. We have adapted the labeling of the x-axis in this figure.

Line173ff: As mentioned earlier, either include this paragraph in the subsection above or split Figure 1 and add respective controls.

We have now taken care of this issue by including the subsection in the one above.

Line179: Please briefly provide rationale, why ADP and GTP were added.

We have now included a rationale of the analysis of aggregate abundance in the presence of GTP and ADP into the manuscript.

Line198: Please introduce the mutants better. Even though these mutants are unable to hydrolyse ATP in the respective NBD domain, the second domain is still functional and, according to in vitro measurements, the ATP hydrolysis rate increases by 300% in the E285Q mutant and its inhibition by ADP is comparable to wildtype Hsp104 (PMID 27223323; PMID 17543332). Further, the E687Q mutant should still possess similar ATPase activity as the wildtype, while its inhibition by ADP is reduced (PMID 27223323). It could thus be a possibility to inhibit wt Hsp104 and E687Q-Hsp104 by adding ADP or assessing cellular ADP levels.

We thank the Reviewer for the suggestions of literature and information that would be relevant to include in the introduction of the mutants. We have included all suggested references, as well as mentioned and discussed the relevant information related to each of the studied mutants.

Line199: Please briefly introduce the term HAP mutants (as used in Jackrel et al 2014)

Response to Reviewers' Comments

We added a better introduction of the HAP mutants and cited related literature.

Line220: A quantification of aggregate number per cell would support the notion of aggregates fusing together in these strains.

This is a very interesting suggestion, and we have now included the results of the per-cell aggregate number analysis into the manuscript as Supplementary Fig. 6. The results show that while in the Hsp104^{E285Q} mutant, a smaller number of larger aggregates is present during glucose starvation. This may imply that the NBD2 domain (still functional in Hsp104^{E285Q} mutant) may play a role in preventing the aggregate fusion in Hsp104^{E285Q} mutant at 2% glucose, a function that may be hampered by low glucose and low ATP. This will be a topic of our further studies.

Line233: The observed phenotypic differences between the mutants could also arise from their different ability to hydrolyse ATP and the different degree of inhibition by ADP. Please add a section to the Discussion, where cellular ADP levels and thus inhibition of Hsp104-variant are brought into context. Further, please provide an explanation why addition of external ADP did not inhibit Hsp104 activity.

We have now added a paragraph discussing these points into the Discussion section.

Line244: Please exchange the word “tagged” with “decorated” or a similar term.

Done.

Line248: Please state that 0.02% glucose was used in this experiment.

Done.

Line248: This stands in contrast to ATP measurements shown in Suppl. Fig. 4A (where the minimal value assessed was around 4 mM after 30 min). Please briefly provide an explanation for this divergence. Further, the minimum is reached after 60 min at about 1 mM.

We are grateful to the Reviewer for drawing our attention to this issue, since we made several mistakes in the presentation of these results. Indeed, the ATP was added to the starving cells after 60 minutes of starvation and remained present for the final 30 minutes of the starvation period. We have now corrected the labeling on the x-axis. Importantly, we corrected also the actual plotted data, which in the previous version corresponded to the time points previously indicated on the x-axis. So, after 30 minutes, the ATP level is indeed around 4 mM, however, after 90 minutes of 0.02% glucose starvation is less than 1 mM, consistently with the results in other figures. Once again, we thank the Reviewer for noticing this issue.

Line249: Rather than using “left and right” panel, split this subpanel into A and B (same for Fig. 3 E).

We have taken care of this and split the previous panel into subpanels A and B.

Line250: Please provide the median of aggregate diameter.

Response to Reviewers' Comments

Done.

Line255: As exemplified above: Even though one Walker B domain is inactive, the other can hydrolyse ATP. Thus, avoid using the term “inability” for the whole protein, rather refer only to NBD2.

We have now taken care of this issue.

Line273/274: Please rephrase this statement.

Done.

Line275: To substantiate this hypothesis, it would be helpful to quantify the number of aggregates per cell over time and compare wt-Hsp104 with mutant-Hsp104 upon glucose starvation and control conditions.

This is a very interesting suggestion, and we have now included the results of the suggested analysis as Supplementary Fig. 7. The results show that in the Hsp104^{E687Q} mutant, the aggregate number per cell is increasing up to ~5 per cell while in the WT the dynamics of per-cell aggregate number clearly reflects a series of aggregate fusion events over time. Details of this process will be a subject to further study.

Line279: Please briefly state that GdnHCl inhibits Hsp104 activity and provide a reference.

This issue was taken care of.

Line279 and Suppl. Fig. 5: A wildtype control is necessary in this experiment.

We have now added the WT control to these results.

Line294: Please state, which promoter was used for the endogenous Hsp104-mCherry chimera.

We have now added the required information in this paragraph, as well as the other places in the text where it was missing.

Line298: To provide quantitative data on colocalization, assessment of e.g. Mander's overlap coefficient and Pearson's correlation coefficient would be suitable (easily done with Image J plugins).

We performed the calculations of Pearson's correlation coefficient for all colocalization analyses in the manuscript, and added the results in supplementary information.

Line310: Please include the respective control and avoid referring to datasets in other figures.

We have now added the control, and also taken care of the issue of referring to datasets from other figures.

Line319: The theory should be substantiated by colocalization experiments (including quantification as

described

above).

We performed the calculations of Pearson's correlation coefficient for all colocalization analyses in the manuscript, and added the results in supplementary information.

Line327: Please be consistent with abbreviations: Stress granules – SG was introduced in the introduction

We took care of this issue.

Line330ff: Please provide a colocalization quantification (as described above).

We performed the calculations of Pearson's correlation coefficient for all colocalization analyses in the manuscript, and added the results in supplementary information.

Line337: The abbreviation SG was introduced earlier.

This issue was taken care off.

Line338: Please provide a reference for Nrp1 as marker for SG.

The reference was added.

Line341: Please provide a colocalization quantification, as described above.

We performed the calculations of Pearson's correlation coefficient for all colocalization analyses in the manuscript, and added the results in supplementary information.

Line358: It would be interesting to see to what extent A2C increases aggregate length compared to control cells (see comment on Suppl. Fig. 7)

This is a very interesting point and we have now added the analysis of the A2C aggregate sizes into the manuscript. Briefly, the results show that the aggregate sizes are in general larger in the presence of A2C than those in the control. Furthermore, we present evidence that an additional aggregate type is formed in the presence of A2C. These results are now a part of Supplementary Fig. 10.

Line363ff: Please do not include the description of a new figure (Suppl. Fig. 8), before CHX treatment of Suppl. Fig. 7 has been described.

We have now taken care of this problem. We thank the Reviewer for drawing our attention to this.

Line373: Please avoid referring to data of other figures and include the respective control.

This issue was taken care of across the entire manuscript.

Response to Reviewers' Comments

Line405: It would be interesting to also include a condition, where Rp-cAMPS is added to cells prior to shifting them to starvation conditions to assess aggregate abundance and size.

We agree with the Reviewer that this would be an interesting experiment to perform. However, due to the volume of work we have decided not to perform this experiment in the context of this study, but rather in our future studies. We find the presented results with the Rp-cAMPS to be sufficient to corroborate the presented conclusions.

Lin408: Referring to Fig. 6 A, B does not fit here.

We corrected this error.

Line411: Please rephrase this statement for more clarity.

Done.

Line446: The assessment of cytosolic pH using pHluorin would require at least measurement of signal intensity at two different wavelengths (or other possibilities as described in the "Methods" section in more detail), to exclude any effects from different expression levels.

We are grateful to the Reviewer for this important suggestion. We have resolved this issue by measuring the protein level of GFP using immunoblotting. The results revealed similar GFP levels in different conditions, and we have therefore not modified our conclusions. The results of the Western blot are included in the manuscript as Supplementary Fig. 15b.

Line450: Even though Pma1 is a dominant proton pump, there are other possible explanations why the cytosol acidifies upon shifting them to glucose starvation medium. As no experiments with Pma1 are presented here, I would rather move this paragraph to the Discussion, where Pma1 but also other possibilities can be discussed in more detail.

We thank the Reviewer for drawing our attention to this. We have now moved the Pma1 paragraph from Results to Discussion and discussed other possibilities.

Line455: Even though addition of DNP led to a change in pHluorin fluorescence intensity it may have many side effects since it also interferes massively with mitochondria (respiratory chain...) and immediately increases cAMP levels.

We are grateful to the Reviewer for drawing out attention to the possible side-effects of DNP. We agree that the side-effects of DNP may present a problem in our experiments, and we have added a description of potential side-effects in the Results section, as well as in Discussion. However, even though we cannot exclude the possibility of the DNP side-effects affecting our results, by carefully reconsidering this possibility we think it unlikely. At pH 5.5 (presence of DNP) in optimal glucose, indeed we observed the formation of multiple (>10) speckles per cell, characterized also by irregular shape, which we now

Response to Reviewers' Comments

quantified (Fig. 6e, 6f). In 2% glucose at pH 7 (presence of DNP), the aggregate abundance is low (~10% in total), and the aggregate shape is regular (Fig. 6d, 6e). Moreover, in 0.02% glucose at pH 7 (in the presence of DNP), the aggregate abundance is increased (~90% in total), but they are characterized by a regular shape (Fig. 6e, 6f). Therefore, we do not exclude that some of the observed effects may be a consequence of the DNP side-effects, but we do not see a consistent phenotype that could be attributed to the presence of DNP.

Line468: The observed phenotype at pH 5.5 in the presence of DNP rather hints at unspecific effects, interpret this phenotype with caution.

We have addressed this issue in the answer above. Again, we would like to thank the Reviewer for drawing our attention to the potential side-effects of DNP.

Line470ff: The same concerns regarding the use of DNP remain, but please provide a respective quantification or remove this paragraph.

We have, at this point, included additional representative images, as Figure 6e, as well as quantification of aggregate shapes, as Figure 6f. Using the Shape descriptors plugin for ImageJ, we quantified the roundness of the observed aggregates in the following experimental conditions: (i) 2% glucose; (ii) 0.02% glucose; (iii) 0.02% glucose, pH 7 in the presence of DNP; (iv) 2% glucose, pH 5.5 in the presence of DNP. While the first three condition give rise to round-shaped aggregates, the aggregates formed under the condition (iv) deviate from round shape. We shortened this paragraph in the manuscript, and we discuss these issues with much more caution, as described above.

Line493: It is unusual that only 20% of exponentially growing cells display acidified vacuoles. Please state the criteria used to distinguish between acidified vs non-acidified vacuoles in the "Methods" section.

This experiment was inspired by and designed according to the one described in Hughes and Gottschling, (Nature 492: 261-265, 2012). According to the findings described in their paper, only very young mother cells (1-3 generations old) display a bright vacuole under the described experimental conditions (exponentially growing culture stained with quinacrine). Therefore, we believe that 20% of cells we observed with quinacrine-stained vacuole is consistent with their findings. Further, we added a description in the Methods regarding the criteria to distinguish the acidified from non-acidified vacuoles.

Line494: Please avoid the term "physiological pH", as no direct and quantitative assessment of vacuolar pH is presented.

We have corrected this issue.

Line497: Please provide details on concA concentration and treatment duration in the "Methods" section.

Response to Reviewers' Comments

Done.

Line505: Even though autophagy requires a functional vacuole, the authors did not measure pH quantitatively and did not directly test aggregate clearance in autophagy-impaired mutants, thus I would either remove this hypothesis or provide experimental evidence.

We agree and we have removed the speculation regarding the involvement of autophagy.

Line514: As described below in further detail, it is necessary to check for any artefact by the kanamycin resistance. Thus, the kanamycin-cassette also needs to be introduced into wildtype cells and the experiments should be repeated.

We agree that this may be an issue. We have therefore performed a competition experiment between the wild type strains: one without the cassette (KanS), and the other one with it (KanR), in optimal glucose, as well as in 0.2% and 0.02% glucose. The results of these competition experiments are now included into the manuscript as Figure 7b. The results showed that there is no significant difference in the cellular fitness due to the kanamycin resistance.

Line536/537: Please rephrase this sentence, as the correlation here is not clear.

Done.

Line541: Please provide the quantification of average number of aggregates per cell (as also stated earlier).

We have now provided the quantification of the per-cell aggregate number and included the results into the manuscript.

Line545: The authors did not show that ATP prevented aggregation. Rather glucose (ATP) depletion led to accumulation of aggregates and its re-feeding resulted in dissociation. The authors may interpret their data but need to rephrase it accordingly.

We have rephrased the statement with more caution.

Line547: As described above, one NBD each remained active, thus the tested Hsp104 variants are not unable to hydrolyse ATP, as only one Walker B domain is mutated.

We thank the Reviewer for pointing this out, and we have now corrected this issue.

Line554: Please discuss the fact that E285Q mutant shows a 300% increase in ATPase activity in vitro.

We have now added a discussion of 300% increase in ATPase activity of the E285Q mutant in vitro.

Response to Reviewers' Comments

Line561: Please also discuss respective in vitro measurements of the E687Q mutant.

We added a discussion of the mentioned measurements.

Line568: Even though DNP indeed leads to immediate intracellular pH adaptation to the extracellular milieu, other effects include increase in cAMP levels Please mention other consequences here.

We added a brief discussion regarding the potential side-effects of the DNP treatment.

Line570: This statement is not substantiated by experimental evidence provided in this paper.

We modified the statement in question.

Line585: Please modify the summary figure by adding more detail. As it is now, it does not help to recapitulate the main findings of the manuscript.

We have now excluded this summary figure from the manuscript, and at the same time we are including a figure as a graphical abstract, which we believe will much better convey the main findings of the manuscript

Figures

Fig.1A: Please provide asterisks in the graph and state, which bars were compared to provide the respective significance level.

Done.

Figure 1: If DMSO was added here as well to enable ATP/ADP/GTP entering the cells, please state this in "Methods" or in the figure legend.

We have now added this information into the Figure caption, as well as the Methods.

Suppl. Fig. 1A: Please provide significance asterisks for all brackets given (e.g. between 0.2% glucose 30 min and 60 min)

Done.

Suppl. Fig. 1B: please exchange "WT S288C" to "2% glucose" or "control". Further, please refer to the commonly-used style of setting gene names in italics (e.g. HSP42).

Done.

Suppl. Fig. 3B: In this graph, the aggregate change upon DMSO treatment is of interest, please include a statistical analysis between untreated/treated cells as well.

Response to Reviewers' Comments

Done.

Suppl. Fig. 4: Please add a control condition without DMSO, as especially the ATP content of cells incubated in 0.02% glucose for 30 min differs from values in Suppl. Fig. 1A (4 mM vs. 2.5 mM).

We have now added the control conditions without DMSO. We have explained elsewhere in this letter that what appears as a discrepancy on the ATP levels between the measurements in Suppl. Fig. 1a and Suppl. Fig. 4, was in fact a mistake in plotting data from our side. We have now corrected this issue.

Fig. 2 C, D: I would recommend to statistically analyse the aggregate diameter.

As mentioned previously, we have now performed Kolmogorov-Smirnov test to compare the aggregate diameter distributions.

Figure2: If DMSO was added here as well to ensure ATP uptake, please state it in the figure legend. Further, the legend does not fit to the data presented.

We have now stated this in the Figure legend.

Fig. 3 A: Please depict the aggregate diameter in nm, as the main text refers to their absolute size.

We have corrected this issue, the aggregate diameters are presented in nm.

Fig. 3C, D: This figure is hard to read, please use either different colours for the legend description "assembly" and "disassembly" or depict assembly and disassembly curves behind each other (starting with 90 min of starvation, followed by 90 min of re-feeding).

We have now used different color and symbols for assembly and disassembly.

Suppl. Fig 5B: please consider comments on Fig 3A.

Done.

Figure 4,5: Please avoid red-green combinations in micrographs (as stated in general comments).

The colors of the red and green channel images have been converted into black and white. The merged images remain in color.

Fig. 4B: Please include the wildtype control.

We have added the WT control.

Fig. 4C: As stated earlier, please provide a statistical analysis here.

Response to Reviewers' Comments

Throughout the manuscript, the Kolmogorov-Smirnov test was used to compare aggregate size distributions and calculate the statistical significance.

Suppl. Fig. 7: Please add the control condition without any treatment.

Done.

Suppl. Fig. 7 B, C: Please provide a statistical analysis.

Done.

Fig. 6: Please add the respective control strains. Further, try to improve the comprehensibility of the graphs (implement comments above, especially clearly state, which colour/shape represents what)

We have added the controls to this figure. Moreover, we believe we have also improved the comprehensibility of the graphs and entire figure.

Fig. 6A: Please provide an explanation, why no cytosolic Hsp104 signal can be observed upon addition of Rp-cAMPS.

We believe that the absence of cytosolic signal is due to the preferential localization of the Hsp104-GFP into the aggregates, which makes the aggregates much brighter, compared to the cytosol. Under the conditions of exposure time used to image this strain, the cytosol appears without the signal of Hsp104-GFP. We have now added this potential explanation into the Results section.

Suppl. Fig. 9: Please clearly describe the conditions used (as already stated above).

We have now added a description of conditions applied in this experiment to the figure legend, as well as in the Methods section.

Suppl. Fig. 11: Please provide the respective untreated control condition.

Done.

Figure 7 B, C: Please provide an explanation, why the competitive fitness of HSP42KO/ Hsp104mutant cells is around 0.9 resp. 0.8 in (B) and above 1 in (C).

We are thankful to the Reviewer for drawing out attention to this problem. Confused by a large number of competition experiments, controls and combinations of strain, we plotted the wrong data. We have now corrected this issue, and made sure that we plotted the correct data, which actually correspond to the indicated condition. We apologize for the sloppiness.

Methods

Line609: Even though the strains can be found in Suppl. Table 1, the respective growth medium is not listed there.

We added the information on the growth medium into the Supplementary Table 1.

Line610: Please provide the composition of the used growth medium.

Done.

Line613: If control cells (in 2% glucose) were also treated the same way as starved cells (including centrifugation and washing steps), please state it here.

The control cells were indeed treated in the same way as the starved cells. We have now added this statement to the description of the experiment.

Line617: Did the authors use both WT and HAP mutants in their study? If so, please state it in the respective figure legend or rephrase this sentence here.

HAP mutants were used only in the part of the study where we analyze the aggregation of the Walker B mutants. We have now added an adequate explanation in the text.

Line645: To avoid artefacts by different expression levels in different media (2% - 0.2% - 0.02% glucose), either use the technique of assessing the fluorescence intensity at two wavelengths (e.g. as described in your reference 45), design a tandem fluorophore construct composed of pH-sensitive pHluorin and pH-insensitive mCherry (similar as used in Rosella) or check respective expression levels with immunoblotting.

We have checked the respective expression levels of GFP in different conditions by immunoblotting. We have now added these results into the manuscript, and adequately described the applied methods.

Line656: Please provide more details on your protocol, as this kit from Abcam is designated for mammalian cells and the protocol might require adaption for yeast, which might help other researchers making use of your method.

We have now added more details regarding the used protocol in the Methods section.

Line662: Please state the glucose concentration you used for preparation of the agar slides for microscopy.

We have now added this information.

Line737: The competition assay would benefit, if both strains were inoculated to the same optical density/cell number. Alternatively, please provide growth curve measurements to ensure that both strains reach the same optical density at "saturation".

Response to Reviewers' Comments

We agree with the Reviewer regarding this issue. Therefore, from the very beginning we made sure that we inoculated the competing strains at the same cell number. We have now added the additional explanation into the manuscript.

Line738: It would be vital to check, if both strains are equally represented in the culture before transferring them to different starvation conditions (e.g. by plating them before treatment as well).

We agree with the Reviewer and we have in fact checked for equal representation of the both strains from the beginning. We failed to mention it in the previous version of the manuscript, and we have now included this statement.

Line746ff: To avoid artefacts by genomic integration of the kanamycin cassette, it is advisable to add the selection marker to wildtype cells and repeat this experiment.

We agree that this may be an issue. We have therefore performed a competition experiment between the wild type strains: one without the cassette (KanS), and the other one with it (KanR), in optimal glucose as well as in 0.2% and 0.02% glucose. The results of these competition experiments are now included into the manuscript as Figure 7b. The results showed that there is no significant difference in the cellular fitness due to the presence of the resistance cassette.

REVIEWERS' COMMENTS:

Reviewer #2 (Remarks to the Author):

My first major concern was not solved completely, because the experiment I suggested had been done long before, but cannot be shown. I accept the explanation of why the authors prefer not to show this result.

This first concern addressed the following issue. The authors concluded that the majority of Hsp104-GFP foci observed after GS represent Q-bodies. This conclusion was based on a correlation. The number of foci found in the Hsp104-GFP "WT" strain equaled the sum of foci visualized with an Hsp42-GFP marker (should visualize all Q-bodies) and the residual foci formed in an hsp42-deletion background (those foci would be additional compartments to which Hsp104-GFP localizes upon GS). I suggested to directly show the localization of Hsp104-GFP to Q-bodies in an experiment where Hsp42 (Q-body marker) co-localizes with Hsp104 foci formed upon starvation. The authors explain in their response letter that they found Hsp42-mcherry foci to co-localize upon starvation with some Rnq1 foci and some Nrp1 foci (SG marker) as well. Therefore, they claim that Hsp42-mcherry cannot be used with confidence as a marker solely for Q-bodies.

I don't know to what extent Hsp42 also stains SGs. It may be a minor fraction, but the authors should be careful with the quantification of Q-bodies based solely on the Hsp42 marker. However, they do indeed never stress that the percentage of Hsp42-GFP foci must represent entirely Q-bodies. Furthermore, Hsp42 is clearly recognized as a Q-body marker. Therefore, the loss of a high percentage of Hsp104-GFP foci after deletion of Hsp42 is a strong indication that Hsp104-GFP localizes to Q-bodies.

Regarding my second major point, I was concerned that the experiment using A2C would not clearly show that the foci formed upon GS really contained misfolded proteins, because next to the previously observed compartments that formed upon GS (SG, Q-bodies), additional Hsp104-GFP positive foci formed. However, the finding that is now presented in Fig S10 that nearly all Hsp104-GFP foci increased in diameter after A2C convinces me that upon A2C, there are indeed more proteins targeted also to Q-bodies and SGs and not only to the newly appearing foci.

My additional minor concerns have all been addressed appropriately. Only one minor thing is still valid. It is now explained what "HAP" means, but I feel it does not come across well why this mutant was used here. Maybe the authors could point out why they used the HAP variant here.

Reviewer #3 (Remarks to the Author):

During the revision, the manuscript has significantly improved and my main concerns have been addressed. Yet some minor changes of mainly textual character would improve the manuscript before publication:

Main text

- General: While the respective WT controls were added (e.g. Suppl. Fig. 10g), a statement is needed that these controls were assessed in the same experiment. As it is now, it is not clear, since control and treatment group are presented in different subpanels.
- L. 80: Please introduce the abbreviations NBD1/2 at first use.
- L. 219: In this paragraph, a reference to Fig. 2 would be needed or the paragraph needs to be merged to the one below.
- L. 299ff: While the authors state that aggregate fusion requires NBD2 domain activity, the size of the aggregates is not changed when compared to WT (Fig. 3a, f). Please comment on this either in the Results or in the Discussion.
- L. 382: Either use stress granules or the abbreviation SG
- L. 630: This statement needs a reference.

- L. 643: Even though this sentence seems logical, the data presented in Fig. 2 shows that neither E285Q nor E687Q respond to external ATP.
- L. 764: Please be consistent with centrifugation speeds and preferentially use g/rcf.

Figures and legends

- General: Figure legend 5/6/7 and others: Please adapt these figure legends to the commonly used style of figure legends (rather describing experimental/statistical details instead of discussing results).
- General: Either use standard deviation or SD throughout the manuscript
- Fig. 1b, Suppl. Fig. 4c and following: The y-axis might need re-labelling, e.g. number of cells.
- Fig. 3d: This subpanel, where assembly and disassembly are shown, is confusing, as aggregate assembly is described in the text, while disassembly is mentioned in the next paragraph. Please merge these passages or put the subpanels in a different order.
- Fig. 4b would benefit if the control (WT) would be shown first
- Fig 6: Please be consistent and use the same timeframe for representative images and quantification
- Fig. 6c and d: The respective controls are missing.
- Figure 7a: Please avoid red-green combinations. Color-blind readers/ readers with red-blue weakness will not be able to differentiate between WT and mutant yeast cells mixed in culture/ on plates.
- Suppl. Fig. 3b and c: Please correct the labelling, since ATP only and not DMSO only have been used (as described in the figure legend).
- Suppl. Fig. 10d and f: Please label these micrographs clearer.
- Suppl. Fig. 10f. These micrographs would benefit from increasing the brightness.
- Suppl. Fig. 17b: Please increase the brightness of the micrographs
- Suppl. Fig. 10f and i: Here, the respective wildtype control is missing
- Suppl. Fig 14: Please put the respective control conditions next to the treatment and state (if data are not in the same subpanel) that untreated/treated cells were assessed in the same experiment.

REVIEWERS' COMMENTS:

Reviewer #2 (Remarks to the Author):

My first major concern was not solved completely, because the experiment I suggested had been done long before, but cannot be shown. I accept the explanation of why the authors prefer not to show this result.

This first concern addressed the following issue. The authors concluded that the majority of Hsp104-GFP foci observed after GS represent Q-bodies. This conclusion was based on a correlation. The number of foci found in the Hsp104-GFP "WT" strain equaled the sum of foci visualized with an Hsp42-GFP marker (should visualize all Q-bodies) and the residual foci formed in an hsp42-deletion background (those foci would be additional compartments to which Hsp104-GFP localizes upon GS). I suggested to directly show the localization of Hsp104-GFP to Q-bodies in an experiment where Hsp42 (Q-body marker) co-localizes with Hsp104 foci formed upon starvation. The authors explain in their response letter that they found Hsp42-mcherry foci to co-localize upon starvation with some Rnq1 foci and some Nrp1 foci (SG marker) as well. Therefore, they claim that Hsp42-mcherry cannot be used with confidence as a marker solely for Q-bodies. I don't know to what extent Hsp42 also stains SGs. It may be a minor fraction, but the authors should be careful with the quantification of Q-bodies based solely on the Hsp42 marker. However, they do indeed never stress that the percentage of Hsp42-GFP foci must represent entirely Q-bodies. Furthermore, Hsp42 is clearly recognized as a Q-body marker. Therefore, the loss of a high percentage of Hsp104-GFP foci after deletion of Hsp42 is a strong indication that Hsp104-GFP localizes to Q-bodies.

We appreciate the Reviewer accepting our explanation. Hsp42 can be used as a marker of Q-bodies in the sense that they will disappear in the Hsp42 knock-out. However, we are currently also investigating the role of Hsp42 in the stress granule assembly and especially dissolution, as well as its colocalization with stress granules. We are confident we will be able to report those results soon.

Regarding my second major point, I was concerned that the experiment using A2C would not clearly show that the foci formed upon GS really contained misfolded proteins, because next to the previously observed compartments that formed upon GS (SG, Q-bodies), additional Hsp104-GFP positive foci formed. However, the finding that is now presented in Fig S10 that nearly all Hsp104-GFP foci increased in diameter after A2C convinces me that upon A2C, there are indeed more proteins targeted also to Q-bodies and SGs and not only to the newly appearing foci.

We thank the Reviewer for the positive feedback.

My additional minor concerns have all been addressed appropriately. Only one minor thing is still valid. It is now explained what "HAP" means, but I feel it does not come across well why this mutant was used here. Maybe the authors could point out why they used the HAP variant here.

We have now introduced a brief comment to the main text to address this concern.

Reviewer #3 (Remarks to the Author):

Response to Reviewers

During the revision, the manuscript has significantly improved and my main concerns have been addressed. Yet some minor changes of mainly textual character would improve the manuscript before publication:

Main text

- General: While the respective WT controls were added (e.g. Suppl. Fig. 10g), a statement is needed that these controls were assessed in the same experiment. As it is now, it is not clear, since control and treatment group are presented in different subpanels.

We have now added such statements throughout the manuscript.

- L. 80: Please introduce the abbreviations NBD1/2 at first use.

Done.

- L. 219: In this paragraph, a reference to Fig. 2 would be needed or the paragraph needs to be merged to the one below.

We have merged the paragraph to the one below.

- L. 299ff: While the authors state that aggregate fusion requires NBD2 domain activity, the size of the aggregates is not changed when compared to WT (Fig. 3a, f). Please comment on this either in the Results or in the Discussion.

We have now added a comment on this issue in the Results section.

- L. 382: Either use stress granules or the abbreviation SG

We have corrected this issue to use the abbreviation SG.

- L. 630: This statement needs a reference.

Done.

- L. 643: Even though this sentence seems logical, the data presented in Fig. 2 shows that neither E285Q nor E687Q respond to external ATP.

We have added a brief explanation of this issue.

- L. 764: Please be consistent with centrifugation speeds and preferentially use g/rcf.

We now use xg throughout the manuscript.

Figures and legends

Response to Reviewers

- General: Figure legend 5/6/7 and others: Please adapt these figure legends to the commonly used style of figure legends (rather describing experimental/statistical details instead of discussing results).

We have worked extensively to improve the quality of our figure legends, while keeping in mind also the 350 word count limit.

- General: Either use standard deviation or SD throughout the manuscript

Done.

- Fig. 1b, Suppl. Fig. 4c and following: The y-axis might need re-labelling, e.g. number of cells.

We have carefully considered this suggestion, however, the y-axis actually displays the aggregate number in each bin displayed on the x-axis.

- Fig. 3d: This subpanel, where assembly and disassembly are shown, is confusing, as aggregate assembly is described in the text, while disassembly is mentioned in the next paragraph. Please merge these passages or put the subpanels in a different order.

We have now merged these paragraphs.

- Fig. 4b would benefit if the control (WT) would be shown first

Done.

- Fig 6: Please be consistent and use the same timeframe for representative images and quantification

We have fixed this issue.

- Fig. 6c and d: The respective controls are missing.

We have considered this suggestion, however, the panel 6c is actually a control for 6d.

- Figure 7a: Please avoid red-green combinations. Color-blind readers/ readers with red-blue weakness will not be able to differentiate between WT and mutant yeast cells mixed in culture/ on plates.

We have now used blue and pink to replace red and green.

- Suppl. Fig. 3b and c: Please correct the labelling, since ATP only and not DMSO only have been used (as described in the figure legend).

Response to Reviewers

We have carefully analyzed this issue, and we thank the Reviewer for pointing this out. We have previously made a mistake in the figure legend, which we have now corrected.

- Suppl. Fig. 10d and f: Please label these micrographs clearer.

Done.

- Suppl. Fig. 10f. These micrographs would benefit from increasing the brightness.

Done.

- Suppl. Fig. 17b: Please increase the brightness of the micrographs

Done.

- Suppl. Fig. 10f and i: Here, the respective wildtype control is missing

We have now included the wild-type control.

- Suppl. Fig 14: Please put the respective control conditions next to the treatment and state (if data are not in the same subpanel) that untreated/treated cells were assessed in the same experiment.

We have now taken care of this issue.